# ON THE CONVERGENCE OF MSGD AND ADAGRAD FOR STOCHASTIC OPTIMIZATION

**Ruinan Jin**
LSC, NCMIS, Academy of Mathematics and Systems Science
Chinese Academy of Sciences, Beijing 100190, China
School of Mathematical Sciences
University of Chinese Academy of Sciences, Beijing 100049, China

**Yu Xing**
Division of Decision and Control Systems
KTH Royal Institute of Technology
SE-100 44 Stockholm, Sweden

**Xingkang He** *
Department of Electrical Engineering
University of Notre Dame, IN, USA

## ABSTRACT

As one of the most fundamental stochastic optimization algorithms, stochastic gradient descent (SGD) has been intensively developed and extensively applied in machine learning in the past decade. There have been some modified SGD-type algorithms, which outperform the SGD in many competitions and applications in terms of convergence rate and accuracy, such as momentum-based SGD (mSGD) and adaptive gradient algorithm (AdaGrad). Despite these empirical successes, the theoretical properties of these algorithms have not been well established due to technical difficulties. With this motivation, we focus on convergence analysis of mSGD and AdaGrad for any smooth (possibly non-convex) loss functions in stochastic optimization. First, we prove that the iterates of mSGD are asymptotically convergent to a connected set of stationary points with probability one, which is more general than existing works on subsequence convergence or convergence of time averages. Moreover, we prove that the loss function of mSGD decays at a certain rate faster than that of SGD. In addition, we prove the iterates of AdaGrad are asymptotically convergent to a connected set of stationary points with probability one. Also, this result extends the results from the literature on subsequence convergence and the convergence of time averages. Despite the generality of the above convergence results, we have relaxed some assumptions of gradient noises, convexity of loss functions, as well as boundedness of iterates.

## 1 INTRODUCTION

In recent years, the rapid development of machine learning has stimulated a lot of applications of optimization algorithms to employing tremendous data in practical scenarios. Many optimization algorithms of machine learning are based on gradient descent (GD). A typical GD algorithm, minimizing a loss function $g(\theta) \in \mathbb{R}$ via seeking an $N$-dimensional real-valued parameter $\theta^* = \arg \min_{\theta \in \mathbb{R}^N} g(\theta)$, writes as follows

$$\theta_{n+1} = \theta_n - \varepsilon_n \nabla_{\theta_n} g(\theta_n), \tag{1}$$

where $\theta_n$ is the estimate of $\theta^*$ at step $n$, $\varepsilon_n$ is a positive step size (learning rate) to be designed, and $\nabla_{\theta_n} g(\theta_n)$ stands for the gradient of $g(\theta_n)$ at step $n$. When certain conditions are satisfied, $\theta_n$ in

---

*Corresponding author: xhe9@nd.edu

equation 1 will converge to the optimal solution. However, since the computation of $\nabla_{\theta_n} g(\theta_n)$ relies on all data at each step, it is inefficient to apply GD-based algorithms (e.g., equation 1) when the data amount is very large. Therefore, more and more attention has been given on how to accelerate the GD-based algorithms. One of the attempts is stochastic gradient descent (SGD) which originated from Robbins & Monro (1951). Instead of calculating $\nabla_{\theta_n} g(\theta_n)$ over all data, SGD algorithms use a relatively small proportion of data to estimate the gradient, i.e., $\nabla_{\theta_n} g(\theta_n, \xi_n)$ where $\xi_n$ is a random vector introduced to choose a subset of data for each update. Then the SGD modified from GD equation 1 is as follows

$$\theta_{n+1} = \theta_n - \varepsilon_n \nabla_{\theta_n} g(\theta_n, \xi_n). \tag{2}$$

Besides reducing gradient computation, $g(\theta_n, \xi_n)$ can be used as an estimate of the gradient for the scenarios where the accurate gradient is unavailable due to external noises.

In recent years, SGD has shown its prominent advantages in dealing with high dimensional optimization problems such as regularized empirical risk minimization and training deep neural networks (see, e.g. Graves et al. (2013); Nguyen et al. (2018); Hinton & Salakhutdinov (2006); Krizhevsky et al. (2012) and references therein). In Nguyen et al. (2018); Bottou (2012), the convergence and the convergence rate of SGD have been analyzed.

Despite outstanding successes of SGD, it usually has a relatively slow convergence rate induced by random gradients and it provides fluctuating iterates in the learning process. In order to improve the theoretical and empirical performance of SGD, there have been a number of investigations, including three typical algorithms for accelerating the convergence rate: (1) The momentum-based stochastic gradient descend (mSGD), which reduces the update variance by averaging the past gradients (Polyak, 1964); (2) The adaptive gradient algorithm (AdaGrad), which replaces the step size of SGD with adaptive step size (Duchi et al., 2011); (3) Adaptive momentum gradient algorithm (Adam), which integrates mSGD and AdaGrad (Kingma & Ba, 2015). However, the convergence results of these algorithms have not been well established, especially for non-convex loss functions. In this paper, we focus on the investigation of mSGD and AdaGrad. We believe that the convergence analysis results in this paper are helpful for the future research on the convergence or generalization of Adam and other stochastic optimization algorithms.

The technique of mSGD is originally developed by Polyak (Polyak, 1964) for the acceleration of convergence rate of gradient-based methods. A typical expression of mSGD is as follows (Zareba et al., 2015; Sutskever et al., 2013)

$$v_n = \alpha v_{n-1} + \varepsilon_n \nabla_{\theta_n} g(\theta_n, \xi_n), \quad \theta_{n+1} = \theta_n - v_n, \tag{3}$$

where $\alpha \in [0, 1)$ and $\varepsilon_n > 0$ are relatively momentum coefficient and step size (learning rate), respectively. Another type of mSGD with the name of stochastic heavy ball (SHB) has also been studied (Gitman et al., 2019; Gupal & Bazhenov, 1972),

$$v_n = \beta_n v_{n-1} + (1 - \beta_n) \nabla_{\theta_n} g(\theta_n, \xi_n), \quad \theta_{n+1} = \theta_n - \gamma_n v_n, \tag{4}$$

where $\beta_n \in (0, 1)$ and $\gamma_n$ are relatively momentum coefficient and step size (learning rate). In recent years, mSGD has been widely employed in the applications of deep learning such as image classification (Krizhevsky et al., 2012), fault diagnosis (Tang et al., 2018), statistical image reconstruction (Kim et al., 2014), etc. Moreover, a number of variants on momentum are emerging, see, e.g., synthesized Nesterov variants (SNV) (Williams & Lovett, 2016), robust momentum (Cyrus et al., 2018), and PID-control based methods (An et al., 2018). The importance of momentum in deep learning has been illustrated in Sutskever et al. (2013) through experiments. However, the theoretical analysis for convergence and convergence rate of mSGD needs further investigation, especially for non-convex loss functions which are common in deep learning. Most existing results are established with guaranteed subsequence convergence or convergence of time averages[1] (see Yang et al. (2016); Gitman et al. (2019); Polyak (1977); Kaniovskii (1983), Li & Orabona (2020) and references therein). Nevertheless, there is a still distance to asymptotic convergence, which is usually more general and useful in practical applications requiring stable iterates for each realization. Sebbouh et al. (2020) studied the asymptotic convergence of mSGD with time-varying parameter $\alpha_n$, which however is less common than the static $\alpha$ in practical applications.

Convergence analysis for SGD unveils that the convergence rate heavily depends on the choice of step size $\varepsilon_n$ (Robbins & Monro, 1951; Nguyen et al., 2018), which may consume a huge amount

---

[1]The convergence definitions are given in Section 3.

of efforts on fine tune. In order to deal with this problem, the algorithm AdaGrad with adaptive step size is proposed in Duchi et al. (2010) and McMahan & Streeter (2010) concurrently. In the literature, there are two main forms of AdaGrad. One is based on the norm of gradients as follows (Streeter & Mcmahan, 2010)

$$S_n = S_{n-1} + \left\| \nabla_{\theta_n} g(\theta_n, \xi_n) \right\|^2, \quad \theta_{n+1} = \theta_n - \frac{\alpha_0}{\sqrt{S_n}} \nabla_{\theta_n} g(\theta_n, \xi_n), \tag{5}$$

where $\alpha_0 > 0$ is a constant. The other form is based on the coordinate-wise gradients (Duchi et al., 2010; Li & Orabona, 2019; 2020)

$$Q_n = Q_{n-1} + \nabla_{\theta_n} g(\theta_n, \xi_n)^T \nabla_{\theta_n} g(\theta_n, \xi_n), \quad \theta_{n+1} = \theta_n - \overline{\alpha}_0 Q_n^{-\frac{1}{2}} \nabla_{\theta_n} g(\theta_n, \xi_n),$$

where $\overline{\alpha}_0$ is a constant. In our paper we focus on the norm form (equation 5). In recent years, AdaGrad has shown its effectiveness in the field of sparse optimization (Duchi et al., 2013), tensor factorization (Lacroix et al., 2018), and deep learning (Heaton & Jeff, 2018). Some algorithm variants like RMSProp (Tieleman & Hinton, 2012) and SAdaGrad (Chen et al., 2018) are also studied. However, there are few results on the convergence of AdaGrad. Most of these results only prove subsequence convergence or convergence of time averages (see, e.g. Zou et al. (2019); Chen et al. (2019); Défossez et al. (2020); Ward et al. (2019)). Although Li & Orabona (2019) and Gadat & Gavra (2020) studied the asymptotic convergence of a modified AdaGrad algorithm, the result is not applicable to AdaGrad in equation 5. Thus, the asymptotic convergence of AdaGrad in equation 5 is still open.

In this theoretical paper, we aim to establish the convergence of mSGD and AdaGrad under mild conditions. The main contributions of this paper are three-fold:

- We prove that the iterates of mSGD are asymptotically convergent to a connected set of stationary points for possibly non-convex loss function almost surely (i.e., with probability one), which is more general than existing works on subsequence convergence.

- We quantify the convergence rate of mSGD for the loss functions. Through this convergence rate we can get a theoretical explanation of why mSGD can be seen as an acceleration of SGD. Moreover, we provide the convergence rate of mean-square gradients and connect it to the convergence of time averages.

- We prove the iterates of AdaGrad are asymptotically convergent to a connected set of stationary points almost surely for possible non-convex loss functions. The convergence result for the AdaGrad extends the subsequence convergence in the literature.

The remainder of the paper is organized as follows. In Section 2, we introduce the related work considering the convergence of mSGD and AdaGrad. The main results of the paper are given in Section 3, where we study the convergence and convergence rate of mSGD as well as the convergence of AdaGrad. Section 4 concludes the whole paper. Sections 5 and 6 are Code of Ethics and Reproducibility, respectively. The proofs are given in Appendix.

## 2 RELATED WORK

**Convergence of mSGD:** For the normalized mSGD (SHB), Polyak (Polyak, 1977; 1964) and Kaniovski (Kaniovskii, 1983) studied its convergence (subsequence convergence and convergence of time averages) properties for convex loss functions. Igor Gitman (Gitman et al., 2019) provided some convergence results of mSGD (SHB) for non-convex loss functions, but there is a considerable distance to the asymptotic convergence. Moreover, there is a requirement for uniform boundedness of a noise term in Gitman et al. (2019), i.e., $\mathbb{E}(\|\nabla_{\theta_n} g(\theta_n, \xi_n) - \nabla_{\theta_n} g(\theta_n)\|) \leq \delta$, which confines the application scope of mSGD (SHB). In addition, the designs of momentum coefficients in Polyak (1977; 1964); Kaniovskii (1983); Gitman et al. (2019) are not consistent with some practical applications (Smith et al., 2018; Sutskever et al., 2013). Therefore, the asymptotic convergence of mSGD for convex and non-convex loss functions needs further investigation.

**Convergence rate of mSGD:** Despite outstanding empirical successes of mSGD, there are few results on convergence rate of mSGD. In these results, Mai & Johansson (2020) studied a class of convex loss functions, and obtained a convergence rate of time averages without reflecting the role

of momentum parameter. Gitman et al. (2019) and Nicolas Loizou & Richtárik (2020) respectively investigated the asymptotic convergence rate of mSGD (SHB) by restricting to quadratic loss functions. Liu et al. (2020) studied the properties of SHB, where the relation between the loss function of SHB and the step size in every step was studied under the setting that $\alpha_n$ and $\beta_n$ are constants. The convergence rate of time averages was also studied. However, since the momentum parameters are not consistent with some applications (Smith et al., 2018; Sutskever et al., 2013) and the standard mSGD in equation 3 is not covered, further studies are needed.

**Convergence of AdaGrad:** In the original work for AdaGrad (Duchi et al., 2011), it was proved that AdaGrad can converge faster in the time averages sense if gradients are sparse and the loss function is convex. Similar results were established by Chen et al. (2019), and Ward et al. (2019). Zou et al. (2019) and Défossez et al. (2020) established convergence results in the subsequence sense. Asymptotic convergence was obtained in Li & Orabona (2019) for non-convex functions, but the form of the algorithm is no longer standard, as discussed in Duchi et al. (2011). Although such a change alleviates the difficulty in the proof of asymptotic convergence, it cannot be applied to the study of the AdaGrad in equation 5. Moreover, they required that a noise term is of point-wise boundedness (i.e., $\left\|\nabla_{\theta_n} g(\theta_n, \xi_n) - \nabla_{\theta_n} g(\theta_n)\right\| \leq \delta$, where $\delta$ is a positive constant), which however is relatively restrictive.

## 3 MAIN RESULTS

In this section, we provide the main results of this paper, including the analysis of convergence and convergence rate of mSGD in equation 2 and the analysis of convergence of AdaGrad in equation 5. In the following, $\mathbb{R}^N$ denotes the $N$-dimensional Euclidean space and $\|\cdot\|$ stands for the 2-norm, i.e., the Euclidean norm. To proceed, we need some definitions, consisting of asymptotic convergence, subsequence convergence, mean-square convergence, and convergence of time averages.

**Definition 1** *(Asymptotic convergence) A sequence $\{x_n\}$ is asymptotically convergent to a set $\mathbb{K}$, if* $\lim_{n \to +\infty} \left( \inf_{x \in \mathbb{K}} \|x_n - x\| \right) = 0$.

**Definition 2** *(Subsequence convergence) A sequence $\{x_n\}$ converges in subsequence to a set $\mathbb{K}$, if there exists at least one subsequence $\{x_{k_n}\}$ of $\{x_n\}$ such that* $\lim_{n \to +\infty} \left( \inf_{x \in \mathbb{K}} \|x_{k_n} - x\| \right) = 0$.

**Definition 3** *(Mean-square convergence) A stochastic sequence $\{x_n\}$ converges in mean square to a fixed vector $x$, if* $\lim_{n \to +\infty} \mathbb{E}(\|x_n - x\|^2) = 0$.

**Definition 4** *(Convergence of time averages) A stochastic sequence $\{x_n\}$ converges in time averages to a fixed vector $x$, if* $\lim_{T \to +\infty} \frac{1}{T} \sum_{n=1}^{T} \mathbb{E}(\|x_n - x\|^2) = 0$.

It is obvious that asymptotic convergence implies subsequence convergence, and that mean-square convergence ensures convergence of time averages, but not vice versa.

### 3.1 CONVERGENCE OF mSGD

In this subsection, with the help of some stochastic approximation techniques (Chen, 2006), we aim to prove that $\theta_n$ in equation 3 is asymptotically convergent to a connected component $J^*$ of the set $J := \{\theta \| \|\nabla_\theta g(\theta)\| = 0\}$ almost surely (a.s.) under proper conditions. When this connected component degenerates to a stationary point $\theta^*$, it holds that $\theta_n \to \theta^*$, a.s..

In contrast to the existing works of subsequence convergence (cf. Zou et al. (2019); Chen et al. (2019); Défossez et al. (2020); Ward et al. (2019)), we aim to prove that $\theta_n$ of mSGD in equation 3 is able to achieve asymptotic convergence. Since mSGD in equation 3 is a stochastic algorithm, we aim to establish its a.s. asymptotic convergence. To proceed, we need some reasonable assumptions with respect to noise sequence $\{\xi_n\}$ and loss function $g(\theta)$.

**Assumption 1** *Noise sequence $\{\xi_n\}$ are mutually independent and independent of $\theta_1$ and $v_0$, such that $g(x) = \mathbb{E}_{\xi_n}\left(g(x, \xi_n)\right)$ for any $x \in \mathbb{R}^N$.*

**Assumption 2** (***Loss function assumption***) *Loss function $g(\theta)$ satisfies the following conditions:*

*1) $g(\theta)$ is a non-negative and continuously differentiable function.*

*2) The set of stationary points of $\|\nabla_\theta g(\theta)\|$ is not an empty set, that is*
$$J := \{\theta \mid \|\nabla_\theta g(\theta)\| = 0\} \neq \emptyset.$$

*3) $\nabla_\theta g(\theta)$ satisfies the Lipschitz condition, i.e., there is a scalar $c > 0$, such that for any $x, y \in \mathbb{R}^N$*
$$\left\| \nabla_x g(x) - \nabla_y g(y) \right\| \leq c \|x - y\|.$$

*4) There is a scalar $M > 0$ such that for any $\theta \in \mathbb{R}^N$ and positive integer n,*
$$\mathbb{E}_{\xi_n} \left( \left\| \nabla_\theta g(\theta) - \nabla_\theta g(\theta, \xi_n) \right\|^2 \right) \leq M \left( 1 + g(\theta) \right). \tag{6}$$

Assumption 1 and conditions 1)–3) of Assumption 2 are common in the literature (Gitman et al., 2019). Assumption 2 does not pose any requirement on the convexity of $g(\theta)$. In other words, we allow any convex or non-convex loss functions $g(\theta)$ as long as they satisfy this assumption. Condition 4) corresponds to the condition in Shalev-Shwartz et al. (2011); Nemirovski et al. (2009); Hazan & Kale (2014); Gitman et al. (2019); Yang et al. (2016); Polyak (1977); Kaniovskii (1983) where the following inequality is assumed to hold for any $\theta \in \mathbb{R}^N$ and positive integer $n$,
$$\mathbb{E}_{\xi_n} \left( \left\| \nabla_\theta g(\theta) - \nabla_\theta g(\theta, \xi_n) \right\|^2 \right) \leq K, \ \forall \theta \in \mathbb{R}^N, \tag{7}$$

where $K$ is a positive scalar. Note that equation 6 reduces to equation 7 if $g(\theta)$ is uniformly upper bounded over the space $\mathbb{R}^N$. Since equation 6 does not need this uniform boundedness, it substantially extends equation 7 such that mSGD is also applicable to the scenarios with unbounded loss functions.

Different from deterministic GD-type algorithms with a constant step size, in order to ensure the convergence of mSGD, we need a decreasing step size for counteracting the randomness induced by noise $\{\xi_n\}$ (Gitman et al., 2019; Robbins & Monro, 1951). Specifically, we make the following assumption on step size $\varepsilon_n$ together with a fixed momentum coefficient $\alpha$.

**Assumption 3** *Momentum coefficient $\alpha \in [0, 1)$ and the sequence of step size $\varepsilon_n$ is positive, monotonically decreasing to zero, such that $\sum_{n=1}^{+\infty} \varepsilon_n = +\infty$ and $\sum_{n=1}^{+\infty} \varepsilon_n^2 < +\infty$.*

The setting of $\varepsilon_n$ in Assumption 3 is consistent with stochastic approximation for root seeking of functions (Chen, 2006) as well as some stochastic optimization algorithms (Gupal & Bazhenov, 1972; Polyak, 1977; Kaniovskii, 1983; Gitman et al., 2019). An explicit example of step size $\varepsilon_n$ satisfying Assumption 3 is $\varepsilon_n = 1/n$. Although SHB in equation 4 shares a similar expression as mSGD, the provided conditions of step sizes in Gupal & Bazhenov (1972); Polyak (1977); Kaniovskii (1983); Gitman et al. (2019) are not applicable to the general cases of mSGD with static $\alpha$ (Smith et al., 2018; Sutskever et al., 2013).

In fact, SHB in equation 4 has a similar expression as mSGD in equation 3. Multiplying $\gamma_n$ on both sides of the first equation of equation 4 yields
$$\gamma_n v_n = \frac{\gamma_n}{\gamma_{n-1}} \beta_n (\gamma_{n-1} v_{n-1}) + \gamma_n (1 - \beta_n) \nabla_{\theta_n} g(\theta_n, \xi_n)$$
$$\theta_{n+1} = \theta_n - \gamma_n v_n. \tag{8}$$
We treat $\gamma_n v_n$ as one term, as the role of $v_n$ in equation 3. By comparing coefficients of equation 8 and equation 3, we get $\alpha^{(n)} = (\gamma_n / \gamma_{n-1}) \beta_n$ and $\varepsilon_n = \gamma_n (1 - \beta_n)$. In the literature (Gupal & Bazhenov, 1972; Polyak, 1977; Kaniovskii, 1983; Gitman et al., 2019), the parameter setting of SHB in equation 4 has the following requirements
$$\sum_{n=1}^{+\infty} \gamma_n = +\infty, \quad \sum_{n=1}^{+\infty} \gamma_n^2 < +\infty, \quad \beta_n < 1, \quad \lim_{n \to +\infty} \beta_n = 0$$
*or*
$$\sum_{n=1}^{+\infty} \gamma_n = +\infty, \quad \sum_{n=1}^{+\infty} (1 - \beta_n)^2 < +\infty, \quad \sum_{n=1}^{+\infty} \frac{\gamma_n^2}{1 - \beta_n} < +\infty, \quad \lim_{n \to +\infty} \beta_n = 1.$$

According to the above requirements on $\beta_n$ and $\gamma_n$, the requirements on $\varepsilon_n$ and $\alpha^{(n)}$ are

$$\sum_{n=1}^{+\infty} \varepsilon_n = +\infty, \quad \sum_{n=1}^{+\infty} \varepsilon_n^2 < +\infty, \quad \lim_{n \to +\infty} \alpha^{(n)} = 0.$$

*or*

$$\sum_{n=1}^{+\infty} \varepsilon_n = \sum_{n=1}^{+\infty} \gamma_n(1-\beta_n) \le \sqrt{\sum_{n=1}^{+\infty}(1-\beta_n)^3 \sum_{n=1}^{+\infty} \frac{\gamma_n^2}{1-\beta_n}} < +\infty, \quad \limsup_{n \to +\infty} \alpha^{(n)} = 1.$$

We can see that the static momentum parameter $\alpha$, which is widely used in practical applications, does not satisfy these conditions. In Sebbouh et al. (2020), the authors studied an algorithm (SHB-IMA) that has a similar form to mSGD given in equation 3, but their conditions for parameters cannot cover the case with a static $\alpha$.

Before providing the main theorem for convergence, we need a useful lemma In the analysis of asymptotic convergence of mSGD, the following lemma plays an important role.

**Lemma 1** *Consider the mSGD in equation 3. If Assumptions 1–3 hold, then for $\forall \theta_1 \in \mathbb{R}^N$ and $v_0 \in \mathbb{R}^N$, there is a scalar $T(\theta_1, v_0)$, such that $\mathbb{E}\big(g(\theta_n)\big) < T(\theta_1, v_0)$ for any $n \ge 1$.*

Lemma 1 actually guarantees stability of mSGD. Let $\mathscr{F}_n = \sigma(\theta_1, v_0, \{\xi_i\}_{i=1}^n)$ be the minimal $\sigma$-algebra generated by $\theta_1, v_0, \{\xi_i\}_{i=1}^n$. As a result, $\theta_n$ is adapted to $\mathscr{F}_{n-1}$. Then for any $n \in \mathbb{N}_+$, it holds that

$$\mathbb{E}\left(\left\|\nabla_{\theta_n} g(\theta_n) - \nabla_{\theta_n} g(\theta_n, \xi_n)\right\|^2\right) = \mathbb{E}\left(\mathbb{E}\left(\left\|\nabla_{\theta_n} g(\theta_n) - \nabla_{\theta_n} g(\theta_n, \xi_n)\right\|^2 \middle| \mathscr{F}_{n-1}\right)\right)$$

$$= \mathbb{E}\left(\mathbb{E}_{\xi_n}\left(\left\|\nabla_{\theta_n} g(\theta_n) - \nabla_{\theta_n} g(\theta_n, \xi_n)\right\|^2\right)\right) \le M\left(1 + \mathbb{E}\big(g(\theta_n)\big)\right) \le M(1 + T(\theta_1, v_0)), \quad (9)$$

where the second equality follows from the independence between $\xi_n$ and $\{\theta_1, v_0, \{\xi_i\}_{i=1}^{n-1}\}$ in Assumption 1, and the last two inequalities hold due to 4) in Assumption 2 and Lemma 1, respectively. Intuitively, the result in equation 9 means that the fluctuation induced by random noise $\{\xi_i\}_{i=1}^{n-1}$ is well restrained. Note that the derived result in equation 9 is totally different from equation 7 required in the literature (Shalev-Shwartz et al., 2011; Nemirovski et al., 2009; Hazan & Kale, 2014; Gitman et al., 2019; Yang et al., 2016; Polyak, 1977; Kaniovskii, 1983), since equation 7 needs a uniformly upper bound $K$ over the whole space (i.e., $\theta \in \mathbb{R}^N$) which is difficult to satisfy when loss function $g(\theta)$ is quadratic or cubic with respect to $\theta$ over unbounded parameter space. In contrast, regardless of the order of $g(\theta)$ with respect to $\theta$, equation 9 ensures boundedness of $\mathbb{E}\left(\left\|\nabla_{\theta_n} g(\theta_n) - \nabla_{\theta_n} g(\theta_n, \xi_n)\right\|^2\right)$ for learning any fixed true parameter $\theta^*$. This reflects a favorable learning process against random noise $\xi_n$ for dealing with general loss functions $g(\theta)$. This result paves the way to the following theorem on asymptotic convergence of mSGD.

**Theorem 1** *Consider the mSGD in equation 3. If Assumptions 1–3 hold, then for $\forall \theta_1 \in \mathbb{R}^N$ and $\forall v_0 \in \mathbb{R}^N$, there exists a connected set $J^* \subseteq J$ such that the iterate $\theta_n$ is convergent to the set $J^*$ almost surely, i.e.,*

$$\lim_{n \to \infty} d(\theta_n, J^*) = 0, \qquad a.s.$$

*where $d(x, J^*) = \inf_y\{\|x - y\|, y \in J^*\}$ denotes the distance between point $x$ and set $J^*$.*

In Theorem 1, we prove that the iterates of mSGD asymptotically converge to a connected set of stationary points almost surely. When this connected set degenerates to a stationary point $\theta^*$, it holds that $\theta_n \to \theta^*$, a.s.. The result enables engineers for the design of proper momentum coefficients and step sizes (like $\alpha = 0.9$, $\varepsilon_n = 0.1/n$) in the related applications of mSGD with mathematic guarantee.

### 3.2 CONVERGENCE RATE OF MSGD

In this subsection, we analyze the convergence rate of mSGD. Before that, given positive real sequences $\{a_n\}$ and $\{b_n\}$, we let $a_n = O(b_n)$ if there is a constant $c > 0$, such that $a_n/b_n < c$ for any $n \ge 1$. For quantitative analysis, we need some extra assumptions.

**Assumption 4** (*Loss function assumption*) *Loss function $g(\theta)$ satisfies the following conditions:*

*1)* *$g(\theta)$ is a non-negative and continuously differentiable function. The set of its stationary points $J = \{\theta | \|\nabla_\theta g(\theta)\| = 0\}$ is a bounded set which has only finite connected components $J_1, ..., J_n$. In addition, there is $\varepsilon' > 0$, such that for any $i \in \{1, 2, ..., n\}$ and $0 < d(\theta, J_i) < \varepsilon'$, it holds that $|g(\theta) - g_i| \neq 0$, where $g_i = \{g(\theta) | \theta \in J_i\}$ is a constant.*

*2)* *For any $i \in \{1, 2, ..., n\}$, it holds that*

$$\liminf_{d(\theta, J_i) \to 0} \frac{\|\nabla_\theta g(\theta)\|^2}{g(\theta) - g_i} \geq s > 0.$$

In many problems of machine learning, especially deep learning, because of strong non-linearity mapping from input data to output data and the structure complexity of employed models, loss functions could be non-convex and may have multiple local critical points. Assumption 4 does not require convexity of the loss function, but guarantees the properties of the loss function around critical points. Assumption 4 2) can be treated as a local version of Polyak-Lojasiewicz (P-L) condition.

**Theorem 2** *Consider the mSGD in equation 3 with the noise following a uniform sampling distribution. If Assumptions 1-4 hold, for $\forall v_0 \in \mathbb{R}^N$ and $\forall \theta_1 \in \mathbb{R}^N$, it holds that*

$$\mathbb{E}\left(\|\nabla_{\theta_n} g(\theta_n)\|^2\right) = O\left(e^{-\sum_{i=1}^n \frac{s\varepsilon_i}{p(1-\alpha)^2}}\right), \tag{10}$$

*where $p = \exp\left\{\sum_{k=1}^\infty M\varepsilon_k^2\right\}$ and M is defined in condition 4) of Assumption 2.*

In Theorem 2, let $\alpha = 0$, then we can obtain the convergence rate of SGD,

$$\mathbb{E}\left(\|\nabla_{\theta_n} g(\theta_n)\|^2\right) = O\left(e^{-\sum_{i=1}^n \frac{s}{p}\varepsilon_i}\right).$$

According to the obtained bounds, the convergence rate of mSGD with $\alpha \in (0, 1)$ is larger than that of SGD, which is the case with $\alpha = 0$. Moreover, Theorem 2 provides a stronger characterization of convergence rate than some existing works considering the convergence rate of time averages $\frac{1}{T} \sum_{i=1}^T \mathbb{E}(\|\nabla_{\theta_n} g(\theta_n)\|^2) = O(T^l)$ ($l = -1$ in Liu et al. (2020) and $l = -1/2$ in Mai & Johansson (2020)). In the following, we will elaborate on this point. From equation 10, there exists a scalar $t_0 > 0$ such that $\forall n > 0$

$$\frac{\mathbb{E}\left(\|\nabla_{\theta_n} g(\theta_n)\|^2\right)}{\exp\left\{-\sum_{i=1}^n \frac{s\varepsilon_i}{p(1-\alpha)^2}\right\}} < t_0,$$

implying

$$\frac{\frac{1}{T} \sum_{n=1}^T \mathbb{E}\left(\|\nabla_{\theta_n} g(\theta_n)\|^2\right)}{\frac{1}{T} \sum_{n=1}^T \exp\left\{-\sum_{i=1}^n \frac{s\varepsilon_i}{p(1-\alpha)^2}\right\}} < t_0.$$

So it holds that

$$\frac{1}{T} \sum_{n=1}^T \mathbb{E}\left(\|\nabla_{\theta_n} g(\theta_n)\|^2\right) = O\left(\frac{1}{T} \sum_{n=1}^T e^{-\sum_{i=1}^n \frac{s\varepsilon_i}{p(1-\alpha)^2}}\right).$$

Let $\varepsilon_n = \frac{1}{n}$, then we know that $\sum_{n=1}^T \varepsilon_n = O(\ln T)$. Hence,

$$\frac{1}{T} \sum_{n=1}^T e^{-\sum_{i=1}^n \frac{s\varepsilon_i}{p(1-\alpha)^2}} = O\left(\frac{1}{T} \sum_{n=1}^T e^{-\frac{s\ln(n)}{p(1-\alpha)^2}}\right) = O\left(\frac{1}{T} \sum_{n=1}^T n^{-\frac{s}{p(1-\alpha)^2}}\right).$$

If $\frac{s}{p(1-\alpha)^2} < 1$, $\sum_{n=1}^T n^{-\frac{s}{p(1-\alpha)^2}} = O\left(T^{-\frac{s}{p(1-\alpha)^2}+1}\right)$, so we have

$$\frac{1}{T} \sum_{n=1}^T e^{-\sum_{i=1}^n \frac{s\varepsilon_i}{p(1-\alpha)^2}} = \frac{1}{T} O\left(T^{-\frac{s}{p(1-\alpha)^2}+1}\right) = O\left(T^{-\frac{s}{p(1-\alpha)^2}}\right).$$

If $\frac{s}{p(1-\alpha)^2} = 1$, it holds that $\sum_{n=1}^{T} n^{-\frac{s}{p(1-\alpha)^2}} = O(\ln T)$, and hence

$$\frac{1}{T}\sum_{n=1}^{T} e^{-\Sigma_{i=1}^{n}\frac{s\varepsilon_i}{p(1-\alpha)^2}} = \frac{1}{T}O(\ln T) = O\left(\frac{\ln T}{T}\right).$$

If $\frac{s}{p(1-\alpha)^2} > 1$, then $\sum_{n=1}^{T} n^{-\frac{s}{p(1-\alpha)^2}} := C_\alpha(T) \to C_\alpha$ as $T \to +\infty$, where $C_\alpha$ is a positive constant, so

$$\frac{1}{T}\sum_{n=1}^{T} e^{-\Sigma_{i=1}^{n}\frac{s\varepsilon_i}{p(1-\alpha)^2}} = \frac{C_\alpha(T)}{T} = O\left(\frac{C_\alpha}{T}\right).$$

Note that the coefficient $C_\alpha$ depends on $s$, $p$, and $\alpha$, where $s$ is the constant given in Assumption 4 only depending on $g(\theta)$, and $p$ is given in Theorem 2 and relies on $M$ in 4) of Assumption 2 and step size $\varepsilon_n$. The above observation indicates that setting $\alpha$ close to one makes mSGD achieve convergence rate of time averages of order $O(1/T)$. As remarked previously, larger $\alpha$ implies smaller coefficient $C_\alpha$ and thus quicker convergence rate. Interestingly, since $C_\alpha(T)$ is monotonically increasing, the convergence rate has a relatively small coefficient when $T$ is small. This could illustrate why mSGD can achieve better performance in the early phase of iteration.

### 3.3 CONVERGENCE OF ADAGRAD

In this subsection, we aim to establish the convergence of AdaGrad. Compared to the study of mSGD, the design of adaptive step size increases the technical difficulties. To proceed, the required conditions on loss function $g(\theta)$ are summarized as follows.

**Assumption 5** *Loss function $g(\theta)$ in equation 5 satisfies the following conditions:*

*1)* $g(\theta)$ *is a non-negative and continuously differentiable function. The set of its stationary points $J = \{\theta \| \|\nabla_\theta g(\theta)\| = 0\}$ is a bounded set which has only finite connected components $J_1, ..., J_n$. In addition, there is $\varepsilon_1 > 0$, such that for any $i$ and $0 < d(\theta, J_i) < \varepsilon_1$, it holds that $|g(\theta) - g_i| \neq 0$, where $g_i = \{g(\theta)|\theta \in J_i\}$ is a constant.*

*2) The gradient $\nabla_\theta g(\theta)$ satisfies the Lipschitz condition, i.e., for any $x, y \in \mathbb{R}^N$,*

$$\left\|\nabla_x g(x) - \nabla_y g(y)\right\| \leq c\|x - y\|.$$

*3) There are two constants $M' > 0$ and $a > 0$ such that for any $\theta \in \mathbb{R}^N$ and $n \in \mathbb{N}_+$,*

$$E_{\xi_n}\left(\left\|\nabla_\theta g(\theta, \xi_n)\right\|^2\right) \leq M'\left\|\nabla_\theta g(\theta)\right\|^2 + a. \tag{11}$$

Condition 2) is the same as in 2) of Assumption 2. Condition 1) is relatively weak, because it does not require any convexity of the loss function or global conditions as P-L condition. There are many functions satisfying Assumption 5 1) but not convex, such as $y = \sin^2(x)$, $y = (x-1)(x-2)(x-3)(x-4)$, and $y = \cos^2(x)$. Similar to condition 4) of Assumption 2, condition 3) is a condition to restrain the noise influence. Equation 11 is milder than $\|\nabla_\theta g(\theta, \xi_n) - \nabla_\theta g(\theta)\| \leq S$ *a.s.* (Li & Orabona, 2019; Défossez et al., 2020), which is relatively restrictive in dealing with unbounded noises, e.g., $\nabla_\theta g(\theta, \xi_n) = \nabla_\theta g(\theta) + \xi_n$, where $\xi_n$ is independent identically distributed and Gaussian.

Compared to SGD and mSGD, there are more challenges in analyzing the convergence of AdaGrad. The challenges mainly come from two aspects: (1) since AdaGrad does not have a decreasing step size, the noise influence on AdaGrad cannot be restrained as mSGD which is with the help of decreasing step size satisfying Assumption 3; (2) adaptive step size of AdaGrad in equation 5 (i.e., $\alpha_0/\sqrt{\sum_{i=1}^{n}\|\nabla_{\theta_i}g(\theta_i, \xi_i)\|^2}$) is a random variable conditionally dependent of $\xi_n$ given $\sigma\{\theta_1, \xi_1, ..., \xi_{n-1}\}$. Then when we deal with terms in the proof like

$$\frac{\alpha_0}{\sqrt{S_n}}\nabla_{\theta_n}g(\theta_n)^T\nabla_{\theta_n}g(\theta_n, \xi_n) \quad \text{or} \quad \frac{\alpha_0^2}{S_n}\left\|\nabla_{\theta_n}g(\theta_n, \xi_n)\right\|^2, \tag{12}$$

we cannot make the conditional expectation to transform equation 12 to

$$\frac{\alpha_0}{\sqrt{S_n}}\left\|\nabla_{\theta_n}g(\theta_n)\right\|^2 \quad \text{or} \quad \frac{\alpha_0^2}{S_n}E_{\xi_n}\left(\left\|\nabla_{\theta_n}g(\theta_n, \xi_n)\right\|^2\right).$$

In the literature, Li & Orabona (2019) changed the step size to

$$\frac{\alpha_0}{\sqrt{\sum_{i=1}^{n-1}\|\nabla_{\theta_i}g(\theta_i,\xi_i)\|^2}},$$

and Gadat & Gavra (2020) changed the step size to $\gamma_{n+1}/\sqrt{\omega_n + \varepsilon}$ where

$$\omega_n = \omega_{n-1} + \gamma_n\big(p_n\|\nabla_{\theta_n}g(\theta_n,\xi_n)\|^2 - q_n\omega_{n-1}\big),$$

and $\gamma_n, p_n, q_n$ are tuned parameters, in order to make it conditionally independent of $\xi_n$ given $\sigma\{\theta_1, \xi_1,..., \xi_{n-1}\}$, which however is no longer the standard AdaGrad as in equation 5 (Streeter & Mcmahan, 2010; Chen et al., 2019).

Before we provide the main theorem of this subsection, a useful lemma is worth discussing.

**Lemma 2** *Let $\{\alpha_k\}$, $\{\beta_k\}$ being non-negative random variable sequences, such that the following conditions hold almost surely*

- $\sum_{k=1}^{+\infty}\alpha_k = \infty$;

- $\sum_{k=1}^{+\infty}\alpha_k\beta_k < +\infty$;

*Then there exists a subsequence $\{\beta_{n_k}\}$ of $\{\beta_k\}$, such that $\beta_{n_k} \xrightarrow{k\to\infty} 0$ almost surely.*

**Proof 1** *We aim to prove $\liminf_{k\to+\infty}\beta_k = 0$ by contradiction. Suppose $\liminf_{k\to+\infty}\beta_k = u > 0$, then $\exists k_0$, such that $\forall k > k_0$, $\beta_k > u/2$. It follows that*

$$\sum_{k=k_0+1}^{+\infty}\alpha_k < \frac{2}{u}\sum_{k=k_0+1}^{+\infty}\alpha_k\beta_k < +\infty. \tag{13}$$

*Obviously, there is a contradiction between equation 13 and the condition $\sum_{k=1}^{+\infty}\alpha_k = +\infty$. Thus, $\liminf_{k\to+\infty}\beta_k = 0$. So we get that there exists a subsequence $\{\beta_{n_k}\}$ of $\{\beta_k\}$ holds $\beta_{n_k} \xrightarrow{k\to\infty} 0$ almost surely.*

This lemma, inspired by Proposition 2 in Ya. I. Alber (Alber et al., 1998), is quite useful in the convergence analysis of AdaGrad. Then we are ready to provide the convergence result of AdaGrad in the following theorem.

**Theorem 3** *Consider the AdaGrad in equation 5. If Assumptions 1 and 5 hold, then for $\forall\theta_1 \in \mathbb{R}^N$ and $S_0 = 0$, there exists a connected component of the set $J^* \subseteq J$, such that the estimate $\theta_n$ is convergent to the set $J^*$ almost surely, i.e.,*

$$\lim_{n\to\infty}d(\theta_n, J^*) = 0. \qquad a.s.$$

In Theorem 3, we prove the standard AdaGrad is convergent almost surely, in contrast to the convergence of time averages in Chen et al. (2019); Ward et al. (2019) which focus on the metric $\frac{1}{T}\sum_{i=1}^{T}\mathbb{E}(\|\nabla_{\theta_n}g(\theta_n)\|^2)$, or the subsequence convergence in Zou et al. (2019); Défossez et al. (2020), focusing on $\min_{n=0,1,2,...,T}\mathbb{E}(\|\nabla_{\theta_n}g(\theta_n)\|)$.

## 4 CONCLUSION AND FUTURE WORK

In this paper, we studied the convergence of two algorithms extensively used in machine learning applications, namely, momentum-based stochastic gradient descent (mSGD) and adaptive step stochastic gradient descent (AdaGrad). By considering general loss functions (either convex or non-convex), we first establish the almost sure asymptotic convergence of mSGD. Moreover, we find the convergence rate of the mSGD and reveal that the mSGD indeed has a faster convergence rate than the SGD. Furthermore, we prove AdaGrad is convergent almost surely under mild conditions. Subsequence convergence and convergence of time averages in the literature are substantially extended in this work to asymptotic convergence. To better understand the AdaGrad, we will study its convergence rate in the future.

## 5 CODE OF ETHICS

This is a theoretical paper focusing on the investigation for the convergence of mSGD and AdaGrad optimization algorithms. The results in this paper do not have issues in fairness, inappropriate potential applications and impact, privacy and security issues, legal compliance, research Integrity Issues, or responsible research practice (e.g., IRB, documentation, research ethics). This paper does no involve human subjects, practices to data set releases, potentially harmful insights, methodologies and applications, or pontential conflicts of interest and sponsorship.

## 6 REPRODUCIBILITY

This is a theoretical paper focusing on the investigation for the convergence of mSGD and AdaGrad optimization algorithms. The developed results are provided in the main paper, i.e., Theorems 1-3. In Appendix, the proofs of these theorems are provided together with some useful lemmas as well as their proofs. An outline of these proofs is given in Section A in Appendix.

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

## A    APPENDIX OUTLINE

Section B aims to verify the convergence results of mSGD. Several auxiliary lemmas are first provided, followed by a proof outline for the main results of mSGD. Then the proofs of these lemmas are given. Theorems 1-2 are proved in Sections B.9 and B.10, respectively. Section C aims to verify the convergence results of AdaGrad. Some auxiliary lemmas are given at the beginning, followed by a proof outline of the main results for AdaGrad in Section C.1. Then we provide the proofs of the lemmas. Theorem 3 is proved in Section C.8.

## B    CONVERGENCE AND CONVERGENCE RATE OF MSGD

The following lemmas are used for the proofs of Theorems 1-2. Hereafter, $\mathbb{N}_+$ denotes the set of all positive integers.

A function $f$ is in a (differentiability) class $C^k$ if its $l$th derivatives exist and are continuous, where $l \leq k$.

**Lemma 3** *(Lemma 1.2.3 in Nesterov (2004)) Suppose $f(x) \in C^1$ $(x \in \mathbb{R}^N)$ with gradient satisfying the following Lipschitz condition*

$$\|\nabla f(x) - \nabla f(y)\| \leq c\|x - y\|,$$

*then for any $x, y \in \mathbb{R}^N$, it holds that*

$$f(y) + \nabla f(y)^T (x - y) - \frac{c}{2}\|x - y\|^2 \leq f(x) \leq f(y) + \nabla f(y)^T (x - y) + \frac{c}{2}\|x - y\|^2.$$

**Lemma 4** *Suppose $f(x) \in C^1$ $(x \in \mathbb{R}^N)$ with gradient satisfying the following Lipschitz condition*

$$\|\nabla f(x) - \nabla f(y)\| \leq c\|x - y\|,$$

*and the set $S = \{x | \nabla f(x) = 0\}$ is bounded and only has finite connected components $\{S_1, S_2, \ldots, S_m\}$. Furthermore, assume there exists $\varepsilon'_1 > 0$, such that for any $i = 1, 2, \ldots, m$ and $x \in \{x | 0 < d(x, S_i) < \varepsilon'_1\}$, it holds that $|f(x) - f_i| \neq 0$, where $f_i = f(x)$ for $x \in S_i$. Then for any $i = 1, 2, \ldots, m$, if there is $\varepsilon'_0 > 0$ satisfying $d(x, S_i) < \varepsilon'_0$, it follows that*

$$\left\|\nabla f(x)\right\|^2 \leq 2c\left|f(x) - f_i\right|.$$

**Lemma 5** *(Wang et al., 2019) Suppose that $\{X_n\} \in \mathbb{R}^N$ is an $\mathscr{L}_2$ martingale difference sequence, and $(X_n, \mathscr{F}_n)$ is an adaptive process. Then it holds that $\sum_{k=0}^{\infty} X_k < +\infty$ a.s., if*

$$\sum_{n=1}^{\infty} \mathbb{E}(\|X_n\|^2) < +\infty, \quad or \quad \sum_{n=1}^{\infty} \mathbb{E}\left(\|X_n\|^2 \big| \mathscr{F}_{n-1}\right) < +\infty. \quad a.s.$$

**Lemma 6** *Suppose that $\{X_n\} \in \mathbb{R}^N$ is a non-negative sequence of random variables, then it holds that $\sum_{n=0}^{\infty} X_n < +\infty$ a.s., if $\sum_{n=0}^{\infty} \mathbb{E}\left(X_n\right) < +\infty$.*

**Lemma 7** *Suppose $\{v_n\}$ is a sequence generated by mSGD in equation 3. Under Assumptions 1–3, it holds that $\sum_{n=0}^{+\infty} \mathbb{E}\left(\|v_n\|^2\right) < c(v_0, \theta_1)$, and $\sum_{n=0}^{+\infty} \|v_n\|^2 < +\infty$ a.s., where $c(v_0, \theta_1)$ is a constant only related to $v_0$ and $\theta_1$.*

**Lemma 8** *Suppose $\{\theta_n\}$ is a sequence generated by mSGD in equation 3. Under Assumptions 1–3, it holds that: for $n \geq 1$,*

$$\sum_{t=1}^{n} \varepsilon_t \mathbb{E}\left(\|\nabla_{\theta_t} g(\theta_t)\|^2\right) < B(v_0, \theta_1) < +\infty, \qquad \sum_{t=1}^{n} \varepsilon_t \|\nabla_{\theta_t} g(\theta_t)\|^2 < +\infty.$$

*where $B(v_0, \theta_1) > 0$ is a constant only related to $v_0$ and $\theta_1$.*

**Lemma 9** *Suppose $\{\theta_n\}$ is a sequence generated by mSGD in equation 3 such that $\{g(\theta_n)\}$ converges a.s.. If Assumptions 1–3 hold, and*

$$g(\theta_{n+1}) \leq \zeta_n - b\sum_{t=1}^{n} \varepsilon_t \|\nabla g(\theta_t)\|^2 \ a.s., \tag{14}$$

*where $\zeta_n$ is a random variable such that $\lim_{n \to +\infty} \zeta_n = \zeta < \infty$ a.s., and $b$ is a positive constant, then there exists a connected component $J^*$ of $J := \{\theta | \nabla_\theta g(\theta) = 0\}$, such that*

$$d(\theta_n, J^*) \xrightarrow{a.s.} 0.$$

### B.1 PROOF OUTLINE OF THEOREMS 1 AND 2

The proof is in light of the Lyapunov method. We aim to prove that $g(\theta_n)$ is convergent a.s., and then to prove $\nabla_{\theta_n} g(\theta_n) \to 0$ a.s. With these two results, we are able to get $\theta_n \to J^*$ a.s. In the following, we provide the proof outline to show how to obtain the provided results of mSGD.

Step 1: We prove mSGD is a stable algorithm, i.e., $\mathbb{E}(g(\theta_n)) < K < +\infty, \forall n$, in Lemma 1. The idea is to prove that a weighted sum of the loss function value, i.e.,

$$U_n := \sum_{t=1}^{n} \left(1/(2-\alpha)\right)^{n-t} \mathbb{E}\left(1 + g(\theta_{t+1})\right)$$

is bounded through a recursion formula (a rough form)

$$U_n - U_{n-1} \leq A\alpha^n + B\underbrace{\sum_{t=1}^{n-1} \left(1/(2-\alpha)\right)^{n-t} \varepsilon_t^2 \mathbb{E}\left\|\nabla_{\theta_t} g(\theta_t, \xi_t)\right\|^2}_{I} \ (A, \ B \ are \ two \ constants).$$

Then we apply Assumption 2 4) to $I$ and then obtain

$$U_n - U_{n-1} \leq A\alpha^n + B\underbrace{\sum_{t=1}^{n-1} \left(1/(2-\alpha)\right)^{n-t} \varepsilon_t^2 \mathbb{E}\left(1 + g(\theta_t)\right)}_{R} \ (A, \ B \ are \ two \ constants).$$

Combining $U_{n-1}$ and $R$ leads to

$$F_n - F_{n-1} \leq A\alpha^n + B'\left(\frac{1}{2-\alpha}\right)^n \ (A, \ B' \ are \ two \ constants),$$

where

$$F_n := \sum_{t=1}^{n} \left(\frac{1}{2-\alpha}\right)^{n-t} Z(t+1) \mathbb{E}\left(e_{t+1}^{(n)}\left(1 + g(\theta_{t+1})\right)\right)$$

and

$$Z(t) = \prod_{k=t}^{+\infty}(1 + M_0\varepsilon_k^2) = (1 + M_0\varepsilon_t^2)\prod_{k=t+1}^{+\infty}(1 + M_0\varepsilon_k^2) = (1 + M_0\varepsilon_t^2)Z(t+1).$$

Thus, we are able to obtain $E(g(\theta_n)) < F_{n-1} \leq A\sum_{t=1}^{+\infty}\alpha^t + B\sum_{t=1}^{+\infty}\left(1/(2-\alpha)\right)^t < +\infty$.

Step 2: From Lemma 1 and the condition $\sum_{t=1}^{+\infty}\varepsilon_n^2 < +\infty$, we are able to prove that $\sum_{t=1}^{n}\|v_t\|^2$ and $\sum \varepsilon_t\|\nabla_{\theta_t} g(\theta_t)\|^2$ are convergent a.s. respectively, as stated in Lemma 7 and Lemma 8.

Step 3: We divide $g(\theta_n)$ into three terms

$$g(\theta_n) = \sum_{t=1}^{n} A(n)\|v_t\|^2 + \sum_{t=1}^{n} B_n\varepsilon_t\|\nabla_{\theta_n} g(\theta_n)\|^2 + \sum_{t=1}^{n} C_n^T\left(\nabla_{\theta_n} g(\theta_n, \xi_n) - \nabla_{\theta_n} g(\theta_n)\right).$$

From Lemma 7 and Lemma 8, we are able to prove that $\sum_{t=1}^{n} A(n)\|v_t\|^2 + \sum_{t=1}^{n} B_n\varepsilon_t\|\nabla_{\theta_n} g(\theta_n)\|^2$ is convergent a.s.. From the convergence theorem for martingale-difference sum (Lemma 5), we prove that $\sum_{t=1}^{n} C_n^T\left(\nabla_{\theta_n} g(\theta_n, \xi_n) - \nabla_{\theta_n} g(\theta_n)\right)$ is convergent a.s. Then we prove $g(\theta_n)$ is convergent a.s.

Step 4: By Lemma 9 and the convergence of $g(\theta_n)$ in Step 3, we get $\theta_n \to J^*$ a.s..

Step 5: After the proof of the convergence of mSGD, we analyze the iterates of $F_n$. Then under a new assumption $\liminf_{d(\theta, J_i) \to 0} \|\nabla_\theta g(\theta)\|^2 / \left(g(\theta) - g_i\right) \geq s > 0$, we are able to obtain the convergent rate of mSGD.

## B.2 PROOF OF LEMMA 4

First we construct a closed and bounded set $S$ which satisfies $S \supset \bigcup_{i=1}^{m} S_i$. Since $\|\nabla f(x)\|$ $(x \in S)$ is a continuous function on a closed set, $\|\nabla f(x)\|$ $(x \in S)$ is a uniformly continuous function. Then $\exists \, \varepsilon_2' > 0, \, \forall \, S_i$, if $d(x, S_i) < \varepsilon_2'$, there is $\|\nabla f(x)\| < c\varepsilon_1'/4$. We assign $\varepsilon_0 = \min\{\varepsilon_1'/4, \varepsilon_2'\}$.

Let $S_i' = \{x | d(x, S_i) < \varepsilon_1'\}$, $S_i'' = \{x | d(x, S_i) < \varepsilon_0'\}$. Since $d(x, S_i)$ is a continuous function, $\forall x_0 \in S_i''$, we can always find a straight line $l_0$ paralle to $\nabla f(x_0)$ and passing through $x_0$, defined as

$$l_0 : \; x = x_0 + \frac{\nabla f(x_0)}{\|\nabla f(x_0)\|} t \; (t \in \mathbb{R}).$$

From $S_i'' \subset S_i'$, we know $x_0 \in S_i'$. Since $S_i'$ is an open set, there exists $\alpha_0 < 0 < \beta_0$, such that $x_0 + t(\nabla f(x_0)/\|\nabla f(x_0)\|) \in S_i'$ $(\alpha_0 < t < \beta_0)$, and

$$\alpha_0 = \sup_{t<0} \left( x_0 + t \left( \frac{\nabla f(x_0)}{\|\nabla f(x_0)\|} \right) \right) \notin S_i'$$

$$\beta_0 = \inf_{t>0} \left( x_0 + t \left( \frac{\nabla f(x_0)}{\|\nabla f(x_0)\|} \right) \right) \notin S_i'.$$

Define function

$$g(t) = \left| f \left( x_0 + \frac{\nabla f(x_0)}{\|\nabla f(x_0)\|} t \right) - f_i \right| \; (t \in (\alpha_0, \beta_0)).$$

Since $x_0 + t(\nabla f(x_0)/\|\nabla f(x_0)\|) \in S_i'$ $(\alpha_0 < t < \beta_0)$, it holds that

$$f \left( x_0 + \frac{\nabla f(x_0)}{\|\nabla f(x_0)\|} t \right) - f_i \neq 0.$$

So $g(t) \in C^1$ $(t \in (\alpha_0, \beta_0))$. Then for $\forall \, t', \, t'' \in (\alpha_0, \beta_0)$, from Newton-Leibniz formula, it follows that

$$g(t') - g(t'') = \int_{t''}^{t'} g'(x) dx = \int_{t''}^{t'} (g'(x) - g'(t') + g'(t')) dx = \int_{t''}^{t'} (g'(x) - g'(t')) dx + \int_{t''}^{t'} g'(t') dx. \tag{15}$$

Next we will prove $g'(x)$ satisfies the Lipschitz condition. According to the definition of $S_i'$, $f(x_0 + t\nabla f(x_0)/\|\nabla f(x_0)\|) - f_i$ keeps the same sign over $t \in (\alpha_0, \beta_0)$. Thus, $\forall \, \tau_1, \, \tau_2 \in (\alpha_0, \beta_0)$, it holds that

$$|g'(\tau_1) - g'(\tau_2)| = \left| \left( \frac{\nabla f(x_0)}{\|\nabla f(x_0)\|} \right)^T \left( \nabla f \left( x_0 + \tau_1 \frac{\nabla f(x_0)}{\|\nabla f(x_0)\|} \right) - \nabla f \left( x_0 + \tau_2 \frac{\nabla f(x_0)}{\|\nabla f(x_0)\|} \right) \right) \right|$$

$$\leq \left\| \frac{\nabla f(x_0)}{\|\nabla f(x_0)\|} \right\| \left\| \nabla f \left( x_0 + \tau_1 \frac{\nabla f(x_0)}{\|\nabla f(x_0)\|} \right) - \nabla f \left( x_0 + \tau_2 \frac{\nabla f(x_0)}{\|\nabla f(x_0)\|} \right) \right\|$$

$$= \left\| \nabla f \left( x_0 + \tau_1 \frac{\nabla f(x_0)}{\|\nabla f(x_0)\|} \right) - \nabla f \left( x_0 + \tau_2 \frac{\nabla f(x_0)}{\|\nabla f(x_0)\|} \right) \right\|.$$

From the Lipschitz condition of $\nabla f$, we have that

$$|g'(\tau_1) - g'(\tau_2)| \leq \left\| \nabla f \left( x_0 + \tau_1 \frac{\nabla f(x_0)}{\|\nabla f(x_0)\|} \right) - \nabla f \left( x_0 + \tau_2 \frac{\nabla f(x_0)}{\|\nabla f(x_0)\|} \right) \right\| \leq c|\tau_1 - \tau_2| \left\| \frac{\nabla f(x_0)}{\|\nabla f(x_0)\|} \right\|$$

$$= c|\tau_1 - \tau_2|.$$

From the above analsis, we obtain the Lipschitz condition of $g'(x)$, that is, $\forall \, \tau_1, \, \tau_2$, there is $|g'(\tau_1) - g'(\tau_2)| \leq c|\tau_1 - \tau_2|$. By using the absolute value inequality, we get that $-c|\tau_1 - \tau_2| \leq g'(\tau_1) - g'(\tau_2) \leq c|\tau_1 - \tau_2|$. Then it follows from equation 15 that

$$g(t') - g(t'') = \int_{t''}^{t'} (g'(x) - g'(t')) dx + \int_{t''}^{t'} g'(t') dx \geq \int_{t''}^{t'} -c|x - t'| dx + \int_{t''}^{t'} g'(t') dx$$

$$= -\frac{1}{2c}(t' - t'')^2 + g'(t')(t' - t''),$$

Let $t' = 0, t'' = -g'(0)/c$. So $t' = 0 \in (\alpha_0, \beta_0)$. Next, we will prove $t'' \in (\alpha_0, \beta_0)$. We separate it into two cases. First, we assume that $g'(0) > 0$. In this case, because

$$\alpha_0 = \sup_{t<0} \left( x_0 + t \left( \frac{\nabla f(x_0)}{\|\nabla f(x_0)\|} \right) \right) \notin S_i',$$

we just need to prove

$$x_0 + t_0 \frac{f(x_0)}{\|\nabla f(x_0)\|} \in S_i', \forall t_0 \in \left( -\frac{g'(0)}{c}, 0 \right). \tag{16}$$

Note that

$$d \left( x_0 + t_0 \frac{f(x_0)}{\|\nabla f(x_0)\|}, S_i \right) \leq d(x_0, S_i) + \left\| t_0 \frac{f(x_0)}{\|\nabla f(x_0)\|} \right\| \leq \varepsilon_0' + |t_0|.$$

From $x_0 \in S_i''$, we get $\|f(x_0)\| < c\varepsilon_1'/4$ and $d(x_0, S_i) < \varepsilon_1'$. Then we have

$$d \left( x_0 + t_0 \frac{f(x_0)}{\|\nabla f(x_0)\|}, S_i \right) \leq \varepsilon_0' + |t_0| \leq \varepsilon_0' + \left| \frac{g'(0)}{c} \right| = \varepsilon_0' + \frac{\|\nabla f(x_0)\|}{c} \leq \varepsilon_0' + \frac{\varepsilon_1'}{4} \leq \frac{\varepsilon_1'}{2} < \varepsilon_1'.$$

Thus, equation 16 holds, meaning when $g'(0) > 0$, $t'' \in (\alpha_0, \beta_0)$.

Secondly, when $g'(0) < 0$, we can also prove $t'' \in (\alpha_0, \beta_0)$. It follows that

$$g(0) = g(t') \geq g(t') - g(t'') \geq -\frac{1}{2c}(t' - t'')^2 + g'(t')(t' - t'') = -\frac{1}{2c}(g'(0))^2 + \frac{1}{c}(g'(0))^2 = \frac{1}{2c}(g'(0))^2$$

$$= \frac{1}{2c} \left( \left( \frac{\nabla f(x_0)}{\|\nabla f(x_0)\|} \right)^T \nabla f(x_0) \right)^2 = \frac{1}{2c}\|\nabla f(x_0)\|^2.$$

That is

$$|f(x_0) - f_i| \geq \frac{1}{2c}\|\nabla f(x_0)\|^2.$$

Because of the arbitrariness of $x_0$, we concludes that $\exists \varepsilon_0' = \min\{\varepsilon_1'/4, \varepsilon_2'\}$, $\forall S_i$ $(i = 1, 2, ..., m)$, if $d(x, S_i) < \varepsilon_0'$, there is

$$\|\nabla f(x)\|^2 \leq 2c|f(x) - f_i|.$$

### B.3 PROOF OF LEMMA 6

For $\forall \delta > 0$, we have

$$P \left( \bigcap_{n=1}^{+\infty} \bigcup_{m=n}^{+\infty} \left( \sum_{t=n}^{m} X_t \geq \delta \right) \right)$$

$$= \lim_{n \to +\infty} P \left( \bigcup_{m=n}^{+\infty} \left( \sum_{t=n}^{m} X_t \geq \delta \right) \right) = \lim_{n \to +\infty} P \left( \sum_{t=n}^{+\infty} X_t \geq \delta \right) \leq \lim_{n \to +\infty} \frac{1}{\delta} \mathbb{E} \left( \sum_{t=n}^{+\infty} X_t \right),$$

where the last inequality is due to Markov inequality. Since $\sum_{t=1}^{\infty} \mathbb{E}(X_t) < \infty$, it follows that

$$P \left( \bigcap_{n=1}^{+\infty} \bigcup_{m=n}^{+\infty} \left( \sum_{t=n}^{m} X_t \geq \delta \right) \right) \leq \lim_{n \to +\infty} \frac{1}{\delta} \mathbb{E} \left( \sum_{t=n}^{+\infty} X_t \right) = 0. \tag{17}$$

By Cauchy's convergence test, we have $\sum_{t=1}^{\infty} X_t < \infty$, $a.s.$

### B.4 PROOF OF LEMMA 1

Recall the mSGD algorithm in equation 3

$$v_n = \alpha v_{n-1} + \varepsilon_n \nabla_{\theta_n} g(\theta_n, \xi_n) \tag{18}$$

$$\theta_{n+1} = \theta_n - v_n. \tag{19}$$

Equation equation 18 is equivalent to

$$v_n = \alpha v_{n-1} + \varepsilon_n \nabla_{\theta_n} g(\theta_n) + \varepsilon_n \left( \nabla_{\theta_n} g(\theta_n, \xi_n) - \nabla_{\theta_n} g(\theta_n) \right).$$

Under Assumption 2 2), it follows from Lemma 3 that

$$-\nabla_{\theta_t} g(\theta_t)^\mathsf{T} v_t - \frac{c}{2} \|v_t\|^2 \leq g(\theta_{t+1}) - g(\theta_t) \leq -\nabla_{\theta_t} g(\theta_t)^\mathsf{T} v_t + \frac{c}{2} \|v_t\|^2. \tag{20}$$

In this subsection, we just use the right side of equation 20. The left side will be used in the next subsection. Consider $\nabla_{\theta_t} g(\theta_t)^\mathsf{T} v_t$ in the following

$$
\begin{aligned}
\nabla_{\theta_t} g(\theta_t)^\mathsf{T} v_t &= (\nabla_{\theta_t} g(\theta_t))^\mathsf{T} (\alpha v_{t-1} + \varepsilon_t \nabla_{\theta_n} g(\theta_t, \xi_t)) \\
&= \alpha (\nabla_{\theta_{t-1}} g(\theta_{t-1}) + \nabla_{\theta_t} g(\theta_t) - \nabla_{\theta_{t-1}} g(\theta_{t-1}))^\mathsf{T} v_{t-1} + \varepsilon_t \nabla_{\theta_t} g(\theta_t)^\mathsf{T} \nabla_{\theta_t} g(\theta_t, \xi_t) \\
&= \alpha \nabla_{\theta_{t-1}} g(\theta_{t-1})^\mathsf{T} v_{t-1} + \alpha (\nabla_{\theta_t} g(\theta_t) - \nabla_{\theta_{t-1}} g(\theta_{t-1}))^T v_{t-1} + \varepsilon_t \nabla_{\theta_t} g(\theta_t)^\mathsf{T} \nabla_{\theta_t} g(\theta_t, \xi_t)
\end{aligned}
$$

Recursively applying the above equation yields

$$
\begin{aligned}
\nabla_{\theta_t} g(\theta_t)^\mathsf{T} v_t &= \alpha^{t-1} \nabla_{\theta_1} g(\theta_1)^\mathsf{T} v_1 + \sum_{i=1}^{t-1} \alpha^{t-i} (\nabla_{\theta_i} g(\theta_i) - \nabla_{\theta_{i-1}} g(\theta_{i-1}))^T v_{i-1} \\
&+ \sum_{i=2}^{t} \alpha^{t-i} \varepsilon_i \nabla_{\theta_i} g(\theta_i)^\mathsf{T} \nabla_{\theta_i} g(\theta_i, \xi_i).
\end{aligned}
\tag{21}
$$

By substituting the above equation into equation 20 and noting $-(\nabla_{\theta_i} g(\theta_i) - \nabla_{\theta_{i-1}} g(\theta_{i-1}))^\mathsf{T} v_{i-1} \leq \|\nabla_{\theta_i} g(\theta_i) - \nabla_{\theta_{i-1}} g(\theta_{i-1})\| \|v_{i-1}\| \leq c \|v_{t-1}\|^2$, we obtain

$$
\begin{aligned}
&g(\theta_{t+1}) - g(\theta_t) \\
&\leq -\alpha^{t-1} \nabla_{\theta_1} g(\theta_1)^\mathsf{T} v_1 - \sum_{i=2}^{t} \alpha^{t-i} \varepsilon_i \nabla_{\theta_i} g(\theta_i)^\mathsf{T} \nabla_{\theta_i} g(\theta_i, \xi_i) + \frac{c}{2} \|v_t\|^2 \\
&\quad - \sum_{i=1}^{t-1} \alpha^{t-i} (\nabla_{\theta_t} g(\theta_t) - \nabla_{\theta_{t-1}} g(\theta_{t-1}))^T v_{t-1} \\
&< -\alpha^{t-1} \nabla_{\theta_1} g(\theta_1)^\mathsf{T} v_1 - \sum_{i=2}^{t} \alpha^{t-i} \varepsilon_i \nabla_{\theta_i} g(\theta_i)^\mathsf{T} \nabla_{\theta_i} g(\theta_i, \xi_i) + c \sum_{i=1}^{t} \alpha^{t-i} \|v_i\|^2.
\end{aligned}
\tag{22}
$$

Denote $Z(t) := \prod_{k=t}^{+\infty} (1 + M_0 \varepsilon_k^2)$, where

$$M_0 = \frac{cM}{\alpha^{1-\delta} (1 - \alpha^\delta)(1 - \alpha)^2},$$

where $\delta > 0$ is a constant and $M$ is introduced in Assumption 2 4). Here we define $M_0$, $Z(t)$ and $\delta$ to facilitate the proof equation 33. From Assumption 3, it holds that $\sum_{t=1}^{+\infty} \varepsilon_t^2 < +\infty$. Thus, $\sum_{t=1}^{+\infty} M_0 \varepsilon_t^2 < +\infty$. From a general inequality $\ln(1+x) \leq x$ for $x > -1$, we get

$$Z(t) \leq \prod_{k=1}^{+\infty} (1 + M_0 \varepsilon_k^2) = \exp \left\{ \sum_{k=1}^{\infty} \ln(1 + M_0 \varepsilon_k^2) \right\} \leq \exp \left\{ \sum_{k=1}^{\infty} M_0 \varepsilon_k^2 \right\} < +\infty,$$

which means that $Z(t)$ is uniformly upper bounded. Then multiplying $Z(t+1)$ on both sides of equation 22 and taking the mathematical expectation yield

$$
\begin{aligned}
Z(t+1) \mathbb{E} \left( g(\theta_{t+1}) - g(\theta_t) \right) &\leq -Z(t+1) \alpha^{t-1} \mathbb{E} \left( \nabla_{\theta_1} g(\theta_1)^\mathsf{T} v_1 \right) + c \sum_{i=1}^{t} \alpha^{t-i} Z(i+1) \mathbb{E} \left( \|v_i\|^2 \right) \\
&\quad - Z(t+1) \sum_{i=2}^{t} \alpha^{t-i} \varepsilon_i \mathbb{E} \left\| \nabla_{\theta_i} g(\theta_i) \right\|^2 \\
&< -Z(t+1) \alpha^{t-1} \mathbb{E} \left( \nabla_{\theta_1} g(\theta_1)^\mathsf{T} v_1 \right) + c \sum_{i=1}^{t} \alpha^{t-i} Z(i+1) \mathbb{E} \left( \|v_i\|^2 \right),
\end{aligned}
\tag{23}
$$

where the first and second inequalities are respectively due to $Z(i+1) < Z(i)$ and $Z(t+1) \sum_{i=2}^{t} \alpha^{t-i} \varepsilon_i \mathbb{E} \left\| \nabla_{\theta_i} g(\theta_i) \right\|^2 > 0$.

Next, we aim to analyze $c\sum_{i=1}^{t}\alpha^{t-i}Z(i+1)\mathbb{E}\left(\|v_i\|^2\right)$ in equation 23. It is proved in Appendix B.5 that

$$
Z(i+1)\mathbb{E}\|v_i\|^2 \leq \alpha^{(1+\delta)i}Z(1)\mathbb{E}\|v_0\|^2 + \frac{1}{\alpha^{1-\delta}}\sum_{k=1}^{i}\alpha^{(1+\delta)(i-k)}\varepsilon_k^2 Z(k+1)\mathbb{E}\left(\|\nabla_{\theta_k}g(\theta_k)-\nabla_{\theta_k}g(\theta_k,\xi_k)\|^2\right)
$$
$$
- \frac{2}{\alpha^{1-\delta}}\sum_{k=1}^{i}\alpha^{(1+\delta)(i-k)}Z(k+1)\left(\mathbb{E}\left(\varepsilon_k g(\theta_{k+1})\right)-\mathbb{E}\left(\varepsilon_{k-1}g(\theta_k)\right)\right).
$$
(24)

Taking a weighted sum of equation 24 yields

$$
\sum_{i=1}^{t}\alpha^{t-i}Z(i+1)\mathbb{E}\|v_i\|^2
$$
$$
\leq \sum_{i=1}^{t}\alpha^{t-i}\alpha^{(1+\delta)i}Z(1)\mathbb{E}\left(\|v_0\|^2\right)
$$
$$
+ \sum_{i=1}^{t}\alpha^{t-i}\left(\frac{1}{\alpha^{1-\delta}}\sum_{k=1}^{i}\alpha^{(1+\delta)(i-k)}\varepsilon_k^2 Z(k+1)\mathbb{E}\left(\|\nabla_{\theta_k}g(\theta_k)-\nabla_{\theta_k}g(\theta_k,\xi_k)\|^2\right)\right)
$$
$$
- \sum_{i=1}^{t}\alpha^{t-i}\left(\frac{2}{\alpha^{1-\delta}}\sum_{k=1}^{i}\alpha^{(1+\delta)(i-k)}Z(k+1)\left(\mathbb{E}\left(\varepsilon_k g(\theta_{k+1})\right)-\mathbb{E}\left(\varepsilon_{k-1}g(\theta_k)\right)\right)\right) := A+B+C.
$$
(25)

We derive that

$$
A = \left(\sum_{i=1}^{t}\alpha^{\delta i}\right)\alpha^t Z(1)\mathbb{E}\left(\|v_0\|^2\right) \leq \frac{\alpha^{t-1}\alpha^{\delta}}{1-\alpha^{\delta}}Z(1)\mathbb{E}\left(\|v_0\|^2\right),
$$
(26)

$$
B = \frac{1}{\alpha^{1-\delta}}\sum_{i=1}^{t}\sum_{k=1}^{i}\alpha^{t-k+\delta(i-k)}\left(\varepsilon_k^2 Z(k+1)\mathbb{E}\left(\|\nabla_{\theta_k}g(\theta_k)-\nabla_{\theta_k}g(\theta_k,\xi_k)\|^2\right)\right)
$$
$$
\leq \frac{1}{\alpha^{1-\delta}(1-\alpha^{\delta})}\sum_{k=1}^{t}\alpha^{t-k}Z(k+1)\varepsilon_k^2\mathbb{E}\left(\|\nabla_{\theta_k}g(\theta_k)-\nabla_{\theta_k}g(\theta_k,\xi_k)\|^2\right),
$$
(27)

$$
C = -\frac{2}{\alpha^{1-\delta}}\sum_{k=1}^{t}\sum_{i=k}^{t}\alpha^{t-k+\delta(i-k)}Z(k+1)\left(\mathbb{E}\left(\varepsilon_k g(\theta_{k+1})\right)-\mathbb{E}\left(\varepsilon_{k-1}g(\theta_k)\right)\right)
$$
$$
= -\frac{2}{\alpha^{1-\delta}}\sum_{k=1}^{t}\left(\sum_{i=0}^{t-k}\alpha^{\delta i}\right)\alpha^{t-k}Z(k+1)\left(\mathbb{E}\left(\varepsilon_k g(\theta_{k+1})\right)-\mathbb{E}\left(\varepsilon_{k-1}g(\theta_k)\right)\right).
$$
(28)

Substituting equation 26–equation 28 into equation 23 yields

$$
Z(t+1)\mathbb{E}\left(\left(g(\theta_{t+1})-g(\theta_t)\right)\right)
$$
$$
\leq -Z(t+1)\alpha^{t-1}\mathbb{E}\left(\nabla_{\theta_1}g(\theta_1)^{\mathsf{T}}v_1\right) + \frac{\alpha^{t-1}c\alpha^{\delta}}{1-\alpha^{\delta}}Z(1)\mathbb{E}\left(\|v_0\|^2\right)
$$
$$
+ \frac{c}{\alpha^{1-\delta}(1-\alpha^{\delta})}\sum_{i=1}^{t}\alpha^{t-i}Z(i+1)\varepsilon_i^2\mathbb{E}\left(\|\nabla_{\theta_i}g(\theta_i)-\nabla_{\theta_i}g(\theta_i,\xi_i)\|^2\right)
$$
$$
- \frac{2c}{\alpha^{1-\delta}}\sum_{i=1}^{t}\left(\sum_{k=0}^{t-i}\alpha^{\delta k}\right)\alpha^{t-i}Z(i+1)\left(\mathbb{E}\left(\varepsilon_i g(\theta_{i+1})\right)-\mathbb{E}\left(\varepsilon_{i-1}g(\theta_i)\right)\right).
$$
(29)

Construct a sequence $\{V_n\}$ as follows

$$
V_n = \sum_{t=1}^{n}\left(\frac{1}{2-\alpha}\right)^{n-t}Z(t+1)\mathbb{E}\left(\left(g(\theta_{t+1})-g(\theta_t)\right)\right).
$$
(30)

By substituting equation 29 into equation 30 following the way of equation 26–equation 28, we have

$$
V_n \leq \frac{\alpha^n(2-\alpha)}{1-\alpha} Z(1) \left| \mathbb{E}\left(\nabla_{\theta_1} g(\theta_1)^\top v_1\right)\right| + \frac{c\alpha^n \alpha^\delta(2-\alpha)}{(1-\alpha)(1-\alpha^\delta)} Z(1) \mathbb{E}\left(\|v_0\|^2\right)
$$
$$
+ \frac{c}{\alpha^{1-\delta}(1-\alpha^\delta)(1-\alpha)^2} \sum_{t=1}^n \left(\frac{1}{2-\alpha}\right)^{n-i} Z(t+1)\varepsilon_t^2 \mathbb{E}\left(\|\nabla_{\theta_t}g(\theta_t) - \nabla_{\theta_t}g(\theta_t,\xi_t)\|^2\right)
$$
$$
- \frac{2c}{\alpha^{1-\delta}(1-\alpha^\delta)} \sum_{t=1}^n f(n-t)\left(\frac{1}{2-\alpha}\right)^{n-t} Z(t+1)\left(\mathbb{E}\left(\varepsilon_t g(\theta_{t+1})\right) - \mathbb{E}\left(\varepsilon_{t-1}g(\theta_t)\right)\right),
$$

where $f(n-t)$ is defined as follows

$$
f(n-t) = \sum_{k=1}^{n-t} \left(\alpha(2-\alpha)\right)^k - \alpha^\delta \sum_{k=1}^{n-t} \left(\alpha^{1+\delta}(2-\alpha)\right)^k.
$$

Move the last term to the left-hand side of the above inequality, then we have

$$
\sum_{t=1}^n \left(\frac{1}{2-\alpha}\right)^{n-t} Z(t+1) \mathbb{E}\left(e_{t+1}^{(n)} g(\theta_{t+1}) - e_t^{(n-1)} g(\theta_t)\right)
$$
$$
\leq \frac{\alpha^n(2-\alpha)}{1-\alpha} Z(1)\left|\mathbb{E}\left(\nabla_{\theta_1}g(\theta_1)^\top v_1\right)\right| + \frac{c\alpha^n\alpha^\delta(2-\alpha)}{(1-\alpha)(1-\alpha^\delta)} Z(1)\mathbb{E}\left(\|v_0\|^2\right) \tag{31}
$$
$$
+ \frac{c}{\alpha^{1-\delta}(1-\alpha^\delta)(1-\alpha)^2} \sum_{t=1}^n \left(\frac{1}{2-\alpha}\right)^{n-t} Z(t+1)\varepsilon_t^2 \mathbb{E}\left(\|\nabla_{\theta_t}g(\theta_t) - \nabla_{\theta_t}g(\theta_t,\xi_t)\|^2\right),
$$

where

$$
e_{t+1}^{(n)} = \left(1 + \frac{2c\varepsilon_t f(n-t)}{\alpha^{1-\delta}(1-\alpha^\delta)}\right). \tag{32}
$$

Because of $\alpha < 1$, it holds that $f(n-t) > 0$ and $e_{t+1}^{(n)} > 1$. It follows from Assumption 3 that

$$
\frac{c}{\alpha^{1-\delta}(1-\alpha^\delta)(1-\alpha)^2} \sum_{t=1}^n \left(\frac{1}{2-\alpha}\right)^{n-t} Z(t+1)\varepsilon_t^2 \mathbb{E}\left(\|\nabla_{\theta_t}g(\theta_t) - \nabla_{\theta_t}g(\theta_t,\xi_t)\|^2\right)
$$
$$
\leq \sum_{t=1}^n \left(\frac{1}{2-\alpha}\right)^{n-t} Z(t+1) M_0 \varepsilon_t^2 e_t^{(n-1)}\left(1 + \mathbb{E}(g(\theta_t))\right), \tag{33}
$$

where

$$
M_0 = \frac{cM}{\alpha^{1-\delta}(1-\alpha^\delta)(1-\alpha)^2}.
$$

Calculate $f(n-t)$, then we obtain

$$
f(n-t) = \sum_{k=1}^{n-t} \left(\alpha(2-\alpha)\right)^k - \alpha^\delta \sum_{k=1}^{n-t} \left(\alpha^{1+\delta}(2-\alpha)\right)^k > 0.
$$

It holds that

$$\sum_{t=1}^{n}\left(\frac{1}{2-\alpha}\right)^{n-t}Z(t+1)\mathbb{E}\left(e_{t+1}^{(n)}g(\theta_{t+1})-e_t^{(n-1)}g(\theta_t))\right)$$

$$-\sum_{t=1}^{n}\left(\frac{1}{2-\alpha}\right)^{n-t}Z(t+1)M_0\varepsilon_t^2 e_t^{(n-1)}\left(1+\mathbb{E}(g(\theta_t))\right)$$

$$=\sum_{t=1}^{n}\left(\frac{1}{2-\alpha}\right)^{n-t}Z(t+1)\mathbb{E}\left(e_{t+1}^{(n)}\left(1+g(\theta_{t+1})\right)\right)$$

$$-\sum_{t=1}^{n}\left(\frac{1}{2-\alpha}\right)^{n-t}Z(t+1)(1+M_0\varepsilon_t^2)\mathbb{E}\left(e_t^{(n-1)}\left(1+g(\theta_t)\right)\right) \tag{34}$$

$$-\frac{c}{\alpha^{1-\delta}(1-\alpha^\delta)}\sum_{t=1}^{n}f(n-t)\left(\frac{1}{2-\alpha}\right)^{n-t}(\varepsilon_t-\varepsilon_{t-1})$$

$$>\sum_{t=1}^{n}\left(\frac{1}{2-\alpha}\right)^{n-t}Z(t+1)\mathbb{E}\left(e_{t+1}^{(n)}\left(1+g(\theta_{t+1})\right)\right)$$

$$-\sum_{t=1}^{n}\left(\frac{1}{2-\alpha}\right)^{n-t}Z(t+1)(1+M_0\varepsilon_t^2)\mathbb{E}\left(e_t^{(n-1)}\left(1+g(\theta_t)\right)\right).$$

Substituting equation 33 and equation 34 into equation 31 yields

$$\sum_{t=1}^{n}\left(\frac{1}{2-\alpha}\right)^{n-t}Z(t+1)\mathbb{E}\left(e_{t+1}^{(n)}\left(1+g(\theta_{t+1})\right)\right)$$

$$-\sum_{t=1}^{n}\left(\frac{1}{2-\alpha}\right)^{n-t}Z(t+1)(1+M_0\varepsilon_t^2)\mathbb{E}\left(e_t^{(n-1)}\left(1+g(\theta_t)\right)\right) \tag{35}$$

$$\leq\frac{\alpha^n(2-\alpha)}{1-\alpha}Z(1)\left|\mathbb{E}\left(\nabla_{\theta_1}g(\theta_1)^\mathsf{T}v_1\right)\right|+\frac{c\alpha^n\alpha^\delta(2-\alpha)}{2(1-\alpha)(1-\alpha^\delta)}Z(1)\mathbb{E}\left(\|v_0\|^2\right).$$

According to the definition of $Z(t)$, we have

$$Z(t)=\prod_{k=t}^{+\infty}(1+M_0\varepsilon_k^2)=(1+M_0\varepsilon_t^2)\prod_{k=t+1}^{+\infty}(1+M_0\varepsilon_k^2)=(1+M_0\varepsilon_t^2)Z(t+1).$$

Let

$$F_n:=\sum_{t=1}^{n}\left(\frac{1}{2-\alpha}\right)^{n-t}Z(t+1)\mathbb{E}\left(e_{t+1}^{(n)}\left(1+g(\theta_{t+1})\right)\right),$$

and it follows from equation 35 that

$$F_n-F_{n-1}\leq Z(1)\left(\frac{1}{2-\alpha}\right)^{n-1}\mathbb{E}\left(g(\theta_1)\right)+\frac{\alpha^n(2-\alpha)}{1-\alpha}Z(1)\left|\mathbb{E}\left(\nabla_{\theta_1}g(\theta_1)^\mathsf{T}v_1\right)\right|$$

$$+\frac{c\alpha^n\alpha^\delta(2-\alpha)}{2(1-\alpha)(1-\alpha^\delta)}Z(1)\mathbb{E}\left(\|v_0\|^2\right). \tag{36}$$

Denote $p=\exp\left\{\sum_{k=1}^{\infty}M_0\varepsilon_k^2\right\}$. By taking the summation of equation 36, we obtain

$$F_n\leq Z(1)\mathbb{E}\left(e_1\left(1+g(\theta_1)\right)\right)\sum_{t=1}^{n}\left(\frac{1}{2-\alpha}\right)^{t-1}+\frac{p(2-\alpha)}{1-\alpha}\left|\mathbb{E}\left(\nabla_{\theta_1}g(\theta_1)^\mathsf{T}v_1\right)\right|\sum_{t=1}^{n}\alpha^t$$

$$+\frac{c\alpha^\delta(2-\alpha)}{2(1-\alpha)(1-\alpha^\delta)}Z(1)\mathbb{E}\left(\|v_0\|^2\right)\sum_{t=1}^{n}\alpha^t$$

$$<\frac{p(2-\alpha)}{1-\alpha}\mathbb{E}\left(e_1\left(1+g(\theta_1)\right)\right)+\frac{p\alpha(2-\alpha)}{(1-\alpha)^2}\left|\mathbb{E}\left(\nabla_{\theta_1}g(\theta_1)^\mathsf{T}v_1\right)\right|$$

$$+\frac{c\alpha^{1+\delta}(2-\alpha)}{2(1-\alpha)^2(1-\alpha^\delta)}Z(1)\mathbb{E}\left(\|v_0\|^2\right).$$

Define

$$T(\theta_1, v_0) = 1 + \frac{p(2-\alpha)}{1-\alpha} \mathbb{E}\left(e_1\left(1 + g(\theta_1)\right)\right) + \frac{p\alpha(2-\alpha)}{(1-\alpha)^2} \left| \mathbb{E}\left(\nabla_{\theta_1} g(\theta_1)^\mathsf{T} v_1\right)\right|$$
$$+ \frac{c\alpha^{1+\delta}(2-\alpha)}{2(1-\alpha)^2(1-\alpha^\delta)} Z(1) \mathbb{E}\left(\|v_0\|^2\right).$$

It follows from the relationship between $g(\theta_{n+1})$ and $F_n$ that

$$\mathbb{E}\left(1 + g(\theta_{n+1})\right) < p\left(\frac{1}{2-\alpha}\right)^{n-n} Z(n+1) \mathbb{E}\left(e_{n+1}^{(n)}\left(1 + g(\theta_{n+1})\right)\right)$$
$$< p\sum_{t=1}^{n} \left(\frac{1}{2-\alpha}\right)^{n-t} Z(t+1) \mathbb{E}\left(e_{t+1}^{(n)}\left(1 + g(\theta_{t+1})\right)\right)$$
$$< pF_n < 1 + T(\theta_1, v_0),$$

which leads to $\mathbb{E}\left(g(\theta_n)\right) < T(\theta_1, v_0)$.

## B.5 PROOF OF EQUATION 24

We consider

$$\varepsilon_i g(\theta_{i+1}) - \varepsilon_{i-1} g(\theta_i)$$
$$= \varepsilon_i\left(g(\theta_{i+1} - g(\theta_i)\right) + (\varepsilon_i - \varepsilon_{i-1})g(\theta_i)$$
$$\leq \varepsilon_i\left(g(\theta_{i+1}) - g(\theta_i)\right)$$
$$\leq -\varepsilon_i \nabla_{\theta_i} g(\theta_i)^\mathsf{T} v_i + \frac{c}{2}\varepsilon_i\|v_i\|^2$$
$$= \left(-\varepsilon_i \nabla_{\theta_i} g(\theta_i, \xi_i)^\mathsf{T} v_i - \left(\varepsilon_i \nabla_{\theta_i} g(\theta_i) - \varepsilon_i \nabla_{\theta_i} g(\theta_i, \xi_i)\right)^\mathsf{T} v_i + \frac{c}{2}\varepsilon_i\|v_i\|^2$$
$$= \alpha v_i^\mathsf{T} v_{i-1} - \|v_i\|^2 + \varepsilon_i^2\left\|\nabla_{\theta_i} g(\theta_i) - \nabla_{\theta_i} g(\theta_i, \xi_i)\right\|^2$$
$$- \left(\varepsilon_i \alpha v_{i-1} + \varepsilon_i^2 \nabla_{\theta_i} g(\theta_i)\right)^\mathsf{T}\left(\nabla_{\theta_i} g(\theta_i) - \nabla_{\theta_i} g(\theta_i, \xi_i)\right) + \frac{c}{2}\varepsilon_i\|v_i\|^2, \tag{37}$$

where the first inequality is due to $\varepsilon_i \leq \varepsilon_{i-1}$ in Assumption 3, and the last equality is from equation 19. Since $\xi_i$ and $\theta_i$ are independent, taking the mathematical expectation of equation 37 and noting that

$$\mathbb{E}\left(\left(\varepsilon_i \alpha v_{i-1} + \varepsilon_i^2 \nabla_{\theta_i} g(\theta_i)\right)^\mathsf{T}\left(\nabla_{\theta_i} g(\theta_i) - \nabla_{\theta_i} g(\theta_i, \xi_i)\right)\right) = 0,$$

yield

$$\mathbb{E}\left(\varepsilon_i g(\theta_{i+1})\right) - \mathbb{E}\left(\varepsilon_{i-1} g(\theta_i)\right)$$
$$\leq \alpha \mathbb{E}(v_i^\mathsf{T} v_{i-1}) - \mathbb{E}\left(\|v_i\|^2\right) + \varepsilon_i^2 \mathbb{E}\left(\left\|\nabla_{\theta_i} g(\theta_i) - \nabla_{\theta_i} g(\theta_i, \xi_i)\right\|^2\right) + \frac{c}{2}\varepsilon_i \mathbb{E}\left(\|v_i\|^2\right). \tag{38}$$

Moreover, it holds that

$$\mathbb{E}\left(\|\nabla_{\theta_i} g(\theta_i) - \nabla_{\theta_i} g(\theta_i, \xi_i)\|^2\right)$$
$$= \mathbb{E}\left(\|\nabla_{\theta_i} g(\theta_i, \xi_i)\|^2\right) - \mathbb{E}\left(\|\nabla_{\theta_i} g(\theta_i)\|^2\right)$$
$$= \frac{1}{\varepsilon_i^2} \mathbb{E}\left(\|v_i - \alpha v_{i-1}\|^2\right) - \mathbb{E}\left(\|\nabla_{\theta_i} g(\theta_i)\|^2\right) \tag{39}$$
$$= \frac{1}{\varepsilon_i^2}\left(\mathbb{E}\left(\|v_i\|^2\right) + \alpha^2 \mathbb{E}\left(\|v_{i-1}\|^2\right) - 2\alpha \mathbb{E}(v_i^\mathsf{T} v_{i-1})\right) - \mathbb{E}\left(\|\nabla_{\theta_i} g(\theta_i)\|^2\right).$$

Combining equation 38 and equation 39, we get

$$\mathbb{E}\left(\varepsilon_i g(\theta_{i+1})\right) - \mathbb{E}\left(\varepsilon_{i-1} g(\theta_i)\right)$$
$$\leq -\frac{1}{2}\left(\mathbb{E}\left(\|v_i\|^2\right) - \alpha^2 \mathbb{E}\left(\|v_{i-1}\|^2\right)\right) - \frac{\varepsilon_i^2}{2} \mathbb{E}\left(\|\nabla_{\theta_i} g(\theta_i)\|^2\right) + \frac{c}{2}\varepsilon_i \mathbb{E}\left(\|v_i\|^2\right) \tag{40}$$
$$+ \frac{\varepsilon_i^2}{2} \mathbb{E}\left(\|\nabla_{\theta_i} g(\theta_i) - \nabla_{\theta_i} g(\theta_i, \xi_i)\|^2\right).$$

Since $\varepsilon_i \to 0$, given any $\delta > 0$, there is an integer $i_0 \geq 0$, such that for $i \geq i_0$, $1 - c\varepsilon_i > \alpha^{1-\delta}(\delta > 0)$. Since $i_0$ is finite, without loss of generality, we assume $i_0 = 0$ for convenience, i.e., $1 - c\varepsilon_i > \alpha^{1-\delta}(\delta > 0)(i \geq 1)$. Thus, we have

$$
\mathbb{E}\left(\varepsilon_i g(\theta_{i+1})\right) - \mathbb{E}\left(\varepsilon_{i-1} g(\theta_i)\right)
$$
$$
\leq -\frac{\alpha^{1-\delta}}{2}\left(\mathbb{E}\left(\|v_i\|^2\right) - \alpha^{1+\delta}\mathbb{E}\left(\|v_{i-1}\|^2\right)\right) + \frac{\varepsilon_i^2}{2}\mathbb{E}\left(\|\nabla_{\theta_i}g(\theta_i) - \nabla_{\theta_i}g(\theta_i, \xi_i)\|^2\right). \tag{41}
$$

Multiplying both sides of equation 41 by $Z(i+1)$, and noticing that $Z(i) > Z(i+1)$, we have

$$
\left(Z(i+1)\mathbb{E}\left(\|v_i\|^2\right) - \alpha^{1+\delta}Z(i)\mathbb{E}\left(\|v_{i-1}\|^2\right)\right)
$$
$$
\leq -\frac{2}{\alpha^{1-\delta}}Z(i+1)\left(\mathbb{E}\left(\varepsilon_i g(\theta_{i+1})\right) - \mathbb{E}\left(\varepsilon_{i-1}g(\theta_i)\right)\right) + \frac{\varepsilon_i^2}{\alpha^{1-\delta}}Z(i+1)\mathbb{E}\left(\|\nabla_{\theta_i}g(\theta_i) - \nabla_{\theta_i}g(\theta_i, \xi_i)\|^2\right).
$$

Then equation 24 is obtained by recursively applying the above inequality.

### B.6 PROOF OF LEMMA 7

From equation 40, we have

$$
\mathbb{E}\left(\varepsilon_n g(\theta_{n+1})\right) - \mathbb{E}\left(\varepsilon_0 g(\theta_1)\right)
$$
$$
< -\frac{1}{2}\sum_{t=1}^n \mathbb{E}\left(\|v_t\|^2\right) + \frac{\alpha^2}{2}\sum_{t=1}^n \mathbb{E}\left(\|v_{t-1}\|^2\right) - \sum_{t=1}^n \frac{\varepsilon_t^2}{2}\mathbb{E}\left(\|\nabla_{\theta_t}g(\theta_t)\|^2\right) + \frac{c}{2}\sum_{t=1}^n \varepsilon_t \mathbb{E}\left(\|v_t\|^2\right)
$$
$$
+ \sum_{t=1}^n \frac{\varepsilon_t^2}{2}\mathbb{E}\left(\|\nabla_{\theta_t}g(\theta_t) - \nabla_{\theta_t}g(\theta_t, \xi_t)\|^2\right). \tag{42}
$$

It follows from Assumption 2 5) and Lemma 1 that

$$
\mathbb{E}\left(\|\nabla_{\theta_t}g(\theta_t) - \nabla_{\theta_t}g(\theta_t, \xi_t)\|^2\right) < M(1 + T(\theta_1, v_0)) < +\infty.
$$

Because of $\sum_{t=1}^n \varepsilon_t^2 < +\infty$, there is a scalar $\bar{M} > 0$ such that for $\forall n$

$$
\sum_{t=1}^n \frac{\varepsilon_t^2}{2}\mathbb{E}\left(\|\nabla_{\theta_t}g(\theta_t) - \nabla_{\theta_t}g(\theta_t, \xi_t)\|^2\right) < \bar{M} < +\infty.
$$

Then it follows from equation 42 that

$$
\frac{1}{2}\sum_{t=1}^n (1 - \alpha^2 - c\varepsilon_t)\mathbb{E}\left(\|v_t\|^2\right)
$$
$$
\leq \bar{M} + \varepsilon_0 g(\theta_1) - \varepsilon_n \mathbb{E}\left(g(\theta_{n+1})\right) + \frac{\alpha^2}{2}\mathbb{E}\left(\|v_0\|^2 - \|v_n\|^2\right)
$$
$$
- \sum_{t=1}^n \frac{\varepsilon_t^2}{2}\mathbb{E}\left(\|\nabla_{\theta_t}g(\theta_t)\|^2\right)
$$
$$
< \bar{M} + \varepsilon_0 g(\theta_1) + \alpha^2 \mathbb{E}\left(\|v_0\|^2\right)/2 < K,
$$

where $K$ is a positive scalar. Since $\varepsilon_n \to 0$ when $n$ is large enough, it holds that $\frac{1}{5}(1 - \alpha^2) < 1 - \alpha^2 - c\varepsilon_n$. Without loss of generality, assume $\frac{1}{5}(1 - \alpha^2) < 1 - \alpha^2 - c\varepsilon_n^2$ for $n \geq 0$, so $\sum_{t=1}^n \mathbb{E}\left(\|v_t\|^2\right) < \frac{10K}{1-\alpha^2} < +\infty$. By Lemma 6, we obtain $\sum_{t=1}^n \|v_t\|^2 < +\infty$.

### B.7 PROOF OF LEMMA 8

Through Taylor expansion, we derive that

$$
g(\theta_{t+1}) - g(\theta_t)
$$
$$
= \nabla_{\theta_{\zeta_t}}g(\theta_{\zeta_t})^T(\theta_{t+1} - \theta_t) = -\nabla_{\theta_t}g(\theta_t)^T v_t + \left(\nabla_{\theta_{\zeta_t}}g(\theta_{\zeta_t}) - \nabla_{\theta_t}g(\theta_t)\right)^T(\theta_{t+1} - \theta_t), \tag{43}
$$

where $\theta_{\zeta_t}$ means a point between $\theta_t$ and $\theta_{t+1}$. Substituting equation 21 into equation 43 yields

$$g(\theta_{t+1}) - g(\theta_t) = -\left(\alpha^{t-1} \nabla_{\theta_1} g(\theta_1)^\mathsf{T} v_1 + \sum_{i=1}^{t-1} \alpha^{t-i} (\nabla_{\theta_i} g(\theta_i) - \nabla_{\theta_{i-1}} g(\theta_{i-1}))^T v_{i-1} \right.$$
$$\left. + \sum_{i=2}^{t} \alpha^{t-i} \varepsilon_i \nabla_{\theta_i} g(\theta_i)^\mathsf{T} \nabla_{\theta_i} g(\theta_i, \xi_i) \right) + \left(\nabla_{\theta_{\zeta_t}} g(\theta_{\zeta_t}) - \nabla_{\theta_t} g(\theta_t)\right)^T (\theta_{t+1} - \theta_t).$$

It follows that

$$g(\theta_{n+1}) = g(\theta_1) + \sum_{t=1}^{n} \left(g(\theta_{t+1}) - g(\theta_t)\right)$$
$$= g(\theta_1) - \frac{1-\alpha^n}{1-\alpha} \nabla_{\theta_1} g(\theta_1)^\mathsf{T} v_1 + \frac{1-\alpha^n}{1-\alpha} \varepsilon_1 \nabla_{\theta_1} g(\theta_1) \nabla_{\theta_1} g(\theta_1, \xi_1)$$
$$- \sum_{t=1}^{n} \frac{1-\alpha^{n-t+1}}{1-\alpha} \varepsilon_t \nabla_{\theta_t} g(\theta_t) \nabla_{\theta_t} g(\theta_t, \xi_t) + \sum_{t=1}^{n} \left(\nabla_{\theta_{\zeta_t}} g(\theta_{\zeta_t}) - \nabla_{\theta_t} g(\theta_t)\right)^T (\theta_{t+1} - \theta_t)$$
$$- \sum_{t=1}^{n} \frac{1-\alpha^{n-t+1}}{1-\alpha} \sum_{i=1}^{t-1} \alpha^{t-i} (\nabla_{\theta_i} g(\theta_i) - \nabla_{\theta_{i-1}} g(\theta_{i-1}))^T v_{i-1}.$$
$$(44)$$

Take the mathematical expectation of equation 44, and notice Assumption 2 3), then we have

$$\mathbb{E}\left(g(\theta_{n+1})\right) \le \mathbb{E}\left(g(\theta_1)\right) + \frac{\alpha}{1-\alpha} \left|\nabla_{\theta_1} g(\theta_1)^\mathsf{T} v_1\right| + \frac{1}{1-\alpha} \varepsilon_1 \mathbb{E}\left(\|\nabla_{\theta_1} g(\theta_1)\|^2\right)$$
$$- \sum_{t=1}^{n} \varepsilon_t \mathbb{E}\left(\|\nabla_{\theta_t} g(\theta_t)\|^2\right) + c \sum_{t=1}^{n} \mathbb{E}\left(\|v_t\|^2\right) + \frac{2c}{1-\alpha} \sum_{t=1}^{n} \mathbb{E}\left(\|v_t\|^2\right).$$

From Lemma 7, it follows that for some positive constant $Q$,

$$c \sum_{t=1}^{n} \mathbb{E}\left(\|v_t\|^2\right) + \frac{2c}{1-\alpha} \sum_{t=1}^{n} \mathbb{E}\left(\|v_t\|^2\right) < Q.$$

Hence,

$$\mathbb{E}\left(g(\theta_{n+1})\right) < Q + \mathbb{E}\left(g(\theta_1)\right) + \frac{\alpha}{1-\alpha} \left|\nabla_{\theta_1} g(\theta_1)^\mathsf{T} v_1\right| + \frac{1}{1-\alpha} \varepsilon_1 \mathbb{E}\left(\|\nabla_{\theta_1} g(\theta_1)\|^2\right)$$
$$- \sum_{t=1}^{n} \varepsilon_t \mathbb{E}\left(\|\nabla_{\theta_t} g(\theta_t)\|^2\right).$$

As a result,

$$\sum_{t=1}^{n} \varepsilon_t \mathbb{E}\left(\|\nabla_{\theta_t} g(\theta_t)\|^2\right) < Q + \mathbb{E}\left(g(\theta_1)\right) + \frac{\alpha}{1-\alpha} \left|\nabla_{\theta_1} g(\theta_1)^\mathsf{T} v_1\right| + \frac{1}{1-\alpha} \varepsilon_1 \mathbb{E}\left(\|\nabla_{\theta_1} g(\theta_1)\|^2\right)$$
$$- \mathbb{E}\left(g(\theta_{n+1})\right)$$
$$< +\infty.$$

From Lemma 6, we have $\sum_{t=1}^{n} \varepsilon_t \|\nabla_{\theta_t} g(\theta_t)\|^2 < +\infty$ $a.s.$.

### B.8 Proof of Lemma 9

We divide the proof into three steps.

The first step is to prove
$$\liminf_{n \to +\infty} \|\nabla_{\theta_n} g(\theta_n)\|^2 = 0 \ a.s..$$

Suppose the above conclusion does not hold, i.e.,
$$\liminf_{n \to +\infty} \|\nabla_{\theta_n} g(\theta_n)\|^2 > s^2 > 0 \ a.s.,$$

where $s$ is a random variable depending on sample paths. Then it holds that

$$P\left(\bigcup_{n=1}^{+\infty}\bigcap_{m=n}^{+\infty}\left(\left\|\nabla_{\theta_m}g(\theta_m)\right\|^2 > \frac{1}{4}s^2\right)\right) = 1.$$

From equation 14 and $\lim_{n\to+\infty}\zeta_n = \zeta < \infty$ $a.s.$, if $\left\|\nabla_{\theta_n}g(\theta_n)\right\|^2 > \frac{1}{4}s^2$, it holds that

$$g(\theta_{n+1}) \le \zeta_n - b\sum_{i=n}^{+\infty}\varepsilon_i = -\infty.$$

It follows that

$$P\left(\lim_{n\to+\infty}g(\theta_{n+1}) = -\infty\right) \ge P\left(\bigcup_{n=1}^{+\infty}\bigcap_{m=n}^{+\infty}\left(\left\|\nabla_{\theta_m}g(\theta_m)\right\|^2 > \frac{1}{4}s^2\right)\right) = 1.$$

As a result, $\lim_{n\to+\infty}g(\theta_n) = -\infty$, meaning a contradiction. Hence,

$$\liminf_{n\to+\infty}\left\|\nabla_{\theta_n}g(\theta_n)\right\|^2 = 0 \ a.s. \tag{45}$$

Step 2 is to prove that the set $\{\theta_n\}$ has an accumulation point contained in $J$ with probability one. From equation 45, we have $\forall \varepsilon > 0$

$$P\left(\bigcup_{n=1}^{+\infty}\bigcap_{m=n}^{+\infty}\left(\left\|\nabla_{\theta_m}g(\theta_m)\right\|^2 > \varepsilon\right)\right) = 0.$$

Since $\|\nabla_\theta g(\theta)\|^2$ is continuous, $\forall \delta > 0$, it holds that

$$P\left(\bigcup_{n=1}^{+\infty}\bigcap_{m=n}^{+\infty}\left(\inf_{\theta\in J}\left\|\theta_m - \theta\right\|^2 > \delta\right)\right) = 0.$$

In addition, under the given conditions, $J$ is a closed set. It holds that

$$P\left(\bigcup_{n=1}^{+\infty}\bigcap_{m=n}^{+\infty}\left(\min_{\theta\in J}\left\|\theta_m - \theta\right\|^2 > \delta\right)\right) = 0.$$

For convenience, let $\theta_n^* := \arg\min_{\theta\in J}\|\theta_n - \theta\|^2$, then

$$P\left(\bigcap_{n=1}^{+\infty}\bigcup_{m=n}^{+\infty}\left(\left(\left\|\theta_m - \theta_m^*\right\| \le \sqrt{\delta}\right)\right)\right) = 1. \tag{46}$$

It follows that

$$P\left(\bigcup_{\{m_i\}\in\mathscr{S}_\mathbb{N}}\bigcap_{i=1}^{+\infty}\left(\left\|\theta_{m_i} - \theta_{m_i}^*\right\| \le \sqrt{\delta}\right)\right) = 1, \tag{47}$$

where $\mathscr{S}_\mathbb{N}$ is the set of all infinite subsequences of $\mathbb{N}$. Since $\{\theta_{m_i}^*\} \subset J$, $\{\theta_{m_i}^*\}$ is bounded. From the accumulative point principle, the event

$$\bigcap_{n=1}^{+\infty}\bigcup_{i,j\ge n}\left(\left\|\theta_{m_i}^* - \theta_{m_j}^*\right\| \le \sqrt{\delta}\right),$$

is a deterministic event, i.e., being true for every sample path. Thus

$$\bigcup_{\{m_{k_i}\}\in\mathscr{S}_\mathbb{N}^{(i)}}\bigcap_{i,j\ge 1}\left(\left\|\theta_{m_{k_i}} - \theta_{m_{k_j}}\right\| \le \sqrt{\delta}\right)$$

is a deterministic event as well, where $\mathscr{S}_\mathbb{N}^{(i)}$ is the set of all infinite subsequences of $\{m_i\}$. Due to the arbitrariness of $\{m_i\}$, we get that

$$\bigcap_{\{m_i\}\in\mathscr{S}_\mathbb{N}}\bigcup_{\{m_{k_i}\}\in\mathscr{S}_\mathbb{N}^{(i)}}\bigcap_{i,j\ge 1}\left(\left\|\theta_{m_{k_i}} - \theta_{m_{k_j}}\right\| \le \sqrt{\delta}\right) \tag{48}$$

is a deterministic event. Therefore,

$$P\left(\bigcup_{\{m_i\}\in\mathscr{S}_{\mathbb{N}}}\bigcap_{i,j\geq 1}\left(\|\theta^*_{m_i}-\theta^*_{m_j}\|<\sqrt{\delta}\right)\bigcap\left(\|\theta_{m_i}-\theta^*_{m_i}\|<\sqrt{\delta}\right)\bigcap\left(\|\theta_{m_j}-\theta^*_{m_j}\|<\sqrt{\delta}\right)\right)$$

(49)

$$=P\left(\bigcup_{\{m_i\}\in\mathscr{S}_{\mathbb{N}}}\bigcup_{\{m_{k_i}\}\in\mathscr{S}_{\mathbb{N}}^{(i)}}\bigcap_{i,j\geq 1}\left(\|\theta^*_{m_{k_i}}-\theta^*_{m_{k_j}}\|<\sqrt{\delta}\right)\bigcap\left(\|\theta_{m_{k_i}}-\theta^*_{m_{k_i}}\|<\sqrt{\delta}\right)\right.$$
$$\left.\bigcap\left(\|\theta_{m_{k_j}}-\theta^*_{m_{k_j}}\|<\sqrt{\delta}\right)\right)$$

$$\geq P\left(\bigcup_{\{m_i\}\in\mathscr{S}_{\mathbb{N}}}\bigcup_{\{m_{k_i}\}\in\mathscr{S}_{\mathbb{N}}^{(i)}}\bigcap_{i,j\geq 1}\left(\|\theta^*_{m_{k_i}}-\theta^*_{m_{k_j}}\|<\sqrt{\delta}\right)\bigcap\left(\bigcap_{i=1}^{+\infty}\|\theta_{m_i}-\theta^*_{m_i}\|<\sqrt{\delta}\right)\right)$$

$$=P\left(\bigcup_{\{m_i\}\in\mathscr{S}_{\mathbb{N}}}\left(\bigcap_{i=1}^{+\infty}\|\theta_{m_i}-\theta^*_{m_i}\|<\sqrt{\delta}\right)\bigcap\bigcup_{\{m_{k_i}\}\in\mathscr{S}_{\mathbb{N}}^{()}}\bigcap_{i,j\geq 1}\left(\|\theta^*_{m_{k_i}}-\theta^*_{m_{k_j}}\|<\sqrt{\delta}\right)\right)$$

$$\geq P\left(\bigcup_{\{m_i\}\in\mathscr{S}_{\mathbb{N}}}\left(\bigcap_{i=1}^{+\infty}\|\theta_{m_i}-\theta^*_{m_i}\|<\sqrt{\delta}\right)\right.$$
$$\left.\bigcap\left(\bigcap_{\{m_i\}\in\mathscr{S}_{\mathbb{N}}}\bigcup_{\{m_{k_i}\}\in\mathscr{S}_{\mathbb{N}}^{(i)}}\bigcap_{i,j\geq 1}\left(\|\theta^*_{m_{k_i}}-\theta^*_{m_{k_j}}\|<\sqrt{\delta}\right)\right)\right)$$

$$=P\left(\left(\bigcap_{\{m_i\}\in\mathscr{S}_{\mathbb{N}}}\bigcup_{\{m_{k_i}\}\in\mathscr{S}_{\mathbb{N}}^{(i)}}\bigcap_{i,j\geq 1}\left(\|\theta^*_{m_{k_i}}-\theta^*_{m_{k_j}}\|<\sqrt{\delta}\right)\right)\right.$$
$$\left.\bigcap\left(\bigcup_{\{m_i\}\in\mathscr{S}_{\mathbb{N}}}\left(\bigcap_{i=1}^{+\infty}\|\theta_{m_i}-\theta^*_{m_i}\|<\sqrt{\delta}\right)\right)\right)$$

Combine (47) and (48), then we get that

$$P\left(\left(\bigcap_{\{m_{k_i}\}\in\mathscr{S}_{\mathbb{N}}}\bigcup_{\{m_i\}\in\mathscr{S}_{\mathbb{N}}^{(i)}}\bigcap_{i,j\geq 1}\left(\|\theta^*_{m_{k_i}}-\theta^*_{m_{k_j}}\|<\sqrt{\delta}\right)\right)\right.$$
$$\left.\bigcap\left(\bigcup_{\{m_i\}\in\mathscr{S}_{\mathbb{N}}}\left(\bigcap_{i=1}^{+\infty}\|\theta_{m_i}-\theta^*_{m_i}\|<\sqrt{\delta}\right)\right)\right)=1.$$

So

$$P\left(\bigcup_{\{m_i\}\in\mathscr{S}_{\mathbb{N}}}\bigcap_{i,j\geq 1}\left(\|\theta^*_{m_i}-\theta^*_{m_j}\|<\sqrt{\delta}\right)\bigcap\left(\|\theta_{m_i}-\theta^*_{m_i}\|<\sqrt{\delta}\right)\right.$$
$$\left.\bigcap\left(\|\theta_{m_j}-\theta^*_{m_j}\|<\sqrt{\delta}\right)\right)=1.$$

Thus it holds that

$$P\left(\bigcap_{t=1}^{+\infty}\bigcup_{i,j>t}\left(\left(\|\theta_i^*-\theta_j^*\|\leq\sqrt{\delta}\right)\bigcap\left(\|\theta_i-\theta_j\|\leq 3\sqrt{\delta}\right)\right)\right)$$

$$\geq P\left(\bigcap_{t=1}^{+\infty}\bigcup_{i,j>t}\left(\left(\|\theta_i^*-\theta_j^*\|\leq\sqrt{\delta}\right)\bigcap\left(\|\theta_i-\theta_i^*\|\leq\sqrt{\delta}\right)\bigcap\left(\|\theta_j-\theta_j^*\|\leq\sqrt{\delta}\right)\right)\right)$$

$$=P\left(\bigcup_{\{m_{k_i}\}\in\mathscr{S}_{\mathbb{N}}}\bigcap_{i,j\geq 1}\left(\|\theta_{m_i}^*-\theta_{m_j}^*\|<\sqrt{\delta}\right)\bigcap\left(\|\theta_{m_i}-\theta_{m_i}^*\|<\sqrt{\delta}\right)\right.$$

$$\left.\bigcap\left(\|\theta_{m_j}-\theta_{m_j}^*\|<\sqrt{\delta}\right)\right)$$

$$=1.$$

This means we can find two "same" convergent subsequences $\{\theta_{k_n}\}$ and $\{\theta_{k_n}^*\}$ with probability one. Because $J$ is a closed set and $\{\theta_{k_n}^*\}\subset J$, we know that there exists $\theta''\in J$ such that

$$\lim_{n\to+\infty}\theta_{k_n}=\theta'' a.s. \tag{50}$$

This indicates that we can find an accumlative point of $\{\theta_n\}$ in $J$ with probability one. Denote the connected component containing $\theta''$ by $J^*$.

Step 3 is to prove that there is a connected component $J^*$ of $J$ such that $\theta_n\to J^*$ a.s. From equation 50, we have

$$P\left(\bigcap_{n=1}^{+\infty}\bigcup_{m=n}^{+\infty}\left(\|\theta_m-\theta''\|<\delta'\right)\right)=1.$$

Since $g(\theta)$ is continuous, we have

$$P\left(\bigcap_{n=1}^{+\infty}\bigcup_{m=n}^{+\infty}\left(|g(\theta_m)-g(\theta'')|<\delta''\right)\right)=1.$$

Since $\{g(\theta_n)\}$ converges a.s., let $M$ be the limit, i.e., $\lim_{n\to+\infty}g(\theta_n)=M$ a.s. Hence $g(\theta'')=M$, and $g(\theta)=M$ for all $\theta\in J^*$. Define $A=\{\theta|g(\theta)=M\}$. From $g(\theta_n)\to M$ a.s. and the continuity of $g(\theta)$, it holds that $\theta_n\to A$ a.s. Obviously, $J^*\subseteq A$. If $J^*=A$, then the conclusion follows. Now assume $J^*\subsetneqq A$. From the definition of $J^*$, it follows that there are two disjoint open sets $V$ and $H$ such that $A\subset H\bigcup V$, $J^*\subset V$, $A/J^*\subset H$, $H'\bigcap V'=\emptyset$ ($H'$ and $V'$ are closures of $H$ and $V$). So we just need to prove that $\theta_n\to V$ a.s. A contradiction argument is to be used. Suppose that $\{\theta_n\}$ has accumulation points in $V$ and $H$ simultaneously with a probability larger than zero. That is,

$$P\left(\left(\bigcap_{n=1}^{+\infty}\bigcup_{m=n}^{+\infty}(\theta_m\in H)\right)\bigcap\left(\bigcap_{s=1}^{+\infty}\bigcup_{t=s}^{+\infty}(\theta_t\in V)\right)\bigcap\left(\bigcup_{r=1}^{+\infty}\bigcap_{v=r}^{+\infty}(\theta_v\in H\bigcup V)\right)\right)>0. \tag{51}$$

Expanding equation 51, we get

$$P\left(\bigcup_{r=1}^{+\infty}\left(\bigcap_{(m,n)\in\mathbb{N}\times\mathbb{N}}\bigcup_{i\geq m,j\geq n}\left((\theta_i\in V)\bigcap(\theta_j\in H)\right)\right)\bigcap\left(\bigcap_{v=r}^{+\infty}(\theta_v\in H\bigcup V)\right)\right)>0.$$

Note that the event

$$\bigcap_{(m,n)\in\mathbb{N}\times\mathbb{N}}\bigcup_{i\geq m,j\geq n}\left((\theta_i\in V)\bigcap(\theta_j\in H)\right)$$

can be written as

$$\bigcup_{\{\alpha_i\}\in\mathscr{S}_{\mathbb{N}},\{\beta_i\}\in\mathscr{S}_{\mathbb{N}}}\bigcap_{i=1}^{+\infty}\left((\theta_{\alpha_i}\in V)\bigcap(\theta_{\beta_i}\in H)\right),$$

where $\{\alpha_i\} \cap \{\beta_i\} = \emptyset$. The event

$$\left( \bigcap_{(m,n)\in\mathbb{N}\times\mathbb{N}} \bigcup_{i\geq m, j\geq n} \left( (\theta_i \in V) \bigcap (\theta_j \in H) \right) \right) \bigcap \left( \bigcap_{v=r}^{+\infty} (\theta_v \in H \cup V) \right) \tag{52}$$

means $\forall t \geq r$, $\theta_t$ must belong to one of $H$ and $V$. So $\{\alpha_i\} \cup \{\beta_i\}$ must include the set $\{r, r+1, r+2, ...\}$. Denote $\{\alpha_i\}$ and $\{\beta_i\}$ as $\{\alpha_i^{(r)}\}$ and $\{\beta_i^{(r)}\}$, and then equation 52 can be written as

$$\bigcup_{\{\alpha_i^{(r)}\}\in\mathscr{S}_\mathbb{N}, \{\beta_i^{(r)}\}\in\mathscr{S}_\mathbb{N}} \bigcap_{i=1}^{+\infty} \left( (\theta_{\alpha_i^{(r)}} \in V) \bigcap (\theta_{\beta_i^{(r)}} \in H) \right).$$

So we get

$$P\left( \bigcup_{r=1}^{+\infty} \bigcup_{\{\alpha_i^{(r)}\}, \{\beta_i^{(r)}\}} \bigcap_{i=1}^{+\infty} \left( (\theta_{\alpha_i^{(r)}} \in V) \bigcap (\theta_{\beta_i^{(r)}} \in H) \right) \right) > 0. \tag{53}$$

It follows from $V' \bigcap H' = \emptyset$ that

$$\inf_{\theta^{(1)}\in H, \theta^{(2)}\in V} \left\| \theta^{(1)} - \theta^{(2)} \right\| = s > 0, \tag{54}$$

where $s$ is a constant. By Lemma 7, it holds that $\forall s > 0$

$$P\left( \bigcap_{n=1}^{+\infty} \bigcup_{m=n}^{+\infty} \|\theta_m - \theta_{m+1}\| \geq s \right) = 0. \tag{55}$$

Now we consider $\{\alpha_i^{(r)} + 1\}$. In fact, $\{\alpha_i^{(r)} + 1\}$ and $\{\beta_i^{(r)}\}$ have a finite number of identical elements with probability one. If not, then

$$P\left( \bigcap_{n=1}^{+\infty} \bigcup_{m=n}^{+\infty} \left( (\theta_{\alpha_m^{(r)}} \in V) \bigcap (\theta_{\alpha_m^{(r)}+1} \in H) \right) \right) > 0.$$

By equation 54, it follows that

$$P\left( \bigcap_{n=1}^{+\infty} \bigcup_{m=n}^{+\infty} \|\theta_m - \theta_{m+1}\| \geq s \right)$$
$$\geq P\left( \bigcap_{n=1}^{+\infty} \bigcup_{m=n}^{+\infty} \left\| \theta_{\alpha_m^{(r)}} - \theta_{\alpha_m^{(r)}+1} \right\| \geq s \right)$$
$$\geq P\left( \bigcap_{n=1}^{+\infty} \bigcup_{m=n}^{+\infty} \left( \left( \theta_{\alpha_m^{(r)}} \in V \right) \bigcap \left( \theta_{\alpha_m^{(r)}+1} \in H \right) \right) \right) > 0,$$

which however contradicts with equation 55. Hence $\{\alpha_i^{(r)} + 1\}$ and $\{\beta_i^{(r)}\}$ have a finite number of identical elements with probability one. From $\{\alpha_i^{(r)}\} \cup \{\beta_i^{(r)}\} \supset \{r, r+1, r+2, ...\}$, we know that

$$P\left( \bigcup_{n=1}^{+\infty} \bigcap_{m=n}^{+\infty} (\alpha_m^{(r)} + 1) \in \{\alpha_i^{(r)}\} \right) = 1. \tag{56}$$

Hence $\{\beta_i^{(r)}\}$ is a finite sequence with probability one, otherwise it contradicts with equation 53. Thus, the assumption does not hold, and $\theta_n \to V$ a.s. Furthermore, $\theta_n \to J^*$. Therefore, there is a connected component $J^*$ of $J$ such that

$$\lim_{n\to\infty} d(\theta_n, J^*) = 0.$$

### B.9 PROOF OF THEOREM 1

First of all we aim to prove that $g(\theta_{n+1})$ is convergent almost surely. Divide equation 44 into four parts as follows.

$$
g(\theta_{n+1}) = \underbrace{g(\theta_1) - \frac{1-\alpha^n}{1-\alpha}\nabla_{\theta_1}g(\theta_1)^\mathsf{T}v_1 + \frac{1-\alpha^n}{1-\alpha}\varepsilon_1\nabla_{\theta_1}g(\theta_1)\nabla_{\theta_1}g(\theta_1,\xi_1)}_{(A)}
$$

$$
\underbrace{-\sum_{t=1}^{n}\frac{1-\alpha^{n-t+1}}{1-\alpha}\varepsilon_t\nabla_{\theta_t}g(\theta_t)^\mathsf{T}\nabla_{\theta_t}g(\theta_t,\xi_t)}_{(B)} + \underbrace{\sum_{t=1}^{n}\left(\nabla_{\theta_{\zeta_t}}g(\theta_{\zeta_t}) - \nabla_{\theta_t}g(\theta_t)\right)^T(\theta_{t+1}-\theta_t)}_{(C)} \tag{57}
$$

$$
\underbrace{-\sum_{t=1}^{n}\frac{1-\alpha^{n-t+1}}{1-\alpha}\sum_{i=1}^{t-1}\alpha^{t-i}(\nabla_{\theta_i}g(\theta_i) - \nabla_{\theta_{i-1}}g(\theta_{i-1}))^T v_{i-1}}_{(D)}.
$$

Due to $\alpha < 1$, $\alpha^n$ is tending to zero, which ensures the convergence of part $(A)$. For $(C)$, we consider the absolute value of $\sum_{t=n}^{m}$. It follows from Assumption 2 3) that

$$
\left|\sum_{t=n}^{m}\left(\nabla_{\theta_{\zeta_t}}g(\theta_{\zeta_t}) - \nabla_{\theta_t}g(\theta_t)\right)^T(\theta_{t+1}-\theta_t)\right| \leq \sum_{t=n}^{m}\left|\left(\nabla_{\theta_{\zeta_t}}g(\theta_{\zeta_t}) - \nabla_{\theta_t}g(\theta_t)\right)^T(\theta_{t+1}-\theta_t)\right|
$$

$$
\leq \sum_{t=n}^{m}\left\|\nabla_{\theta_{\zeta_t}}g(\theta_{\zeta_t}) - \nabla_{\theta_t}g(\theta_t)\right\|\|v_t\| \leq c\sum_{t=n}^{m}\|v_t\|^2.
$$

Through Lemma 7, we get $\sum_{t=n}^{m}\|v_t\|^2 \to 0$ $a.s.$, leading to

$$
\left|\sum_{t=n}^{m}\left(\nabla_{\theta_{\zeta_t}}g(\theta_{\zeta_t}) - \nabla_{\theta_t}g(\theta_t)\right)^T(\theta_{t+1}-\theta_t)\right| \to 0 \ \ a.s..
$$

Through *Cauchy's test for convergence*, we know that $(C)$ is convergent almost surely. By using the same function, it holds that $(D)$ is convergent almost surely. For $(B)$, we have

$$
(B) = \sum_{t=1}^{n}\frac{1-\alpha^{n-t+1}}{1-\alpha}\varepsilon_t\nabla_{\theta_t}g(\theta_t)^\mathsf{T}\nabla_{\theta_t}g(\theta_t,\xi_t)
$$
$$
= \sum_{t=1}^{n}\frac{1-\alpha^{n-t+1}}{1-\alpha}\varepsilon_t\|\nabla_{\theta_t}g(\theta_t)\|^2 + \sum_{t=1}^{n}\frac{1-\alpha^{n-t+1}}{1-\alpha}\varepsilon_t\nabla_{\theta_t}g(\theta_t)^\mathsf{T}\left(\nabla_{\theta_t}g(\theta_t,\xi_t) - \nabla_{\theta_t}g(\theta_t)\right). \tag{58}
$$

From Lemma 7, it follows that $(C)$ is convergent almost surely. For $(B)$, it holds that

$$
(B) = \sum_{t=1}^{n}\frac{1-\alpha^{n-t+1}}{1-\alpha}\varepsilon_t\nabla_{\theta_t}g(\theta_t)^\mathsf{T}\nabla_{\theta_t}g(\theta_t,\xi_t)
$$
$$
= \sum_{t=1}^{n}\frac{1-\alpha^{n-t+1}}{1-\alpha}\varepsilon_t\|\nabla_{\theta_t}g(\theta_t)\|^2 + \sum_{t=1}^{n}\frac{1-\alpha^{n-t+1}}{1-\alpha}\varepsilon_t\nabla_{\theta_t}g(\theta_t)^\mathsf{T}\left(\nabla_{\theta_t}g(\theta_t,\xi_t) - \nabla_{\theta_t}g(\theta_t)\right). \tag{59}
$$

By Lemma 8, we know

$$
\sum_{t=1}^{n}\frac{1-\alpha^{n-t+1}}{1-\alpha}\varepsilon_t\|\nabla_{\theta_t}g(\theta_t)\|^2 < +\infty \ \ a.s..
$$

From Lemmas 5 and 8 it follows that

$$
\sum_{t=1}^{n-1}\frac{1-\alpha^{n-t}}{1-\alpha}\varepsilon_{t+1}\nabla_{\theta_t}g(\theta_t)^\mathsf{T}\left(\nabla_{\theta_t}g(\theta_t,\xi_t) - \nabla_{\theta_t}g(\theta_t)\right)
$$

is convergent a.s. Thus $(B)$ is convergent a.s., and $g(\theta_{n+1})$ is convergent a.s.. Substituting equation 59 into equation 57 leads to

$$
g(\theta_{n+1}) \leq \zeta_n' - \sum_{t=1}^{n}\varepsilon_t\left\|\nabla_{\theta_t}g(\theta_t)\right\|^2,
$$

where $\{\zeta'_n\}$ is defined as follows

$$
\begin{aligned}
\zeta'_n =\, & g(\theta_1) - \frac{1-\alpha^n}{1-\alpha}\nabla_{\theta_1}g(\theta_1)^\mathsf{T}v_1 + \frac{1-\alpha^n}{1-\alpha}\varepsilon_1\nabla_{\theta_1}g(\theta_1)\nabla_{\theta_1}g(\theta_1,\xi_1) \\
& - \sum_{t=1}^{n}\frac{1-\alpha^{n-t+1}}{1-\alpha}\varepsilon_t\nabla_{\theta_t}g(\theta_t)^\mathsf{T}\left(\nabla_{\theta_t}g(\theta_t,\xi_t)-\nabla_{\theta_t}g(\theta_t)\right) + \sum_{t=1}^{n}\left(\nabla_{\theta_{\zeta_t}}g(\theta_{\zeta_t})-\nabla_{\theta_t}g(\theta_t)\right)^T(\theta_{t+1}-\theta_t) \\
& - \sum_{t=1}^{n}\frac{1-\alpha^{n-t+1}}{1-\alpha}\sum_{i=1}^{t-1}\alpha^{t-i}(\nabla_{\theta_i}g(\theta_i)-\nabla_{\theta_{i-1}}g(\theta_{i-1}))^T v_{i-1}.
\end{aligned}
$$

we know $\{\zeta'_n\}$ is convergent a.s. It follows from Lemma 9 that there exists a connected component $J^*$ of $J$ such that $\lim\limits_{n\to\infty} d(\theta_n, J^*) = 0$.

### B.10  PROOF OF THEOREM 2

First of all we can get that

$$
\begin{aligned}
g(\theta_{t+1})-g(\theta_t) = & -\Big(\alpha^{t-1}\nabla_{\theta_1}g(\theta_1)^\mathsf{T}v_1 + \sum_{i=1}^{t-1}\alpha^{t-i}(\nabla_{\theta_i}g(\theta_i)-\nabla_{\theta_{i-1}}g(\theta_{i-1}))^T v_{i-1} \\
& + \sum_{i=2}^{t}\alpha^{t-i}\varepsilon_i\nabla_{\theta_i}g(\theta_i)^\mathsf{T}\nabla_{\theta_i}g(\theta_i,\xi_i)\Big) + \left(\nabla_{\theta_{\zeta_t}}g(\theta_{\zeta_t})-\nabla_{\theta_t}g(\theta_t)\right)^T(\theta_{t+1}-\theta_t).
\end{aligned}
\tag{60}
$$

From Theorem 1, it follows that $g(\theta_n)$ is convergent a.s., and it is orbitally convergent to $g_i$ ($i = 1, 2, ..., N$) a.s.. Then it holds that

$$
\lim_{n\to+\infty} g(\theta_n) = \sum_{i=1}^{N} I_i g_i \quad a.s.,
$$

where

$$
I_i = \begin{cases} 1 & \lim_{n\to+\infty} g(\theta_n) = g_i \\[2mm] 0 & \lim_{n\to+\infty} g(\theta_n) \neq g_i \end{cases}
$$

For convenient, we let $g^* = \sum_{i=1}^{\infty} I_i g_i$. Then we make some transformation on equation 60

$$
\begin{aligned}
\left(g(\theta_{t+1})-g^*\right)-\left(g(\theta_t)-g^*\right) = & -\Big(\alpha^{t-1}\nabla_{\theta_1}g(\theta_1)^\mathsf{T}v_1 + \sum_{i=1}^{t-1}\alpha^{t-i}(\nabla_{\theta_i}g(\theta_i)-\nabla_{\theta_{i-1}}g(\theta_{i-1}))^T v_{i-1} \\
& + \sum_{i=2}^{t}\alpha^{t-i}\varepsilon_i\nabla_{\theta_i}g(\theta_i)^\mathsf{T}\nabla_{\theta_i}g(\theta_i,\xi_i)\Big) + \left(\nabla_{\theta_{\zeta_t}}g(\theta_{\zeta_t})-\nabla_{\theta_t}g(\theta_t)\right)^T(\theta_{t+1}-\theta_t).
\end{aligned}
\tag{61}
$$

Then we make some transformations, take absolute values, take the mathematical expectation, and use same techniques in Theorem 1. Since the sampling noise follows a uniform distribution, there is $\mathbb{E}(\|\nabla_{\theta_n}g(\theta_n,\xi_n)\|^2) \leq M\,\mathbb{E}(\|\nabla_{\theta_n}g(\theta_n)\|^2)$. So it follows that $F'_n - F'_{n-1} \leq P_{n-1} - Q_{n-1}$, where $P_{n-1}$, $Q_{n-1}$, and $F'_n$ are defined as follows

$$
\begin{aligned}
P_{n-1} = & \frac{1}{(1-\alpha)^2}\left(\frac{1}{2-\alpha}\right)^{n-1}\varepsilon_1\,\mathbb{E}\left(\|\nabla_{\theta_1}g(\theta_1)\|^2\right) + Z(1)L\left(\frac{1}{2-\alpha}\right)^{n-1}\mathbb{E}\left(g(\theta_1)\right) \\
& - \frac{\alpha^n(2-\alpha)}{1-\alpha}Z(n+1)\,\mathbb{E}\left(\nabla_{\theta_1}g(\theta_1)^\mathsf{T}v_1\right) + \frac{c\alpha^n\alpha^\delta(2-\alpha)}{2(1-\alpha)(1-\alpha^\delta)}Z(1)\,\mathbb{E}\left(\|v_0\|^2\right), \\
F'_n = & \sum_{t=1}^{n}\left(\frac{1}{2-\alpha}\right)^{n-t}Z(t+1)\,\mathbb{E}\left(e_{t+1}^{(n)}\big|g(\theta_{t+1})-g^*\big|\right), \\
Q_{n-1} = & \frac{1}{(1-\alpha)^2}\sum_{t=1}^{n}\left(\frac{1}{2-\alpha}\right)^{n-t}\varepsilon_t\,\mathbb{E}\left(\|\nabla_{\theta_t}g(\theta_t)\|^2\right),
\end{aligned}
\tag{62}
$$

where $g^* = \inf_{\theta\in\mathbb{R}^N} g(\theta)$. Then we have

$$
F'_n \leq \left(1 - \frac{Q_{n-1}}{F'_{n-1}}\right)F'_{n-1} + P_{n-1}.
\tag{63}
$$

Derive $Q_n/\varepsilon_{n+1}F'_n$ as follows

$$
\frac{Q_n}{\varepsilon_{n+1}F'_n} = \frac{1}{(1-\alpha)^2} \frac{\sum_{t=1}^{n+1}\left(\frac{1}{2-\alpha}\right)^{n+1-t}\varepsilon_{t+1}\,\mathbb{E}\left(\|\nabla_{\theta_t}g(\theta_t)\|^2\right)}{\sum_{t=1}^{n}\left(\frac{1}{2-\alpha}\right)^{-t}Z(t+1)Z(t+1)\,\mathbb{E}\left(e^{(n)}_{t+1}\big|g(\theta_{t+1})-g^*\big|\right)}
$$

$$
= \frac{1}{(1-\alpha)^2}\frac{\left(\frac{1}{2-\alpha}\right)^{n}\varepsilon_2\,\mathbb{E}\left(\|\nabla_{\theta_1}g(\theta_1)\|^2\right)+\sum_{t=1}^{n}\left(\frac{1}{2-\alpha}\right)^{-t}\mathbb{E}\left(\|\nabla_{\theta_{t+1}}g(\theta_{t+1})\|^2\right)}{\sum_{t=1}^{n}\left(\frac{1}{2-\alpha}\right)^{-t}Z(t+1)Z(t+1)\,\mathbb{E}\left(e^{(n)}_{t+1}\big|g(\theta_{t+1})-g^*\big|\right)}.
$$

It follows that

$$
\liminf_{n\to+\infty}\frac{Q_n}{\varepsilon_{n+1}F'_n}
$$
$$
= \frac{1}{(1-\alpha)^2}\liminf_{n\to+\infty}\frac{\left(\frac{1}{2-\alpha}\right)^{n}\varepsilon_2\,\mathbb{E}\left(\|\nabla_{\theta_1}g(\theta_1)\|^2\right)+\sum_{t=1}^{n}\left(\frac{1}{2-\alpha}\right)^{-t}\mathbb{E}\left(\|\nabla_{\theta_{t+1}}g(\theta_{t+1})\|^2\right)}{\sum_{t=1}^{n}\left(\frac{1}{2-\alpha}\right)^{-t}Z(t+1)\,\mathbb{E}\left(e^{(n)}_{t+1}\big|g(\theta_{t+1})-g^*\big|\right)} \tag{64}
$$
$$
\geq \frac{1}{(1-\alpha)^2}\liminf_{n\to+\infty}\frac{\sum_{t=1}^{n}\left(\frac{1}{2-\alpha}\right)^{-t}\mathbb{E}\left(\|\nabla_{\theta_{t+1}}g(\theta_{t+1})\|^2\right)}{\sum_{t=1}^{n}\left(\frac{1}{2-\alpha}\right)^{-t}Z(t+1)\,\mathbb{E}\left(e^{(n)}_{t+1}\big|g(\theta_{t+1})-g^*\big|\right)}.
$$

From equation 32, we have

$$
e^{(n)}_{t+1} = 1 + \frac{2c\varepsilon_t f(n-t)}{\alpha^{1-\delta}(1-\alpha^\delta)}
$$
$$
= 1 + \frac{2c\varepsilon_t}{\alpha^{1-\delta}(1-\alpha^\delta)}\left(\sum_{k=1}^{n-t}\left(\alpha(2-\alpha)\right)^k - \alpha^\delta\sum_{k=1}^{n-t}\left(\alpha^{1+\delta}(2-\alpha)\right)^k\right) \tag{65}
$$
$$
\leq 1 + C_\alpha\varepsilon_t,
$$

where $C_\alpha = \frac{2c(2-\alpha)}{\alpha^{-\delta}(1-\alpha^\delta)(1-\alpha(2-\alpha))}$. Substituting equation 65 into equation 64 yields

$$
\liminf_{n\to+\infty}\frac{Q_n}{\varepsilon_{n+1}F'_n}
$$
$$
\geq \frac{1}{(1-\alpha)^2}\liminf_{n\to+\infty}\frac{\sum_{t=1}^{n}\left(\frac{1}{2-\alpha}\right)^{-t}\mathbb{E}\left(\|\nabla_{\theta_{t+1}}g(\theta_{t+1})\|^2\right)}{\sum_{t=1}^{n}\left(\frac{1}{2-\alpha}\right)^{-t}Z(t+1)\,\mathbb{E}\left(e^{(n)}_{t+1}\big|g(\theta_{t+1})-g^*\big|\right)}. \tag{66}
$$

Then we proceed with the proof under two different cases, namely, $\sum_{t=1}^{n}\left(\frac{1}{2-\alpha}\right)^{-t}\mathbb{E}\left(\|\nabla_{\theta_{t+1}}g(\theta_{t+1})\|^2\right) = +\infty$ and $\sum_{t=1}^{n}\left(\frac{1}{2-\alpha}\right)^{-t}\mathbb{E}\left(\|\nabla_{\theta_{t+1}}g(\theta_{t+1})\|^2\right) < +\infty$.

First, if $\sum_{t=1}^{n}\left(\frac{1}{2-\alpha}\right)^{-t}\mathbb{E}\left(\|\nabla_{\theta_{t+1}}g(\theta_{t+1})\|^2\right) = +\infty$ (the proof for this condition is up to equation 76). It follows from Assumption 4 2) and the uniform convergence and $O'stolz\ theorem$ that

$$
\liminf_{n\to+\infty}\frac{Q_n}{\varepsilon_{n+1}F'_n}
$$
$$
\geq \frac{1}{(1-\alpha)^2}\liminf_{n\to+\infty}\frac{\sum_{t=1}^{n}\left(\frac{1}{2-\alpha}\right)^{-t}\mathbb{E}\left(\|\nabla_{\theta_{t+1}}g(\theta_{t+1})\|^2\right)}{\sum_{t=1}^{n}\left(\frac{1}{2-\alpha}\right)^{-t}Z(t+1)\,\mathbb{E}\left(\left(1+C_\alpha\varepsilon_t\right)\big|g(\theta_{t+1})-g^*\big|\right)}
$$
$$
\geq \frac{1}{(1-\alpha)^2}\liminf_{n\to+\infty}\frac{\mathbb{E}\left(\|\nabla_{\theta_{n+1}}g(\theta_{n+1})\|^2\right)}{Z(n+1)\,\mathbb{E}\left(\left(1+C_\alpha\varepsilon_t\right)\big|g(\theta_{t+1})-g^*\big|\right)} \tag{67}
$$
$$
\geq \frac{1}{(1-\alpha)^2}\liminf_{n\to+\infty}\frac{s}{Z(n+1)\left(1+C_\alpha\varepsilon_n\right)} = \frac{s}{p(1-\alpha)^2},
$$

where $p = \exp\left\{\sum_{k=1}^{\infty} M\varepsilon_k^2\right\}$. By using O'stolz theorem on $Q_n/F_n'$, it follows that

$$\lim_{n \to +\infty} \frac{Q_n}{F_n'} = \frac{1}{(1-\alpha)^2} \lim_{n \to +\infty} \frac{\sum_{t=1}^{n} \left(\frac{1}{2-\alpha}\right)^{-t} \varepsilon_{t+1} \mathbb{E}\left(\|\nabla_{\theta_{t+1}} g(\theta_{t+1})\|^2\right)}{\sum_{t=1}^{n} \left(\frac{1}{2-\alpha}\right)^{-t} Z(t+1) \mathbb{E}\left(e_{t+1}^{(n)} |g(\theta_{t+1}) - g^*|\right)}$$

$$\leq \frac{1}{(1-\alpha)^2} \lim_{n \to +\infty} \frac{\sum_{t=1}^{n} \left(\frac{1}{2-\alpha}\right)^{-t} \varepsilon_{t+1} \mathbb{E}\left(\|\nabla_{\theta_{t+1}} g(\theta_{t+1})\|^2\right)}{\sum_{t=1}^{n} \left(\frac{1}{2-\alpha}\right)^{-t} Z(t+1) \mathbb{E}|g(\theta_{t+1}) - g^*|}$$

$$= \frac{1}{(1-\alpha)^2} \lim_{n \to +\infty} \frac{\varepsilon_n \mathbb{E}\left(\|\nabla_{\theta_n} g(\theta_n)\|^2\right)}{Z(n+1) \mathbb{E}|g(\theta_{n+1}) - g^*|}$$

$$\leq \frac{1}{(1-\alpha)^2} \lim_{n \to +\infty} \frac{t\varepsilon_n \mathbb{E}\left(g(\theta_{n+1}) - g^*\right)}{Z(n+1) \mathbb{E}|g(\theta_{n+1}) - g^*|} = 0.$$

So we conclude that $\exists n_0 \in \mathbb{N}_+$, such that $\forall n \geq n_0$,

$$\frac{Q_n}{F_n'} < 1 - \frac{3-\alpha}{2(2-\alpha)}. \tag{68}$$

From equation 63, it follows that

$$F_n' \leq \prod_{i=n_0}^{n-1} \left(1 - \frac{Q_i}{F_i'}\right) F_{n_0}' + \sum_{t=n_0+1}^{n} \prod_{i=t}^{n-1} \left(1 - \frac{Q_i}{F_i'}\right) P_{t-1}. \tag{69}$$

Using the inequality $\ln(1+x) \leq x$ and equation 67, we get

$$\prod_{i=n_0}^{n-1} \left(1 - \frac{Q_i}{F_i'}\right) = \exp\left(\sum_{i=n_0}^{n-1} \ln\left(1 - \frac{Q_i}{F_i'}\right)\right) \leq \exp\left(-\sum_{i=n_0}^{n-1} \frac{Q_i}{F_i'}\right) \leq k e^{-\sum_{i=1}^{n} \frac{s\varepsilon_i}{p(1-\alpha)^2}}$$

$$= O\left(e^{-\sum_{i=1}^{n} \frac{s\varepsilon_i}{p(1-\alpha)^2}}\right), \tag{70}$$

where $k$ is a constant. It follows that

$$\frac{\sum_{t=n_0+1}^{n-1} \prod_{i=t}^{n-1} \left(1 - \frac{Q_i}{F_i'}\right) P_{t-1}}{\prod_{i=n_0}^{n-1} \left(1 - \frac{Q_i}{F_i'}\right) F_1'} = \sum_{t=n_0+1}^{n} \frac{P_{t-1}}{\prod_{i=n_0}^{t-1} \left(1 - \frac{Q_i}{F_i'}\right)}. \tag{71}$$

From equation 62, we have

$$P_{t-1} = \frac{1}{(1-\alpha)^2} \left(\frac{1}{2-\alpha}\right)^{t-1} \varepsilon_1 \mathbb{E}\left(\|\nabla_{\theta_1} g(\theta_1)\|^2\right) + Z(1)L \left(\frac{1}{2-\alpha}\right)^{t-1} \mathbb{E}\left(g(\theta_1)\right)$$

$$+ \frac{c\alpha^t \alpha^\delta (2-\alpha)}{2(1-\alpha)(1-\alpha^\delta)} Z(1) \mathbb{E}\left(\|v_0\|^2\right) = \bar{p} \left(\frac{1}{2-\alpha}\right)^{t-1} + \bar{q}\alpha^{t-1}, \tag{72}$$

where $\bar{p} = \frac{1}{(1-\alpha)^2} \varepsilon_1 \mathbb{E}\left(\|\nabla_{\theta_1} g(\theta_1)\|^2\right) + Z(1)L \mathbb{E}\left(g(\theta_1)\right)$ and $\bar{q} = \frac{c\alpha^{\delta+1}(2-\alpha)}{2(1-\alpha)(1-\alpha^\delta)} Z(1) \mathbb{E}\left(\|v_0\|^2\right)$. Substituting equation 72 into equation 71 and noting equation 68 yield

$$\frac{\sum_{t=n_0+1}^{n-1} \prod_{i=t}^{n-1} \left(1 - \frac{Q_i}{F_i'}\right) P_{t-1}}{\prod_{i=n_0}^{n-1} \left(1 - \frac{Q_i}{F_i'}\right) F_1'}$$

$$= \sum_{t=n_0+1}^{n} \frac{P_{t-1}}{\prod_{i=n_0}^{t-1} \left(1 - \frac{Q_i}{F_i'}\right)} < \sum_{t=n_0+1}^{n} \frac{\bar{p}\left(\frac{1}{2-\alpha}\right)^{t-1}}{\prod_{i=n_0}^{t-1} \left(1 - \frac{Q_i}{F_i'}\right)} + \sum_{t=n_0+1}^{n} \frac{\bar{q}\alpha^{t-1}}{\prod_{i=n_0}^{t-1} \left(1 - \frac{Q_i}{F_i'}\right)}$$

$$\leq \sum_{t=n_0+1}^{n} \frac{\bar{p}\left(\frac{1}{2-\alpha}\right)^{t-1}}{\left(\frac{3-\alpha}{2(2-\alpha)}\right)^{t-n_0}} + \sum_{t=n_0+1}^{n} \frac{\bar{q}\alpha^{t-1}}{\left(\frac{3-\alpha}{2(2-\alpha)}\right)^{t-n_0}}$$

$$< \left(\frac{3-\alpha}{2(2-\alpha)}\right)^{n_0-1} (\bar{p} + \bar{q}) \sum_{t=n_0+1}^{n} \left(\frac{2}{3-\alpha}\right)^{t-1} < c_0, \tag{73}$$

where $c_0$ is a positive constant. It follows that

$$\sum_{t=n_0+1}^{n-1} \prod_{i=t}^{n-1} \left(1 - \frac{Q_i}{F_i'}\right) P_{t-1} < c_0 \prod_{i=n_0}^{n-1} \left(1 - \frac{Q_i}{F_i'}\right) F_1' = O\left(\prod_{i=n_0}^{n-1} \left(1 - \frac{Q_i}{F_i'}\right) F_1'\right) = O\left(e^{-\sum_{i=1}^{n} \frac{s\varepsilon_i}{p(1-\alpha)^2}}\right).$$

(74)

Combine equation 69, equation 70 and equation 74, then we get

$$F_n' \le \prod_{i=n_0}^{n-1} \left(1 - \frac{Q_i}{F_i'}\right) F_1' + \sum_{t=n_0+1}^{n-1} \prod_{i=t}^{n-1} \left(1 - \frac{Q_i}{F_i'}\right) P_{t-1} = O\left(e^{-\sum_{i=1}^{n} \frac{s\varepsilon_i}{p(1-\alpha)^2}}\right).$$

(75)

In addition, we have

$$\mathbb{E}\left(g(\theta_{t+1}) - g^*\right) < F_n' = O\left(e^{-\sum_{i=1}^{n} \frac{s\varepsilon_i}{p(1-\alpha)^2}}\right).$$

(76)

If $\sum_{t=1}^{n} \left(\frac{1}{2-\alpha}\right)^{-t} \mathbb{E}\left(\|\nabla_{\theta_{t+1}} g(\theta_{t+1})\|^2\right) < +\infty$, it holds that

$$\lim_{n \to +\infty} \left(\frac{1}{2-\alpha}\right)^{-n} \mathbb{E}\left(\|\nabla_{\theta_{n+1}} g(\theta_{n+1})\|^2 = 0,$$

that is

$$\mathbb{E}\left(\|\nabla_{\theta_{n+1}} g(\theta_{n+1})\|^2\right) = O\left(\left(\frac{1}{2-\alpha}\right)^n\right).$$

Under Assumption 4 2), we have

$$\limsup_{n \to +\infty} \frac{\mathbb{E}\left(g(\theta_{n+1}) - g^*\right)}{\left(\frac{1}{2-\alpha}\right)^n} \le \limsup_{n \to +\infty} \frac{\mathbb{E}\left(\|\nabla_{\theta_{n+1}} g(\theta_{n+1})\|^2\right)}{s\left(\frac{1}{2-\alpha}\right)^n} \le \frac{1}{s} \frac{\mathbb{E}\left(\|\nabla_{\theta_{n+1}} g(\theta_{n+1})\|^2\right)}{\left(\frac{1}{2-\alpha}\right)^n} = 0.$$

It follows that

$$\mathbb{E}\left|g(\theta_{n+1}) - g^*\right| = O\left(\left(\frac{1}{2-\alpha}\right)^n\right).$$

Now we compare $\left(\frac{1}{2-\alpha}\right)^n$ with $e^{-\sum_{i=1}^{n} \frac{s\varepsilon_i}{p(1-\alpha)^2}}$. It holds that

$$\frac{\left(\frac{1}{2-\alpha}\right)^n}{\exp\left(-\sum_{i=1}^{n} \frac{s\varepsilon_i}{p(1-\alpha)^2}\right)} = \frac{\exp\left(-n\ln(2-\alpha)\right)}{\exp\left(-\sum_{i=1}^{n} \frac{s\varepsilon_i}{p(1-\alpha)^2}\right)} = \exp\left(\sum_{i=1}^{n} \frac{s\varepsilon_i}{p(1-\alpha)^2} - n\ln(2-\alpha)\right).$$

It follows from Assumption 3 that $\varepsilon_n \to 0$, indicating $\sum_{i=1}^{n} \frac{s\varepsilon_i}{p(1-\alpha)^2} - n\ln(2-\alpha) < 0$ when $n$ is sufficiently large. Thus, we have

$$\limsup_{n \to +\infty} \frac{\left(\frac{1}{2-\alpha}\right)^n}{\exp\left(-\sum_{i=1}^{n} \frac{s\varepsilon_i}{p(1-\alpha)^2}\right)} = \limsup_{n \to +\infty} \exp\left(\sum_{i=1}^{n} \frac{s\varepsilon_i}{p(1-\alpha)^2} - n\ln(2-\alpha)\right) < 1.$$

Then it holds that

$$\mathbb{E}\left|g(\theta_{n+1}) - g^*\right| = O\left(\left(\frac{1}{2-\alpha}\right)^n\right) = O\left(e^{-\sum_{i=1}^{n} \frac{s\varepsilon_i}{p(1-\alpha)^2}}\right).$$

(77)

Combining equation 77 and equation 76 leads to

$$\mathbb{E}\left|g(\theta_{n+1}) - g^*\right| = O\left(e^{-\sum_{i=1}^{n} \frac{s\varepsilon_i}{p(1-\alpha)^2}}\right),$$

if $\sum_{t=1}^{n} \left(\frac{1}{2-\alpha}\right)^{-t} \mathbb{E}\left(\|\nabla_{\theta_{t+1}} g(\theta_{t+1})\|^2\right) < +\infty$. The above bound holds trivially if $\sum_{t=1}^{n} \left(\frac{1}{2-\alpha}\right)^{-t} \mathbb{E}\left(\|\nabla_{\theta_{t+1}} g(\theta_{t+1})\|^2\right) = +\infty$. It follows from Lemma 4 that

$$\mathbb{E}\left(\|\nabla_{\theta_n} g(\theta_n)\|^2\right) = O\left(e^{-\sum_{i=1}^{n} \frac{s\varepsilon_i}{p(1-\alpha)^2}}\right)$$

## C CONVERGENCE OF ADAGRAD

The following lemmas are used for the proof of Theorem 3.

**Lemma 10** *Suppose $f(x) \in C^1$ $(x \in \mathbb{R}^N)$ with $f(x) > -\infty$ and its gradient satisfying the following Lipschitz condition*

$$\left\| \nabla f(x) - \nabla f(y) \right\| \leq c \|x - y\|,$$

*then $\forall x_0 \in \mathbb{R}^N$, there is*

$$\left\| \nabla f(x_0) \right\|^2 \leq 2c\big(f(x_0) - f^*\big),$$

*where $f^* = \inf_{x \in \mathbb{R}^N} f(x)$*

**Lemma 11** *Suppose $\{\theta_n\}$ is a sequence generated by AdaGrad in equation 5, and Assumptions 1 and 5 hold. If $\left\| \nabla_{\theta_n} g(\theta) \right\|^2 > a$ where $a$ is given in Assumption 5 3), then for any $n \in \mathbb{N}_+, \theta_1 \in \mathbb{R}^N$, and $\varepsilon \in (0, \frac{1}{2})$, it holds that*

$$\frac{g(\theta_{n+1})}{S_{n+1}^\varepsilon} - \frac{g(\theta_n)}{S_n^\varepsilon} \leq \frac{\alpha_0}{2}(M+1)\left( \frac{\left\| \nabla_{\theta_{n-1}} g(\theta_{n-1}) \right\|^2}{S_{n-1}^{\frac{1}{2}+\varepsilon}} - \frac{\left\| \nabla_{\theta_n} g(\theta_n) \right\|^2}{S_n^{\frac{1}{2}+\varepsilon}} \right)$$

$$- \frac{\alpha_0}{20} \frac{\left\| \nabla_{\theta_n} g(\theta_n) \right\|^2}{S_{n-1}^{\frac{1}{2}+\varepsilon}} + \frac{\alpha_0}{20}\left( \frac{\left\| \nabla_{\theta_{n-1}} g(\theta_{n-1}) \right\|^2}{S_{n-2}^{\frac{1}{2}+\varepsilon}} - \frac{\left\| \nabla_{\theta_n} g(\theta_n) \right\|^2}{S_{n-1}^{\frac{1}{2}+\varepsilon}} \right)$$

$$+ 4M^2 \alpha_0^3 c^2 \frac{\left\| \nabla_{\theta_{n-1}} g(\theta_{n-1}, \xi_{n-1}) \right\|^2}{S_{n-1}^{\frac{3}{2}+\varepsilon}} + \frac{c\alpha_0^2}{2} \frac{\left\| \nabla_{\theta_n} g(\theta_n, \xi_n) \right\|^2}{S_n^{1+\varepsilon}} + X_n^{(\varepsilon)} + Y_n^{(\varepsilon)},$$

*where*

$$X_n^{(\varepsilon)} = \frac{\alpha_0}{2} \frac{1}{S_{n-1}^{\frac{1}{2}+\varepsilon}} \nabla_{\theta_n} g(\theta_n)^T \big( \nabla_{\theta_n} g(\theta_n) - \nabla_{\theta_n} g(\theta_n, \xi_n) \big)$$

$$Y_n^{(\varepsilon)} = \frac{\alpha_0}{2} \left( \frac{1}{M+1} \frac{\mathbb{E}\left( \left\| \nabla_{\theta_n} g(\theta_n, \xi_n) \right\|^2 \big| \mathscr{F}_{n-1} \right)}{S_{n-1}^{\frac{1}{2}+\varepsilon}} - \frac{1}{M+1} \frac{\left\| \nabla_{\theta_n} g(\theta_n, \xi_n) \right\|^2}{S_{n-1}^{\frac{1}{2}+\varepsilon}} \right).$$

**Lemma 12** *Suppose $\{\theta_n\}$ is a sequence generated by AdaGrad in equation 5, and Assumptions 1 and 5 hold. If $\left\| \nabla_{\theta_n} g(\theta) \right\|^2 \leq a$ where $a$ is given in Assumption 5 3), then for any $n \in \mathbb{N}_+, \theta_1 \in \mathbb{R}^N$, and $\varepsilon \in (0, \frac{1}{2})$, it holds that*

$$\frac{g(\theta_{n+1})}{S_{n+1}^\varepsilon} - \frac{g(\theta_n)}{S_n^\varepsilon} \leq - \frac{\alpha_0 \left\| \nabla_{\theta_n} g(\theta_n) \right\|^2}{20 S_{n-1}^{\frac{1}{2}+\varepsilon}} + \frac{\alpha_0^3 c^2 (M+1)^2}{2} \frac{\left\| \nabla_{\theta_{n-1}} g(\theta_{n-1}, \xi_{n-1}) \right\|^2}{S_{n-1}^{1+\varepsilon}}$$

$$+ \frac{(M+1)\alpha_0^3 c^2}{2 S_{n-1}^{\frac{3}{2}+\varepsilon}} \left\| \nabla_{\theta_{n-1}} g(\theta_{n-1}, \xi_{n-1}) \right\|^2 + \frac{\alpha_0}{20}\left( \frac{\left\| \nabla_{\theta_{n-1}} g(\theta_{n-1}) \right\|^2}{S_{n-2}^{\frac{1}{2}+\varepsilon}} - \frac{\left\| \nabla_{\theta_n} g(\theta_n) \right\|^2}{S_{n-1}^{\frac{1}{2}+\varepsilon}} \right)$$

$$+ \frac{\alpha_0(M+1)}{2}\left( \frac{\left\| \nabla_{\theta_{n-1}} g(\theta_{n-1}) \right\|^2}{S_{n-1}^{\frac{1}{2}+\varepsilon}} - \frac{\left\| \nabla_{\theta_n} g(\theta_n) \right\|^2}{S_n^{\frac{1}{2}+\varepsilon}} \right) + \frac{\alpha_0 a(M+1)}{2}\left( \frac{1}{S_{n-1}^{\frac{1}{2}+\varepsilon}} - \frac{1}{S_n^{\frac{1}{2}+\varepsilon}} \right)$$

$$+ \frac{c\alpha_0^2}{2} \frac{\left\| \nabla_{\theta_n} g(\theta_n, \xi_n) \right\|^2}{S_n^{1+\varepsilon}} + A_n^{(\varepsilon)} + B_n^{(\varepsilon)},$$

*where*

$$A_n^{(\varepsilon)} = \frac{\alpha_0}{S_{n-1}^{\frac{1}{2}+\varepsilon}}\left( \left\| \nabla_{\theta_n} g(\theta_n) \right\|^2 - \nabla_{\theta_n} g(\theta_n)^T \nabla_{\theta_n} g(\theta_n, \xi_n) \right)$$

$$B_n^{(\varepsilon)} = \frac{\alpha_0 \left\| \nabla_{\theta_n} g(\theta_n) \right\|^2}{2a(M+1) S_{n-1}^{\frac{1}{2}+\varepsilon}}\left( \left\| \nabla_{\theta_n} g(\theta_n, \xi_n) \right\|^2 - \mathbb{E}\left( \left\| \nabla_{\theta_n} g(\theta_n) \right\|^2 \big| \mathscr{F}_{n-1} \right) \right).$$

**Lemma 13** *Suppose $\{\theta_n\}$ is a sequence generated by AdaGrad in equation 5, and Assumptions 1 and 5 hold. Then $\forall n \in \mathbb{N}_+$, $\forall \theta_1 \in \mathbb{R}^N$ $\forall \varepsilon \in (0, \frac{1}{2})$, it holds that*

$$\sum_{k=3}^{n} \frac{\left\| \nabla_{\theta_k} g(\theta_k) \right\|^2}{S_{k-1}^{\frac{1}{2}+\varepsilon}} < +\infty \ \ a.s..$$

**Lemma 14** *Suppose $\{\theta_n\}$ is a sequence generated by AdaGrad in equation 5, and Assumptions 1 and 5 hold. Then $\forall n \in \mathbb{N}_+$ $\forall \theta_1 \in \mathbb{R}^N$ $\forall \varepsilon \in (0, \frac{1}{2})$, $\exists \zeta < +\infty$, it holds that*

$$\frac{g(\theta_{n+1}) - g^*}{S_{n+1}^{\varepsilon}} \le \zeta < +\infty \ \ a.s.,$$

*which $g^* = \inf_{\theta \in \mathbb{R}^N} g(\theta)$.*

**Lemma 15** *Suppose $\{\theta_n\}$ is a sequence generated by AdaGrad in equation 5 subject to $S_n = \sum_{k=1}^{n} \|\nabla_{\theta_k} g(\theta_k, \xi_k)\|^2 = +\infty$ a.s.. Under Assumptions 1 and 5, $\forall n \in \mathbb{N}_+$ $\forall \theta_1 \in \mathbb{R}^N$, $\forall \varepsilon_0 \in (0, \frac{3}{8})$, it holds that*

$$\frac{\left\| \nabla_{\theta_n} g(\theta_n) \right\|^2}{S_{n-1}^{\varepsilon_0}} \to 0 \ \ a.s..$$

**Lemma 16** *Suppose that $\{X_n\} \in \mathbb{R}^N$ is a vector sequence and $f(x) \in C^1$ is a monotonically non-increasing non-negative function with $\int_a^{+\infty} f(x)dx < \infty$ ($\forall a > 0$). Then $\forall N \in \mathbb{N}_+$, it holds that*

$$\sum_{n=1}^{N} \|X_n\|^2 f\left( \sum_{k=1}^{n} \|X_k\|^2 \right) < \int_{\|X_1\|^2}^{\sum_{k=1}^{N} \|X_k\|^2} f(x)dx < \sum_{n=1}^{N} \|X_n\|^2 f\left( \sum_{k=1}^{n-1} \|X_k\|^2 \right).$$

## C.1 PROOF OUTLINE OF THEOREM 3

Like the proof of mSGD, the proof for AdaGrad is also in light of the Lyapunov method. We aim to prove $\nabla g(\theta_n) \to 0$ *a.s.*, and then to get $\theta_n \to J^*$ *a.s.* The key step to prove convergence of AdaGrad is to show

$$-\mathbb{E}\left( \frac{\alpha_0 \nabla_{\theta_n} g(\theta_n, \xi_n)^T \nabla_{\theta_n} g(\theta_n, \xi_n)}{\sqrt{S_n}} \middle| \mathscr{F}_n \right) \le 0.$$

but the learning rate of AdaGrad is a random variable and it is not conditionally independent of $\nabla_{\theta_n} g(\theta_n, \xi_n)$, meaning that

$$-\mathbb{E}\left( \frac{\alpha_0 \nabla_{\theta_n} g(\theta_n, \xi_n)^T \nabla_{\theta_n} g(\theta_n, \xi_n)}{\sqrt{S_n}} \middle| \mathscr{F}_n \right) \ne -\frac{\alpha_0}{\sqrt{S_n}} \left\| \nabla_{\theta_n} g(\theta_n) \right\|^2 \le 0.$$

In the following, we provide the proof outline of Theorem 3.

Step 1: This step is to ensure

$$-\mathbb{E}\left( \frac{\alpha_0 \nabla_{\theta_n} g(\theta_n, \xi_n)^T \nabla_{\theta_n} g(\theta_n, \xi_n)}{\sqrt{S_n}} \middle| \mathscr{F}_n \right) \le 0.$$

We are able to obtain the following equation

$$-\frac{\nabla_{\theta_n} g(\theta_n, \xi_n)^T \nabla_{\theta_n} g(\theta_n)}{\sqrt{S_n}} = \underbrace{\frac{1}{2\sqrt{S_n}} \left\| \frac{1}{\sqrt{M+1}} \nabla_{\theta_n} g(\theta_n, \xi_n) - \sqrt{M+1} \nabla_{\theta_n} g(\theta_n) \right\|^2}_{(K)}$$

$$\underbrace{-\frac{M+1}{2\sqrt{S_n}} \left\| \nabla_{\theta_n} g(\theta_n) \right\|^2 - \frac{1}{2(M+1)\sqrt{S_n}} \left\| \nabla_{\theta_n} g(\theta_n, \xi_n) \right\|^2}_{(L)}.$$

For $(K)$, due to $S_{n-1} \leq S_n$, it follows that

$$(K) = \frac{1}{2\sqrt{S_n}} \left\| \frac{1}{\sqrt{M+1}} \nabla_{\theta_n} g(\theta_n, \xi_n) - \sqrt{M+1} \nabla_{\theta_n} g(\theta_n) \right\|^2$$

$$\leq \frac{1}{2\sqrt{S_{n-1}}} \left\| \frac{1}{\sqrt{M+1}} \nabla_{\theta_n} g(\theta_n, \xi_n) - \sqrt{M+1} \nabla_{\theta_n} g(\theta_n) \right\|^2 .$$

Note that $S_{n-1}$ is conditional independent on $\nabla_{\theta_n} g(\theta_n, \xi_n)$, thus it holds that

$$\mathbb{E}\left( \frac{1}{\sqrt{S_{n-1}}} \left\| \frac{1}{\sqrt{M+1}} \nabla_{\theta_n} g(\theta_n, \xi_n) - \sqrt{M+1} \nabla_{\theta_n} g(\theta_n) \right\|^2 \bigg| \mathscr{F}_n \right)$$

$$= \frac{\alpha_0}{2} \frac{1}{\sqrt{S_{n-1}}} \left( \frac{1}{M+1} \mathbb{E}\left( \left\| \nabla_{\theta_n} g(\theta_n, \xi_n) \right\|^2 \big| \mathscr{F}_n \right) + (M-1) \left\| \nabla_{\theta_n} g(\theta_n) \right\|^2 \right).$$

Next we prove $(K)$ can be controlled by $(L)$ according to Lemmas 11 and 12. These two lemmas deal with the cases of $\|\nabla_{\theta_n} g(\theta_n)\| \leq a$ and $\|\nabla_{\theta_n} g(\theta_n)\| > a$, respectively. In these two lemmas, we introduce a constant $\varepsilon$ for the reason as stated in Step 2 in the following.

Step 2: Through Lemmas 11 and 12, we obtain that

$$\sum_{k=3}^{+\infty} \|\nabla_k g(\theta_k)\|^2 / S_{k-1}^{1/2+\varepsilon} < \zeta + \sum_{k=3}^{+\infty} \|\nabla_k g(\theta_k, \xi_k)\|^2 / S_k^{1+2\varepsilon}.$$

From Lemma 16, we obtain Lemma 13, stating that $\sum_{k=3}^{+\infty} \|\nabla_k g(\theta_k, \xi_k)\|^2 / S_k^{1+2\varepsilon} < +\infty$. Note that if we do not introduce $\varepsilon$, the term $\sum_{k=3}^{+\infty} \|\nabla_k g(\theta_k, \xi_k)\|^2 / S_k^{1+2\varepsilon}$ will become $\sum_{k=3}^{+\infty} \|\nabla_k g(\theta_k, \xi_k)\|^2 / S_k = O(\ln S_n)$, and this term may not be bounded.

Step 3: Lemma 13 ensures that $\nabla_{\theta_n} g(\theta_n)$ has a subsequence satisfying $\nabla_{\theta_{k_n}} g(\theta_{k_n}) \to 0$ a.s. By using the recursion formula $g(\theta_{n+1}) - g(\theta_n) \leq 1/S_n + P_n$, where $\sum_{n=1}^{+\infty} P_n < +\infty$ a.s., and Lemma 4, we obtain $\nabla_{\theta_n} g(\theta_n) \to 0$ a.s. Consequently, $\theta_n \to J^*$.

## C.2 PROOF OF LEMMA 10

For $\forall x \in \mathbb{R}^N$, we define function

$$g(t) = f\left( x + t \frac{x' - x}{\|x' - x\|} \right),$$

where $x'$ is a constant point such that $x' - x$ is parallel to $\nabla f(x)$. By taking the derivative, we obtain

$$g'(t) = \nabla_{x+t\frac{x'-x}{\|x'-x\|}} f\left( x + t \frac{x'-x}{\|x'-x\|} \right)^T \frac{x'-x}{\|x'-x\|}. \tag{78}$$

Through the Lipschitz condition of $\nabla f(x)$, we get $\forall t_1, t_2$

$$\left| g'(t_1) - g'(t_2) \right| = \left| \left( \nabla_{x+t\frac{x'-x}{\|x'-x\|}} f\left( x + t_1 \frac{x'-x}{\|x'-x\|} \right) - \nabla_{x+t\frac{x'-x}{\|x'-x\|}} f\left( x + t_2 \frac{x'-x}{\|x'-x\|} \right) \right)^T \frac{x'-x}{\|x'-x\|} \right|$$

$$\leq \left\| \nabla_{x+t\frac{x'-x}{\|x'-x\|}} f\left( x + t_1 \frac{x'-x}{\|x'-x\|} \right) - \nabla_{x+t\frac{x'-x}{\|x'-x\|}} f\left( x + t_2 \frac{x'-x}{\|x'-x\|} \right) \right\| \left\| \frac{x'-x}{\|x'-x\|} \right\| \leq c|t_1 - t_2|.$$

So $g'(t)$ satisfies the Lipschitz condition, and we have $\inf_{t \in \mathbb{R}} g(t) \geq \inf_{x \in \mathbb{R}^N} f(x) > -\infty$. Let $g^* = \inf x \in_{\mathbb{R}} g(x)$, then it holds that for $\forall t_0 \in \mathbb{R}$,

$$g(0) - g^* \geq g(0) - g(t_0). \tag{79}$$

By using the Newton-Leibniz's formula, we get that

$$g(0) - g(t_0) = \int_{t_0}^0 g'(\alpha) d\alpha = \int_{t_0}^0 \left( g'(\alpha) - g'(0) \right) d\alpha + \int_{t_0}^0 g'(0) d\alpha.$$

Through the *Lipschitz condition* of $g'$, we get that

$$g(0) - g(t_0) \geq \int_{t_0}^0 -c|\alpha - 0|d\alpha + \int_{t_0}^0 g'(0)d\alpha = \frac{1}{2c}\left(g'(0)\right)^2.$$

Then we take a special value of $t_0$. Let $t_0 = -g'(0)/c$, then we get

$$
\begin{aligned}
g(0) - g(t_0) &\geq -\int_{t_0}^0 c|\alpha|d\alpha + \int_{t_0}^0 g(0)dt = -\frac{c}{2}(0 - t_0)^2 + g'(0)(-t_0) \\
&= -\frac{1}{2c}\left(g'(0)\right)^2 + \frac{1}{c}\left(g'(0)\right)^2 = \frac{1}{2c}\left(g'(0)\right)^2.
\end{aligned}
\tag{80}
$$

Substituting equation 80 into equation 79, we get

$$g(0) - g^* \geq \frac{1}{2c}\left(g'(0)\right)^2.$$

Due to $g^* \geq f^*$ and $\left(g'(0)\right)^2 = \|\nabla f(x)\|^2$, it follows that

$$\left\|\nabla f(x)\right\|^2 \leq 2c\left(f(x) - f^*\right).$$

## C.3  PROOF OF LEMMA 11

First of all, it follows from Lemma 3 that

$$
\begin{aligned}
g(\theta_{n+1}) - g(\theta_n) &\leq \nabla_{\theta_n}g(\theta_n)^T(\theta_{n+1} - \theta_n) + \frac{c\alpha_0^2}{2}\frac{\left\|\nabla_{\theta_n}g(\theta_n,\xi_n)\right\|^2}{S_n} \\
&= -\frac{\alpha_0 \nabla_{\theta_n}g(\theta_n)^T \nabla_{\theta_n}g(\theta_n,\xi_n)}{\sqrt{S_n}} + \frac{c\alpha_0^2}{2}\frac{\left\|\nabla_{\theta_n}g(\theta_n,\xi_n)\right\|^2}{S_n},
\end{aligned}
\tag{81}
$$

where

$$S_n = \sum_{k=1}^n \left\|\nabla_{\theta_n}g(\theta_n,\xi_n)\right\|^2.$$

Note that

$$
\begin{aligned}
&\left\|\frac{1}{\sqrt{M+1}}\nabla_{\theta_n}g(\theta_n,\xi_n) - \sqrt{M+1}\nabla_{\theta_n}g(\theta_n)\right\|^2 \\
&= \frac{1}{M+1}\left\|\nabla_{\theta_n}g(\theta_n,\xi_n)\right\|^2 + (M+1)\left\|\nabla_{\theta_n}g(\theta_n)\right\|^2 - 2\nabla_{\theta_n}g(\theta_n)^T\nabla_{\theta_n}g(\theta_n,\xi_n),
\end{aligned}
\tag{82}
$$

where $M = M' + 2$ and $M'$ is defined in Assumption 5 3). Substitute equation 82 into equation 81, then we get that

$$
\begin{aligned}
&g(\theta_{n+1}) - g(\theta_n) \\
&\leq -\frac{\alpha_0}{2}\left(\frac{1}{M+1}\frac{\left\|\nabla_{\theta_n}g(\theta_n,\xi_n)\right\|^2}{\sqrt{S_n}} + (M+1)\frac{\left\|\nabla_{\theta_n}g(\theta_n)\right\|^2}{\sqrt{S_n}}\right) \\
&\quad + \frac{\alpha_0}{2}\frac{1}{\sqrt{S_n}}\left\|\frac{1}{\sqrt{M+1}}\nabla_{\theta_n}g(\theta_n,\xi_n) - \sqrt{M+1}\nabla_{\theta_n}g(\theta_n)\right\|^2 + \frac{c\alpha_0^2}{2}\frac{\left\|\nabla_{\theta_n}g(\theta_n,\xi_n)\right\|^2}{S_n}.
\end{aligned}
\tag{83}
$$

Due to $S_n \geq S_{n-1}$, it follows that

$$
\begin{aligned}
&\frac{\alpha_0}{2}\frac{1}{\sqrt{S_n}}\left\|\frac{1}{\sqrt{M+1}}\nabla_{\theta_n}g(\theta_n,\xi_n) - \sqrt{M+1}\nabla_{\theta_n}g(\theta_n)\right\|^2 \\
&\leq \frac{\alpha_0}{2}\frac{1}{\sqrt{S_{n-1}}}\left\|\frac{1}{\sqrt{M+1}}\nabla_{\theta_n}g(\theta_n,\xi_n) - \sqrt{M+1}\nabla_{\theta_n}g(\theta_n)\right\|^2.
\end{aligned}
\tag{84}
$$

Substitute equation 84 into equation 83, then we have

$$
\begin{aligned}
&g(\theta_{n+1}) - g(\theta_n) \\
&\leq -\frac{\alpha_0}{2}\left(\frac{1}{M+1}\frac{\left\|\nabla_{\theta_n}g(\theta_n,\xi_n)\right\|^2}{\sqrt{S_n}} + (M+1)\frac{\left\|\nabla_{\theta_n}g(\theta_n)\right\|^2}{\sqrt{S_n}}\right) \\
&+ \frac{\alpha_0}{2}\frac{1}{\sqrt{S_{n-1}}}\left\|\frac{1}{\sqrt{M+1}}\nabla_{\theta_n}g(\theta_n,\xi_n) - \sqrt{M+1}\nabla_{\theta_n}g(\theta_n)\right\|^2 + \frac{c\alpha_0^2}{2}\frac{\left\|\nabla_{\theta_n}g(\theta_n,\xi_n)\right\|^2}{S_n}.
\end{aligned}
\tag{85}
$$

Notice that

$$
\begin{aligned}
&\frac{\alpha_0}{2}\frac{1}{\sqrt{S_{n-1}}}\left\|\frac{1}{\sqrt{M+1}}\nabla_{\theta_n}g(\theta_n,\xi_n) - \sqrt{M+1}\nabla_{\theta_n}g(\theta_n)\right\|^2 \\
&= \frac{\alpha_0}{2}\frac{1}{\sqrt{S_{n-1}}}\left(\frac{1}{M+1}\left\|\nabla_{\theta_n}g(\theta_n,\xi_n)\right\|^2 + (M+1)\left\|\nabla_{\theta_n}g(\theta_n)\right\|^2 - 2\nabla_{\theta_n}g(\theta_n,\xi_n)^T\nabla_{\theta_n}g(\theta_n)\right) \\
&= \frac{\alpha_0}{2}\frac{1}{\sqrt{S_{n-1}}}\left(\frac{1}{M+1}\left\|\nabla_{\theta_n}g(\theta_n,\xi_n)\right\|^2 + (M+1)\left\|\nabla_{\theta_n}g(\theta_n)\right\|^2 - 2\left\|\nabla_{\theta_n}g(\theta_n)\right\|^2\right) \\
&+ \frac{\alpha_0}{\sqrt{S_{n-1}}}\nabla_{\theta_n}g(\theta_n)^T\left(\nabla_{\theta_n}g(\theta_n) - \nabla_{\theta_n}g(\theta_n,\xi_n)\right).
\end{aligned}
\tag{86}
$$

Substitute equation 86 into equation 85, and divide both sides of the inequality by $S_n^\varepsilon$ ($\varepsilon < \frac{1}{2}$), then we get

$$
\begin{aligned}
&\frac{g(\theta_{n+1})}{S_n^\varepsilon} - \frac{g(\theta_n)}{S_n^\varepsilon} \\
&\leq -\frac{\alpha_0}{2}\left(\frac{1}{M+1}\frac{\left\|\nabla_{\theta_n}g(\theta_n,\xi_n)\right\|^2}{S_n^{\frac{1}{2}+\varepsilon}} + (M+1)\frac{\left\|\nabla_{\theta_n}g(\theta_n)\right\|^2}{S_n^{\frac{1}{2}+\varepsilon}}\right) \\
&+ \frac{\alpha_0}{2}\frac{1}{S_{n-1}^{\frac{1}{2}+\varepsilon}}\left(\frac{1}{M+1}\left\|\nabla_{\theta_n}g(\theta_n,\xi_n)\right\|^2 + (M+1)\left\|\nabla_{\theta_n}g(\theta_n)\right\|^2 - 2\left\|\nabla_{\theta_n}g(\theta_n)\right\|^2\right) \\
&+ \frac{c\alpha_0^2}{2}\frac{\left\|\nabla_{\theta_n}g(\theta_n,\xi_n)\right\|^2}{S_n^{1+\varepsilon}} + \frac{\alpha_0}{S_{n-1}^{\frac{1}{2}+\varepsilon}}\nabla_{\theta_n}g(\theta_n)^T\left(\nabla_{\theta_n}g(\theta_n) - \nabla_{\theta_n}g(\theta_n,\xi_n)\right).
\end{aligned}
$$

Notice that $\frac{g(\theta_{n+1})}{S_n^\varepsilon} > \frac{g(\theta_{n+1})}{S_{n+1}^\varepsilon}$, then we obtain

$$
\begin{aligned}
&\frac{g(\theta_{n+1})}{S_{n+1}^\varepsilon} - \frac{g(\theta_n)}{S_n^\varepsilon} \\
&\leq -\frac{\alpha_0}{2}\left(\frac{1}{M+1}\frac{\left\|\nabla_{\theta_n}g(\theta_n,\xi_n)\right\|^2}{S_n^{\frac{1}{2}+\varepsilon}} + (M+1)\frac{\left\|\nabla_{\theta_n}g(\theta_n)\right\|^2}{S_n^{\frac{1}{2}+\varepsilon}}\right) \\
&+ \frac{\alpha_0}{2}\frac{1}{S_{n-1}^{\frac{1}{2}+\varepsilon}}\left(\frac{1}{M+1}\left\|\nabla_{\theta_n}g(\theta_n,\xi_n)\right\|^2 + (M+1)\left\|\nabla_{\theta_n}g(\theta_n)\right\|^2 - 2\left\|\nabla_{\theta_n}g(\theta_n)\right\|^2\right) \\
&+ \frac{c\alpha_0^2}{2}\frac{\left\|\nabla_{\theta_n}g(\theta_n,\xi_n)\right\|^2}{S_n^{1+\varepsilon}} + \frac{\alpha_0}{S_{n-1}^{\frac{1}{2}+\varepsilon}}\nabla_{\theta_n}g(\theta_n)^T\left(\nabla_{\theta_n}g(\theta_n) - \nabla_{\theta_n}g(\theta_n,\xi_n)\right).
\end{aligned}
\tag{87}
$$

Rearrange the above inequality, then it holds that

$$
\frac{g(\theta_{n+1})}{S_{n+1}^{\varepsilon}} - \frac{g(\theta_n)}{S_n^{\varepsilon}}
$$

$$
\leq -\frac{\alpha_0}{2}(M+1)\frac{\left\|\nabla_{\theta_n}g(\theta_n)\right\|^2}{S_n^{\frac{1}{2}+\varepsilon}}
$$

$$
+\frac{\alpha_0}{2}\left(\frac{1}{M+1}\frac{\left\|\nabla_{\theta_n}g(\theta_n,\xi_n)\right\|^2}{S_{n-1}^{\frac{1}{2}+\varepsilon}}+(M-1)\frac{\left\|\nabla_{\theta_n}g(\theta_n)\right\|^2}{S_{n-1}^{\frac{1}{2}+\varepsilon}}\right)+\frac{c\alpha_0^2}{2}\frac{\left\|\nabla_{\theta_n}g(\theta_n,\xi_n)\right\|^2}{S_n^{1+\varepsilon}}+X_n^{(\varepsilon)}
$$

$$
=\frac{\alpha_0}{2}(M+1)\left(\frac{\left\|\nabla_{\theta_{n-1}}g(\theta_{n-1})\right\|^2}{S_{n-1}^{\frac{1}{2}+\varepsilon}}-\frac{\left\|\nabla_{\theta_n}g(\theta_n)\right\|^2}{S_n^{\frac{1}{2}+\varepsilon}}\right)
$$

$$
+\frac{\alpha_0}{2}\left(\frac{1}{M+1}\frac{\left\|\nabla_{\theta_n}g(\theta_n,\xi_n)\right\|^2}{S_{n-1}^{\frac{1}{2}+\varepsilon}}+\frac{(M-1)\left\|\nabla_{\theta_n}g(\theta_n)\right\|^2}{S_{n-1}^{\frac{1}{2}+\varepsilon}}-\frac{(M+1)\left\|\nabla_{\theta_{n-1}}g(\theta_{n-1})\right\|^2}{S_{n-1}^{\frac{1}{2}+\varepsilon}}\right)
$$

$$
+\frac{c\alpha_0^2}{2}\frac{\left\|\nabla_{\theta_n}g(\theta_n,\xi_n)\right\|^2}{S_n^{1+\varepsilon}}+X_n^{(\varepsilon)}. \tag{88}
$$

$X_n^{(\varepsilon)}$ is defined as follow

$$
X_n^{(\varepsilon)}=\frac{\alpha_0}{S_{n-1}^{\frac{1}{2}+\varepsilon}}\nabla_{\theta_n}g(\theta_n)^T\left(\nabla_{\theta_n}g(\theta_n)-\nabla_{\theta_n}g(\theta_n,\xi_n)\right).
$$

Due to $\|\nabla_{\theta_n}g(\theta_n)\|^2 > a$, we have

$$
\mathbb{E}\left(\left\|\nabla_{\theta_n}g(\theta_n,\xi_n)\right\|^2\Big|\mathscr{F}_{n-1}\right)\leq M\left\|\nabla_{\theta_n}g(\theta_n)\right\|^2+a
$$

$$
<(M+1)\left\|\nabla_{\theta_n}g(\theta_n)\right\|^2, \tag{89}
$$

Moreover, using the Taylor formula, we obtain

$$
\left\|\nabla_{\theta_n}g(\theta_n)\right\|^2=\left\|\nabla_{\theta_{n-1}}g(\theta_{n-1})+\left(\nabla_{\theta_n}g(\theta_n)-\nabla_{\theta_{n-1}}g(\theta_{n-1})\right)\right\|^2
$$

$$
=\left\|\nabla_{\theta_{n-1}}g(\theta_{n-1})\right\|^2+2\nabla_{\theta_{n-1}}g(\theta_{n-1})^T\left(\nabla_{\theta_n}g(\theta_n)-\nabla_{\theta_{n-1}}g(\theta_{n-1})\right)
$$

$$
+\left\|\nabla_{\theta_n}g(\theta_n)-\nabla_{\theta_{n-1}}g(\theta_{n-1})\right\|^2\leq\left\|\nabla_{\theta_{n-1}}g(\theta_{n-1})\right\|^2
$$

$$
+2\left\|\nabla_{\theta_{n-1}}g(\theta_{n-1})\right\|\left\|\nabla_{\theta_n}g(\theta_n)-\nabla_{\theta_{n-1}}g(\theta_{n-1})\right\|+\left\|\nabla_{\theta_n}g(\theta_n)-\nabla_{\theta_{n-1}}g(\theta_{n-1})\right\|^2.
$$

Under Assumption 5 3), we get that

$$
\left\|\nabla_{\theta_n}g(\theta_n)\right\|^2\leq\left\|\nabla_{\theta_{n-1}}g(\theta_{n-1})\right\|^2
$$

$$
+2\left\|\nabla_{\theta_{n-1}}g(\theta_{n-1})\right\|\left\|\nabla_{\theta_n}g(\theta_n)-\nabla_{\theta_{n-1}}g(\theta_{n-1})\right\|+\left\|\nabla_{\theta_n}g(\theta_n)-\nabla_{\theta_{n-1}}g(\theta_{n-1})\right\|^2
$$

$$
\leq\left\|\nabla_{\theta_{n-1}}g(\theta_{n-1})\right\|^2+\frac{2\alpha_0 c}{\sqrt{S_{n-1}}}\left\|\nabla_{\theta_{n-1}}g(\theta_{n-1})\right\|\left\|\nabla_{\theta_{n-1}}g(\theta_{n-1},\xi_{n-1})\right\| \tag{90}
$$

$$
+c^2\alpha_0^2\frac{\left\|\nabla_{\theta_{n-1}}g(\theta_{n-1},\xi_{n-1})\right\|^2}{S_{n-1}}.
$$

From inequality $2a^Tb\leq\lambda\|a\|^2+\frac{1}{\lambda}\|b\|^2$ $(\lambda>0)$, it follows that

$$
(M-1)\left\|\nabla_{\theta_n}g(\theta_n)\right\|^2+\left\|\nabla_{\theta_n}g(\theta_n)\right\|^2
$$

$$
\leq(M+1)\left\|\nabla_{\theta_{n-1}}g(\theta_{n-1})\right\|^2-\frac{M-1}{4M-3}\left\|\nabla_{\theta_n}g(\theta_n)\right\|^2+4M^2\alpha_0^2c^2\frac{\left\|\nabla_{\theta_{n-1}}g(\theta_{n-1},\xi_{n-1})\right\|^2}{S_{n-1}}. \tag{91}
$$

By substituting equation 89 into equation 91, we have

$$
(M-1)\left\|\nabla_{\theta_n}g(\theta_n)\right\|^2+\frac{1}{M+1}\mathbb{E}\left(\left\|\nabla_{\theta_n}g(\theta_n,\xi_n)\right\|^2\Big|\mathscr{F}_{n-1}\right)
$$

$$
\leq(M+1)\left\|\nabla_{\theta_{n-1}}g(\theta_{n-1})\right\|^2-\frac{M-1}{4M-3}\left\|\nabla_{\theta_n}g(\theta_n)\right\|^2+4M^2\alpha_0^2c^2\frac{\left\|\nabla_{\theta_{n-1}}g(\theta_{n-1},\xi_{n-1})\right\|^2}{S_{n-1}}, \tag{92}
$$

Divide both sides of equation 92 by $S_{n-1}^{\frac{1}{2}+\varepsilon}$, and notice $\frac{M-1}{4M-3} > \frac{1}{5}$ from $M > 2$, then it holds that

$$
\begin{aligned}
&\frac{1}{M+1}\frac{\left\|\nabla_{\theta_n}g(\theta_n,\xi_n)\right\|^2}{S_{n-1}^{\frac{1}{2}+\varepsilon}} + \frac{\left\|\nabla_{\theta_n}g(\theta_n)\right\|^2}{S_{n-1}^{\frac{1}{2}+\varepsilon}} \\
&\leq (M+1)\frac{\left\|\nabla_{\theta_{n-1}}g(\theta_{n-1})\right\|^2}{S_{n-1}^{\frac{1}{2}+\varepsilon}} - \frac{1}{5}\left\|\nabla_{\theta_n}g(\theta_n)\right\|^2 + 4M^2\alpha_0^2 c^2\frac{\left\|\nabla_{\theta_{n-1}}g(\theta_{n-1},\xi_{n-1})\right\|^2}{S_{n-1}} \\
&+ \frac{2}{\alpha_0}Y_n^{(\varepsilon)},
\end{aligned}
\tag{93}
$$

where

$$
Y_n^{(\varepsilon)} = \frac{\alpha_0}{2}\left(\frac{1}{M+1}\frac{\mathbb{E}\left(\left\|\nabla_{\theta_n}g(\theta_n,\xi_n)\right\|^2\,\Big|\,\mathscr{F}_{n-1}\right)}{S_{n-1}^{\frac{1}{2}+\varepsilon}} - \frac{1}{M+1}\frac{\left\|\nabla_{\theta_n}g(\theta_n,\xi_n)\right\|^2}{S_{n-1}^{\frac{1}{2}+\varepsilon}}\right).
$$

Making some simple transformations on equation 93 leads to

$$
\begin{aligned}
&\frac{\alpha_0}{2}\left(\frac{1}{M+1}\frac{\left\|\nabla_{\theta_n}g(\theta_n,\xi_n)\right\|^2}{S_{n-1}^{\frac{1}{2}+\varepsilon}} + \frac{(M-1)\left\|\nabla_{\theta_n}g(\theta_n)\right\|^2}{S_{n-1}^{\frac{1}{2}+\varepsilon}} - \frac{(M+1)\left\|\nabla_{\theta_{n-1}}g(\theta_{n-1})\right\|^2}{S_{n-1}^{\frac{1}{2}+\varepsilon}}\right) \\
&\leq -\frac{\alpha_0}{10}\frac{\left\|\nabla_{\theta_n}g(\theta_n)\right\|^2}{S_{n-1}^{\frac{1}{2}+\varepsilon}} + 4M^2\alpha_0^3 c^2\frac{\left\|\nabla_{\theta_{n-1}}g(\theta_{n-1},\xi_{n-1})\right\|^2}{S_{n-1}^{\frac{3}{2}+\varepsilon}} + Y_n^{(\varepsilon)}.
\end{aligned}
\tag{94}
$$

Substitute equation 94 into equation 88, then we get

$$
\begin{aligned}
&\frac{g(\theta_{n+1})}{S_{n+1}^\varepsilon} - \frac{g(\theta_n)}{S_n^\varepsilon} \\
&\leq \frac{\alpha_0}{2}(M+1)\left(\frac{\left\|\nabla_{\theta_{n-1}}g(\theta_{n-1})\right\|^2}{S_{n-1}^{\frac{1}{2}+\varepsilon}} - \frac{\left\|\nabla_{\theta_n}g(\theta_n)\right\|^2}{S_n^{\frac{1}{2}+\varepsilon}}\right) \\
&- \frac{\alpha_0}{10}\frac{\left\|\nabla_{\theta_n}g(\theta_n)\right\|^2}{S_{n-1}^{\frac{1}{2}+\varepsilon}} + 4M^2\alpha_0^3 c^2\frac{\left\|\nabla_{\theta_{n-1}}g(\theta_{n-1},\xi_{n-1})\right\|^2}{S_{n-1}^{\frac{3}{2}+\varepsilon}} + \frac{c\alpha_0^2}{2}\frac{\left\|\nabla_{\theta_n}g(\theta_n,\xi_n)\right\|^2}{S_n^{1+\varepsilon}} \\
&+ X_n^{(\varepsilon)} + Y_n^{(\varepsilon)}.
\end{aligned}
\tag{95}
$$

It follows that

$$
\begin{aligned}
-\frac{\alpha_0}{10}\frac{\left\|\nabla_{\theta_n}g(\theta_n)\right\|^2}{S_{n-1}^{\frac{1}{2}+\varepsilon}} &= -\frac{\alpha_0}{20}\frac{\left\|\nabla_{\theta_n}g(\theta_n)\right\|^2}{S_{n-1}^{\frac{1}{2}+\varepsilon}} - \frac{\alpha_0}{20}\frac{\left\|\nabla_{\theta_n}g(\theta_n)\right\|^2}{S_{n-1}^{\frac{1}{2}+\varepsilon}} \\
&= -\frac{\alpha_0}{20}\frac{\left\|\nabla_{\theta_n}g(\theta_n)\right\|^2}{S_{n-1}^{\frac{1}{2}+\varepsilon}} + \frac{\alpha_0}{20}\left(\frac{\left\|\nabla_{\theta_{n-1}}g(\theta_{n-1})\right\|^2}{S_{n-2}^{\frac{1}{2}+\varepsilon}} - \frac{\left\|\nabla_{\theta_n}g(\theta_n)\right\|^2}{S_{n-1}^{\frac{1}{2}+\varepsilon}}\right) \\
&- \frac{\alpha_0}{20}\frac{\left\|\nabla_{\theta_{n-1}}g(\theta_{n-1})\right\|^2}{S_{n-2}^{\frac{1}{2}+\varepsilon}} \\
&\leq -\frac{\alpha_0}{20}\frac{\left\|\nabla_{\theta_n}g(\theta_n)\right\|^2}{S_{n-1}^{\frac{1}{2}+\varepsilon}} + \frac{\alpha_0}{20}\left(\frac{\left\|\nabla_{\theta_{n-1}}g(\theta_{n-1})\right\|^2}{S_{n-2}^{\frac{1}{2}+\varepsilon}} - \frac{\left\|\nabla_{\theta_n}g(\theta_n)\right\|^2}{S_{n-1}^{\frac{1}{2}+\varepsilon}}\right).
\end{aligned}
\tag{96}
$$

Substituting equation 96 into equation 95 yields

$$
\begin{aligned}
\frac{g(\theta_{n+1})}{S_{n+1}^{\varepsilon}} &- \frac{g(\theta_n)}{S_n^{\varepsilon}} \\
&\leq \frac{\alpha_0}{2}(M+1)\left( \frac{\left\|\nabla_{\theta_{n-1}}g(\theta_{n-1})\right\|^2}{S_{n-1}^{\frac{1}{2}+\varepsilon}} - \frac{\left\|\nabla_{\theta_n}g(\theta_n)\right\|^2}{S_n^{\frac{1}{2}+\varepsilon}} \right) \\
&- \frac{\alpha_0}{20}\frac{\left\|\nabla_{\theta_n}g(\theta_n)\right\|^2}{S_{n-1}^{\frac{1}{2}+\varepsilon}} + \frac{\alpha_0}{20}\left( \frac{\left\|\nabla_{\theta_{n-1}}g(\theta_{n-1})\right\|^2}{S_{n-2}^{\frac{1}{2}+\varepsilon}} - \frac{\left\|\nabla_{\theta_n}g(\theta_n)\right\|^2}{S_{n-1}^{\frac{1}{2}+\varepsilon}} \right) \\
&+ 4M^2\alpha_0^3c^2\frac{\left\|\nabla_{\theta_{n-1}}g(\theta_{n-1},\xi_{n-1})\right\|^2}{S_{n-1}^{\frac{3}{2}+\varepsilon}} + \frac{c\alpha_0^2}{2}\frac{\left\|\nabla_{\theta_n}g(\theta_n,\xi_n)\right\|^2}{S_n^{1+\varepsilon}} + X_n^{(\varepsilon)} + Y_n^{(\varepsilon)}.
\end{aligned}
\tag{97}
$$

## C.4 PROOF OF LEMMA 12

First of all, dividing both sides of equation 81 by $S_n^{\varepsilon}$ yields

$$
\frac{g(\theta_{n+1})}{S_n^{\varepsilon}} - \frac{g(\theta_n)}{S_n^{\varepsilon}} \leq -\frac{\alpha_0\nabla_{\theta_n}g(\theta_n)^T\nabla_{\theta_n}g(\theta_n,\xi_n)}{S_n^{\frac{1}{2}+\varepsilon}} + \frac{c\alpha_0^2}{2}\frac{\left\|\nabla_{\theta_n}g(\theta_n,\xi_n)\right\|^2}{S_n^{1+\varepsilon}}.
\tag{98}
$$

Due to $S_{n+1} \geq S_n$, it holds that

$$
\frac{g(\theta_{n+1})}{S_{n+1}^{\varepsilon}} - \frac{g(\theta_n)}{S_n^{\varepsilon}} \leq -\frac{\alpha_0\nabla_{\theta_n}g(\theta_n)^T\nabla_{\theta_n}g(\theta_n,\xi_n)}{S_n^{\frac{1}{2}+\varepsilon}} + \frac{c\alpha_0^2}{2}\frac{\left\|\nabla_{\theta_n}g(\theta_n,\xi_n)\right\|^2}{S_n^{1+\varepsilon}}.
\tag{99}
$$

Then we make some transformations to obtain that

$$
\begin{aligned}
&-\frac{\alpha_0\nabla_{\theta_n}g(\theta_n)^T\nabla_{\theta_n}g(\theta_n,\xi_n)}{S_n^{\frac{1}{2}+\varepsilon}} \\
&= -\frac{\alpha_0\nabla_{\theta_n}g(\theta_n)^T\nabla_{\theta_n}g(\theta_n,\xi_n)}{S_{n-1}^{\frac{1}{2}+\varepsilon}} + \alpha_0\nabla_{\theta_n}g(\theta_n)^T\nabla_{\theta_n}g(\theta_n,\xi_n)\left( \frac{1}{S_{n-1}^{\frac{1}{2}+\varepsilon}} - \frac{1}{S_n^{\frac{1}{2}+\varepsilon}} \right) \\
&\leq -\frac{\alpha_0\nabla_{\theta_n}g(\theta_n)^T\nabla_{\theta_n}g(\theta_n,\xi_n)}{S_{n-1}^{\frac{1}{2}+\varepsilon}} + \alpha_0\left( \frac{(M+1)a}{2} + \frac{1}{2(M+1)a}\left\|\nabla_{\theta_n}g(\theta_n)\right\|^2\left\|\nabla_{\theta_n}g(\theta_n,\xi_n)\right\|^2 \right) \\
&\quad\left( \frac{1}{S_{n-1}^{\frac{1}{2}+\varepsilon}} - \frac{1}{S_n^{\frac{1}{2}+\varepsilon}} \right) \\
&\leq -\frac{\alpha_0\left\|\nabla_{\theta_n}g(\theta_n)\right\|^2}{S_{n-1}^{\frac{1}{2}+\varepsilon}} + \frac{(M+1)\alpha_0a}{2}\left( \frac{1}{S_{n-1}^{\frac{1}{2}+\varepsilon}} - \frac{1}{S_n^{\frac{1}{2}+\varepsilon}} \right) \\
&\quad+ \left( \frac{\alpha_0}{2a(M+1)}\left\|\nabla_{\theta_n}g(\theta_n)\right\|^2\mathbb{E}\left( \left\|\nabla_{\theta_n}g(\theta_n,\xi_n)\right\|^2 \Big| \mathscr{F}_{n-1} \right) \right)\frac{1}{S_{n-1}^{\frac{1}{2}+\varepsilon}} + A_n^{(\varepsilon)} + B_n^{(\varepsilon)}.
\end{aligned}
\tag{100}
$$

where

$$
\begin{aligned}
A_n^{(\varepsilon)} &= \frac{\alpha_0}{S_{n-1}^{\frac{1}{2}+\varepsilon}}\left( \left\|\nabla_{\theta_n}g(\theta_n)\right\|^2 - \nabla_{\theta_n}g(\theta_n)^T\nabla_{\theta_n}g(\theta_n,\xi_n) \right) \\
B_n^{(\varepsilon)} &= \frac{\alpha_0\left\|\nabla_{\theta_n}g(\theta_n)\right\|^2}{2a(M+1)S_{n-1}^{\frac{1}{2}+\varepsilon}}\left( \left\|\nabla_{\theta_n}g(\theta_n,\xi_n)\right\|^2 - \mathbb{E}\left( \left\|\nabla_{\theta_n}g(\theta_n,\xi_n)\right\|^2 \Big| \mathscr{F}_{n-1} \right) \right).
\end{aligned}
\tag{101}
$$

Due to $\left\|\nabla_{\theta_n}g(\theta_n)\right\|^2 \leq a$, we get

$$
\mathbb{E}\left( \left\|\nabla_{\theta_n}g(\theta_n,\xi_n)\right\|^2 \Big| \mathscr{F}_{n-1} \right) \leq M\left\|\nabla_{\theta_n}g(\theta_n)\right\|^2 + a \leq (M+1)a.
\tag{102}
$$

Substitute it into equation 100, then we get

$$
\begin{aligned}
\frac{g(\theta_{n+1})}{S_{n+1}^{\varepsilon}} - \frac{g(\theta_n)}{S_n^{\varepsilon}} &\leq -\frac{\alpha_0\left\|\nabla_{\theta_n}g(\theta_n)\right\|^2}{S_{n-1}^{\frac{1}{2}+\varepsilon}} + \frac{\alpha_0 a(M+1)}{2}\left(\frac{1}{S_{n-1}^{\frac{1}{2}+\varepsilon}} - \frac{1}{S_n^{\frac{1}{2}+\varepsilon}}\right) \\
&+ \frac{\alpha_0\left\|\nabla_{\theta_n}g(\theta_n)\right\|^2}{2S_{n-1}^{\frac{1}{2}+\varepsilon}} + \frac{c\alpha_0^2}{2}\frac{\left\|\nabla_{\theta_n}g(\theta_n,\xi_n)\right\|^2}{S_n^{1+\varepsilon}} + A_n^{(\varepsilon)} + B_n^{(\varepsilon)} \\
&= -\frac{\alpha_0\left\|\nabla_{\theta_n}g(\theta_n)\right\|^2}{2S_{n-1}^{\frac{1}{2}+\varepsilon}} + \frac{\alpha_0 a(M+1)}{2}\left(\frac{1}{S_{n-1}^{\frac{1}{2}+\varepsilon}} - \frac{1}{S_n^{\frac{1}{2}+\varepsilon}}\right) \\
&+ \frac{c\alpha_0^2}{2}\frac{\left\|\nabla_{\theta_n}g(\theta_n,\xi_n)\right\|^2}{S_n^{1+\varepsilon}} + A_n^{(\varepsilon)} + B_n^{(\varepsilon)}.
\end{aligned}
\tag{103}
$$

We make some transformations on $-\dfrac{\alpha_0\left\|\nabla_{\theta_n}g(\theta_n)\right\|^2}{2S_{n-1}^{\frac{1}{2}+\varepsilon}}$ to obtain that

$$
\begin{aligned}
-\frac{\alpha_0\left\|\nabla_{\theta_n}g(\theta_n)\right\|^2}{2S_{n-1}^{\frac{1}{2}+\varepsilon}} &\leq -\frac{\alpha_0\left\|\nabla_{\theta_n}g(\theta_n)\right\|^2}{20S_{n-1}^{\frac{1}{2}+\varepsilon}} - \frac{\alpha_0\left\|\nabla_{\theta_n}g(\theta_n)\right\|^2}{20S_{n-1}^{\frac{1}{2}+\varepsilon}} \\
&= -\frac{\alpha_0\left\|\nabla_{\theta_n}g(\theta_n)\right\|^2}{20S_{n-1}^{\frac{1}{2}+\varepsilon}} - \frac{\alpha_0}{20}\frac{\left\|\nabla_{\theta_{n-1}}g(\theta_{n-1})\right\|^2}{S_{n-2}^{\frac{1}{2}+\varepsilon}} + \frac{\alpha_0}{20}\left(\frac{\left\|\nabla_{\theta_{n-1}}g(\theta_{n-1})\right\|^2}{S_{n-2}^{\frac{1}{2}+\varepsilon}} - \frac{\left\|\nabla_{\theta_n}g(\theta_n)\right\|^2}{S_{n-1}^{\frac{1}{2}+\varepsilon}}\right).
\end{aligned}
\tag{104}
$$

Then we use inequality $2a^Tb \leq \lambda\|a\|^2 + \frac{1}{\lambda}\|b\|^2$ $(\lambda > 0)$ on equation 90 to get

$$
\begin{aligned}
\left\|\nabla_{\theta_n}g(\theta_n)\right\|^2 - \left\|\nabla_{\theta_{n-1}}g(\theta_{n-1})\right\|^2 &\leq \frac{\left\|\nabla_{\theta_{n-1}}g(\theta_{n-1})\right\|^2}{10(M+1)} \\
&+ \frac{10\alpha_0^2c^2(M+1)}{S_{n-1}}\left\|\nabla_{\theta_{n-1}}g(\theta_{n-1},\xi_{n-1})\right\|^2 + \frac{\alpha_0^2c^2}{S_{n-1}}\left\|\nabla_{\theta_{n-1}}g(\theta_{n-1},\xi_{n-1})\right\|^2.
\end{aligned}
\tag{105}
$$

Divide both sides of equation 105 by $S_{n-1}^{\frac{1}{2}+\varepsilon}$ and notice $S_{n-2} \leq S_{n-1} \leq S_n$, then we have

$$
\begin{aligned}
\frac{\left\|\nabla_{\theta_n}g(\theta_n)\right\|^2}{S_n^{\frac{1}{2}+\varepsilon}} - \frac{\left\|\nabla_{\theta_{n-1}}g(\theta_{n-1})\right\|^2}{S_{n-1}^{\frac{1}{2}+\varepsilon}} &\leq \frac{1}{M+1}\frac{\left\|\nabla_{\theta_{n-1}}g(\theta_{n-1})\right\|^2}{10S_{n-2}^{\frac{1}{2}+\varepsilon}} \\
&+ \frac{10\alpha_0^2c^2(M+1)}{S_{n-1}^{1+\varepsilon}}\left\|\nabla_{\theta_{n-1}}g(\theta_{n-1},\xi_{n-1})\right\|^2 + \frac{\alpha_0^2c^2}{S_{n-1}^{\frac{3}{2}+\varepsilon}}\left\|\nabla_{\theta_{n-1}}g(\theta_{n-1},\xi_{n-1})\right\|^2.
\end{aligned}
\tag{106}
$$

Then we calculate $\frac{\alpha_0}{2}(M+1)equation\ 106 + equation\ 104$

$$
\begin{aligned}
&-\frac{\alpha_0\left\|\nabla_{\theta_n}g(\theta_n)\right\|^2}{2S_{n-1}^{\frac{1}{2}+\varepsilon}} + \frac{\alpha_0(M+1)}{2}\left(\frac{\left\|\nabla_{\theta_n}g(\theta_n)\right\|^2}{S_n^{\frac{1}{2}+\varepsilon}} - \frac{\left\|\nabla_{\theta_{n-1}}g(\theta_{n-1})\right\|^2}{S_{n-1}^{\frac{1}{2}+\varepsilon}}\right) \\
&\leq -\frac{\alpha_0\left\|\nabla_{\theta_n}g(\theta_n)\right\|^2}{20S_{n-1}^{\frac{1}{2}+\varepsilon}} + 5\alpha_0^3c^2(M+1)^2\frac{\left\|\nabla_{\theta_{n-1}}g(\theta_{n-1},\xi_{n-1})\right\|^2}{S_{n-1}^{1+\varepsilon}} \\
&+ \frac{(M+1)\alpha_0^3c^2}{2S_{n-1}^{\frac{3}{2}+\varepsilon}}\left\|\nabla_{\theta_{n-1}}g(\theta_{n-1},\xi_{n-1})\right\|^2 + \frac{\alpha_0}{20}\left(\frac{\left\|\nabla_{\theta_{n-1}}g(\theta_{n-1})\right\|^2}{S_{n-2}^{\frac{1}{2}+\varepsilon}} - \frac{\left\|\nabla_{\theta_n}g(\theta_n)\right\|^2}{S_{n-1}^{\frac{1}{2}+\varepsilon}}\right).
\end{aligned}
$$

Move $\frac{\alpha_0(M+1)}{2}\left(\frac{\left\|\nabla_{\theta_n}g(\theta_n)\right\|^2}{S_n^{\frac{1}{2}+\varepsilon}} - \frac{\left\|\nabla_{\theta_{n-1}}g(\theta_{n-1})\right\|^2}{S_{n-1}^{\frac{1}{2}+\varepsilon}}\right)$ to the right-hand side of the above inequality, then we have

$$
\begin{aligned}
-\frac{\alpha_0\left\|\nabla_{\theta_n}g(\theta_n)\right\|^2}{2S_{n-1}^{\frac{1}{2}+\varepsilon}} \leq &-\frac{\alpha_0\left\|\nabla_{\theta_n}g(\theta_n)\right\|^2}{20S_{n-1}^{\frac{1}{2}+\varepsilon}} + 5\alpha_0^3 c^2(M+1)^2\frac{\left\|\nabla_{\theta_{n-1}}g(\theta_{n-1},\xi_{n-1})\right\|^2}{S_{n-1}^{1+\varepsilon}} \\
&+\frac{(M+1)\alpha_0^3 c^2}{2S_{n-1}^{\frac{3}{2}+\varepsilon}}\left\|\nabla_{\theta_{n-1}}g(\theta_{n-1},\xi_{n-1})\right\|^2 + \frac{\alpha_0}{20}\left(\frac{\left\|\nabla_{\theta_{n-1}}g(\theta_{n-1})\right\|^2}{S_{n-2}^{\frac{1}{2}+\varepsilon}} - \frac{\left\|\nabla_{\theta_n}g(\theta_n)\right\|^2}{S_{n-1}^{\frac{1}{2}+\varepsilon}}\right) \\
&+\frac{\alpha_0(M+1)}{2}\left(\frac{\left\|\nabla_{\theta_{n-1}}g(\theta_{n-1})\right\|^2}{S_{n-1}^{\frac{1}{2}+\varepsilon}} - \frac{\left\|\nabla_{\theta_n}g(\theta_n)\right\|^2}{S_n^{\frac{1}{2}+\varepsilon}}\right).
\end{aligned}
\tag{107}
$$

Substitute equation 107 into equation 103, then we have

$$
\begin{aligned}
\frac{g(\theta_{n+1})}{S_{n+1}^\varepsilon} - \frac{g(\theta_n)}{S_n^\varepsilon} \\
\leq &-\frac{\alpha_0\left\|\nabla_{\theta_n}g(\theta_n)\right\|^2}{20S_{n-1}^{\frac{1}{2}+\varepsilon}} + 5\alpha_0^3 c^2(M+1)^2\frac{\left\|\nabla_{\theta_{n-1}}g(\theta_{n-1},\xi_{n-1})\right\|^2}{S_{n-1}^{1+\varepsilon}} \\
&+\frac{(M+1)\alpha_0^3 c^2}{2S_{n-1}^{\frac{3}{2}+\varepsilon}}\left\|\nabla_{\theta_{n-1}}g(\theta_{n-1},\xi_{n-1})\right\|^2 + \frac{\alpha_0}{20}\left(\frac{\left\|\nabla_{\theta_{n-1}}g(\theta_{n-1})\right\|^2}{S_{n-2}^{\frac{1}{2}+\varepsilon}} - \frac{\left\|\nabla_{\theta_n}g(\theta_n)\right\|^2}{S_{n-1}^{\frac{1}{2}+\varepsilon}}\right) \\
&+\frac{\alpha_0(M+1)}{2}\left(\frac{\left\|\nabla_{\theta_{n-1}}g(\theta_{n-1})\right\|^2}{S_{n-1}^{\frac{1}{2}+\varepsilon}} - \frac{\left\|\nabla_{\theta_n}g(\theta_n)\right\|^2}{S_n^{\frac{1}{2}+\varepsilon}}\right) + \frac{\alpha_0 a(M+1)}{2}\left(\frac{1}{S_{n-1}^{\frac{1}{2}+\varepsilon}} - \frac{1}{S_n^{\frac{1}{2}+\varepsilon}}\right) \\
&+\frac{c\alpha_0^2}{2}\frac{\left\|\nabla_{\theta_n}g(\theta_n,\xi_n)\right\|^2}{S_n^{1+\varepsilon}} + A_n^{(\varepsilon)} + B_n^{(\varepsilon)}.
\end{aligned}
\tag{108}
$$

## C.5 PROOF OF LEMMA 13

First of all, it holds that

$$
\begin{aligned}
\frac{g(\theta_{n+1})}{S_{n+1}^\varepsilon} - \frac{g(\theta_n)}{S_n^\varepsilon} \\
= I\left(\left\|\nabla_{\theta_n}g(\theta_n)\right\|^2 \leq a\right)\left(\frac{g(\theta_{n+1})}{S_{n+1}^\varepsilon} - \frac{g(\theta_n)}{S_n^\varepsilon}\right) + I\left(\left\|\nabla_{\theta_n}g(\theta_n)\right\|^2 > a\right)\left(\frac{g(\theta_{n+1})}{S_{n+1}^\varepsilon} - \frac{g(\theta_n)}{S_n^\varepsilon}\right).
\end{aligned}
$$

$I\left(\left\|\nabla_{\theta_n}g(\theta_n)\right\|^2 \leq a\right) \in \mathscr{F}_{n-1}$ is the indicator function such that

$$
I\left(\left\|\nabla_{\theta_n}g(\theta_n)\right\|^2 \leq a\right) = \begin{cases} 1 & \left\|\nabla_{\theta_n}g(\theta_n)\right\|^2 \leq a \\ 0 & \left\|\nabla_{\theta_n}g(\theta_n)\right\|^2 > a \end{cases}
$$

For convenient, we abbreviate $I\left(\left\|\nabla_{\theta_n}g(\theta_n)\right\|^2 \leq a\right)$ as $I_n^{\leq a}$ and $I\left(\left\|\nabla_{\theta_n}g(\theta_n)\right\|^2 > a\right)$ as $I_n^{>a}$ in the following.

Through Lemma 12, we get that

$$
\begin{aligned}
&I_n^{\leq a}\left(\frac{g(\theta_{n+1})}{S_{n+1}^{\varepsilon}} - \frac{g(\theta_n)}{S_n^{\varepsilon}}\right)\\
&\leq -I_n^{\leq a}\frac{\alpha_0\left\|\nabla_{\theta_n}g(\theta_n)\right\|^2}{20S_{n-1}^{\frac{1}{2}+\varepsilon}} + 5\alpha_0^3 c^2(M+1)^2 I_n^{\leq a}\frac{\left\|\nabla_{\theta_{n-1}}g(\theta_{n-1},\xi_{n-1})\right\|^2}{S_{n-1}^{1+\varepsilon}}\\
&\quad + I_n^{\leq a}\frac{(M+1)\alpha_0^3 c^2}{2S_{n-1}^{\frac{3}{2}+\varepsilon}}\left\|\nabla_{\theta_{n-1}}g(\theta_{n-1},\xi_{n-1})\right\|^2 + \frac{\alpha_0}{20}I_n^{\leq a}\left(\frac{\left\|\nabla_{\theta_{n-1}}g(\theta_{n-1})\right\|^2}{S_{n-2}^{\frac{1}{2}+\varepsilon}} - \frac{\left\|\nabla_{\theta_n}g(\theta_n)\right\|^2}{S_{n-1}^{\frac{1}{2}+\varepsilon}}\right)\\
&\quad + \frac{\alpha_0(M+1)}{2}I_n^{\leq a}\left(\frac{\left\|\nabla_{\theta_{n-1}}g(\theta_{n-1})\right\|^2}{S_{n-1}^{\frac{1}{2}+\varepsilon}} - \frac{\left\|\nabla_{\theta_n}g(\theta_n)\right\|^2}{S_n^{\frac{1}{2}+\varepsilon}}\right) + \frac{\alpha_0 a(M+1)}{2}I_n^{\leq a}\left(\frac{1}{S_{n-1}^{\frac{1}{2}+\varepsilon}} - \frac{1}{S_n^{\frac{1}{2}+\varepsilon}}\right)\\
&\quad + \frac{c\alpha_0^2}{2}I_n^{\leq a}\frac{\left\|\nabla_{\theta_n}g(\theta_n,\xi_n)\right\|^2}{S_n^{1+\varepsilon}} + I_n^{\leq a}A_n^{(\varepsilon)} + I_n^{\leq a}B_n^{(\varepsilon)}.
\end{aligned}
\tag{109}
$$

Through Lemma 11, we get

$$
\begin{aligned}
&I_n^{>a}\left(\frac{g(\theta_{n+1})}{S_{n+1}^{\varepsilon}} - \frac{g(\theta_n)}{S_n^{\varepsilon}}\right)\\
&\leq -I_n^{>a}\frac{\alpha_0}{20}\frac{\left\|\nabla_{\theta_n}g(\theta_n)\right\|^2}{S_{n-1}^{\frac{1}{2}+\varepsilon}} + \frac{\alpha_0(M+1)}{2}\cdot I_n^{>a}\left(\frac{\left\|\nabla_{\theta_{n-1}}g(\theta_{n-1})\right\|^2}{S_{n-1}^{\frac{1}{2}+\varepsilon}} - \frac{\left\|\nabla_{\theta_n}g(\theta_n)\right\|^2}{S_n^{\frac{1}{2}+\varepsilon}}\right)\\
&\quad + \frac{\alpha_0}{20}I_n^{>a}\left(\frac{\left\|\nabla_{\theta_{n-1}}g(\theta_{n-1})\right\|^2}{S_{n-2}^{\frac{1}{2}+\varepsilon}} - \frac{\left\|\nabla_{\theta_n}g(\theta_n)\right\|^2}{S_{n-1}^{\frac{1}{2}+\varepsilon}}\right)\\
&\quad 4M^2\alpha_0^3 c^2 I_n^{>a}\frac{\left\|\nabla_{\theta_{n-1}}g(\theta_{n-1}),\xi_{n-1}\right\|^2}{S_{n-1}^{\frac{3}{2}+\varepsilon}} + \frac{c\alpha_0^2}{2}I_n^{>a}\frac{\left\|\nabla_{\theta_n}g(\theta_n,\xi_n)\right\|^2}{S_n^{1+\varepsilon}}\\
&\quad + I_n^{>a}X_n^{(\varepsilon)} + I_n^{>a}Y_n^{(\varepsilon)}.
\end{aligned}
\tag{110}
$$

Calculate *equation* 109 + *equation* 110, then it holds that

$$
\begin{aligned}
&I_n^{\leq a}\left(\frac{g(\theta_{n+1})}{S_{n+1}^{\varepsilon}} - \frac{g(\theta_n)}{S_n^{\varepsilon}}\right) + I_n^{>a}\left(\frac{g(\theta_{n+1})}{S_{n+1}^{\varepsilon}} - \frac{g(\theta_n)}{S_n^{\varepsilon}}\right)\\
&\leq \frac{\alpha_0}{20}(I_n^{\leq a}+I_n^{>a})\left(\frac{\left\|\nabla_{\theta_{n-1}}g(\theta_{n-1})\right\|^2}{S_{n-2}^{\frac{1}{2}+\varepsilon}} - \frac{\left\|\nabla_{\theta_n}g(\theta_n)\right\|^2}{S_{n-1}^{\frac{1}{2}+\varepsilon}}\right)\\
&\quad + \frac{\alpha_0}{2}(M+1)(I_n^{\leq a}+I_n^{>a})\left(\frac{\left\|\nabla_{\theta_{n-1}}g(\theta_{n-1})\right\|^2}{S_{n-1}^{\frac{1}{2}+\varepsilon}} - \frac{\left\|\nabla_{\theta_n}g(\theta_n)\right\|^2}{S_n^{\frac{1}{2}+\varepsilon}}\right)\\
&\quad - \frac{\alpha_0}{20}(I_n^{\leq a}+I_n^{>a})\frac{\left\|\nabla_{\theta_n}g(\theta_n)\right\|^2}{S_{n-1}^{\frac{1}{2}+\varepsilon}} + 4M^2\alpha_0^3 c^2 I_n^{>a}\frac{\left\|\nabla_{\theta_{n-1}}g(\theta_{n-1}),\xi_{n-1}\right\|^2}{S_{n-1}^{\frac{3}{2}+\varepsilon}}\\
&\quad + \frac{c\alpha_0^2}{2}I_n^{>a}\frac{\left\|\nabla_{\theta_n}g(\theta_n,\xi_n)\right\|^2}{S_n^{1+\varepsilon}} + 5\alpha_0^3 c^2(M+1)^2 I_n^{\leq a}\frac{\left\|\nabla_{\theta_{n-1}}g(\theta_{n-1},\xi_{n-1})\right\|^2}{S_{n-1}^{1+\varepsilon}}\\
&\quad + I_n^{\leq a}\frac{(M+1)\alpha_0^3 c^2}{2S_{n-1}^{\frac{3}{2}+\varepsilon}}\left\|\nabla_{\theta_{n-1}}g(\theta_{n-1},\xi_{n-1})\right\|^2\\
&\quad + \frac{\alpha_0 a(M+1)}{2}I_n^{\leq a}\left(\frac{1}{S_{n-1}^{\frac{1}{2}+\varepsilon}} - \frac{1}{S_n^{\frac{1}{2}+\varepsilon}}\right) + \frac{c\alpha_0^2}{2}I_n^{\leq a}\frac{\left\|\nabla_{\theta_n}g(\theta_n,\xi_n)\right\|^2}{S_n^{1+\varepsilon}}\\
&\quad + I_n^{\leq a}A_n^{(\varepsilon)} + I_n^{\leq a}B_n^{(\varepsilon)} + I_n^{>a}X_n^{(\varepsilon)} + I_n^{>a}Y_n^{(\varepsilon)}.
\end{aligned}
\tag{111}
$$

Notice $I_n^{\leq a} \leq 1$, then we get

$$\frac{\alpha_0 a(M+1)}{2} I_n^{\leq a} \left( \frac{1}{S_{n-1}^{\frac{1}{2}+\varepsilon}} - \frac{1}{S_n^{\frac{1}{2}+\varepsilon}} \right) \leq \frac{\alpha_0 a(M+1)}{2} \left( \frac{1}{S_{n-1}^{\frac{1}{2}+\varepsilon}} - \frac{1}{S_n^{\frac{1}{2}+\varepsilon}} \right). \tag{112}$$

Substitute equation 112 into equation 111, then we get

$$\frac{g(\theta_{n+1})}{S_{n+1}^\varepsilon} - \frac{g(\theta_n)}{S_n^\varepsilon}$$
$$\leq \frac{\alpha_0}{20} \left( \frac{\left\| \nabla_{\theta_{n-1}} g(\theta_{n-1}) \right\|^2}{S_{n-2}^{\frac{1}{2}+\varepsilon}} - \frac{\left\| \nabla_{\theta_n} g(\theta_n) \right\|^2}{S_{n-1}^{\frac{1}{2}+\varepsilon}} \right)$$
$$+ \frac{\alpha_0}{2}(M+1) \left( \frac{\left\| \nabla_{\theta_{n-1}} g(\theta_{n-1}) \right\|^2}{S_{n-1}^{\frac{1}{2}+\varepsilon}} - \frac{\left\| \nabla_{\theta_n} g(\theta_n) \right\|^2}{S_n^{\frac{1}{2}+\varepsilon}} \right)$$
$$- \frac{\alpha_0}{20} \frac{\left\| \nabla_{\theta_n} g(\theta_n) \right\|^2}{S_{n-1}^{\frac{1}{2}+\varepsilon}} + 4M^2 \alpha_0^3 c^2 I_n^{>a} \frac{\left\| \nabla_{\theta_{n-1}} g(\theta_{n-1}), \xi_{n-1} \right\|^2}{S_{n-1}^{\frac{3}{2}+\varepsilon}} + \frac{c\alpha_0^2}{2} I_n^{>a} \frac{\left\| \nabla_{\theta_n} g(\theta_n, \xi_n) \right\|^2}{S_n^{1+\varepsilon}}$$
$$+ 5\alpha_0^3 c^2 (M+1)^2 I_n^{\leq a} \frac{\left\| \nabla_{\theta_{n-1}} g(\theta_{n-1}, \xi_{n-1}) \right\|^2}{S_{n-1}^{1+\varepsilon}} + I_n^{\leq a} \frac{(M+1)\alpha_0^3 c^2}{2 S_{n-1}^{\frac{3}{2}+\varepsilon}} \left\| \nabla_{\theta_{n-1}} g(\theta_{n-1}, \xi_{n-1}) \right\|^2$$
$$+ \frac{\alpha_0 a(M+1)}{2} \left( \frac{1}{S_{n-1}^{\frac{1}{2}+\varepsilon}} - \frac{1}{S_n^{\frac{1}{2}+\varepsilon}} \right) + \frac{c\alpha_0^2}{2} I_n^{\leq a} \frac{\left\| \nabla_{\theta_n} g(\theta_n, \xi_n) \right\|^2}{S_n^{1+\varepsilon}} + I_n^{\leq a} A_n^{(\varepsilon)} + I_n^{\leq a} B_n^{(\varepsilon)} + I_n^{>a} X_n^{(\varepsilon)}$$
$$+ I_n^{>a} Y_n^{(\varepsilon)}. \tag{113}$$

We make a summation of equation 113 to get

$$\sum_{k=3}^n \left( \frac{g(\theta_{k+1})}{S_{k+1}^\varepsilon} - \frac{g(\theta_k)}{S_k^\varepsilon} \right)$$
$$\leq \frac{\alpha_0}{20} \sum_{k=3}^n \left( \frac{\left\| \nabla_{\theta_{k-1}} g(\theta_{k-1}) \right\|^2}{S_{k-2}^{\frac{1}{2}+\varepsilon}} - \frac{\left\| \nabla_{\theta_k} g(\theta_k) \right\|^2}{S_{k-1}^{\frac{1}{2}+\varepsilon}} \right)$$
$$+ \frac{\alpha_0}{2}(M+1) \sum_{k=3}^n \left( \frac{\left\| \nabla_{\theta_{k-1}} g(\theta_{k-1}) \right\|^2}{S_{k-1}^{\frac{1}{2}+\varepsilon}} - \frac{\left\| \nabla_{\theta_k} g(\theta_k) \right\|^2}{S_k^{\frac{1}{2}+\varepsilon}} \right)$$
$$- \frac{\alpha_0}{20} \sum_{k=3}^n \frac{\left\| \nabla_{\theta_k} g(\theta_k) \right\|^2}{S_{k-1}^{\frac{1}{2}+\varepsilon}} + 4M^2 \alpha_0^3 c^2 \sum_{k=3}^n I_k^{>a} \frac{\left\| \nabla_{\theta_{k-1}} g(\theta_{k-1}), \xi_{k-1} \right\|^2}{S_{k-1}^{\frac{3}{2}+\varepsilon}} \tag{114}$$
$$+ \frac{c\alpha_0^2}{2} \sum_{k=3}^n I_k^{>a} \frac{\left\| \nabla_{\theta_k} g(\theta_k, \xi_k) \right\|^2}{S_k^{1+\varepsilon}} + 5\alpha_0^3 c^2 (M+1)^2 \sum_{k=2}^n I_k^{\leq a} \frac{\left\| \nabla_{\theta_{k-1}} g(\theta_{k-1}, \xi_{k-1}) \right\|^2}{S_{k-1}^{1+\varepsilon}}$$
$$+ \sum_{k=3}^n I_k^{\leq a} \frac{(M+1)\alpha_0^3 c^2}{2 S_{k-1}^{\frac{3}{2}+\varepsilon}} \left\| \nabla_{\theta_{k-1}} g(\theta_{k-1}, \xi_{k-1}) \right\|^2$$
$$+ \frac{\alpha_0 a(M+1)}{2} \sum_{k=3}^n \left( \frac{1}{S_{k-1}^{\frac{1}{2}+\varepsilon}} - \frac{1}{S_k^{\frac{1}{2}+\varepsilon}} \right) + \frac{c\alpha_0^2}{2} \sum_{k=3}^n I_k^{\leq a} \frac{\left\| \nabla_{\theta_k} g(\theta_k, \xi_k) \right\|^2}{S_k^{1+\varepsilon}}$$
$$+ \sum_{k=3}^n \left( I_k^{\leq a} A_k^{(\varepsilon)} + I_k^{\leq a} B_k^{(\varepsilon)} + I_k^{>a} X_k^{(\varepsilon)} + I_k^{>a} Y_k^{(\varepsilon)} \right).$$

It follows from Lemma 16 that

$$\sum_{k=3}^n \frac{\left\| \nabla_{\theta_k} g(\theta_k, \xi_k) \right\|^2}{S_k^{1+\varepsilon}} \leq \int_{S_3}^{+\infty} \frac{1}{x^{1+\varepsilon}} dx = \frac{1}{\varepsilon S_3^\varepsilon}, \tag{115}$$

and

$$\sum_{k=3}^{n} \frac{\left\|\nabla_{\theta_{k-1}} g(\theta_{k-1}, \xi_{k-1})\right\|^2}{S_{k-1}^{1+\varepsilon}} \le \int_{S_2}^{+\infty} \frac{1}{x^{1+\varepsilon}} dx = \frac{1}{\varepsilon S_2^{\varepsilon}}, \tag{116}$$

and

$$\sum_{k=3}^{n} \frac{\left\|\nabla_{\theta_{k-1}} g(\theta_{k-1}, \xi_{k-1})\right\|^2}{S_{k-1}^{\frac{3}{2}+\varepsilon}} \le \int_{S_2}^{+\infty} \frac{1}{x^{\frac{3}{2}+\varepsilon}} dx = \frac{2}{(1+2\varepsilon)S_2^{\frac{1}{2}+\varepsilon}}. \tag{117}$$

Due to $I_k^{\le a} \le 1$ and $I_k^{>a} \le 1$, we get that

$$\begin{aligned}
&4M^2\alpha_0^3 c^2 \sum_{k=3}^{n} I_k^{>a} \frac{\left\|\nabla_{\theta_{k-1}} g(\theta_{k-1}), \xi_{k-1}\right\|^2}{S_{k-1}^{\frac{3}{2}+\varepsilon}} + \frac{c\alpha_0^2}{2} \sum_{k=3}^{n} I_k^{>a} \frac{\left\|\nabla_{\theta_k} g(\theta_k, \xi_k)\right\|^2}{S_k^{1+\varepsilon}} \\
&+ 5\alpha_0^3 c^2 (M+1)^2 \sum_{k=3}^{n} I_k^{\le a} \frac{\left\|\nabla_{\theta_{k-1}} g(\theta_{k-1}, \xi_{k-1})\right\|^2}{S_{k-1}^{1+\varepsilon}} \\
&+ \sum_{k=3}^{n} I_k^{\le a} \frac{(M+1)\alpha_0^3 c^2}{2 S_{k-1}^{\frac{3}{2}+\varepsilon}} \left\|\nabla_{\theta_{k-1}} g(\theta_{k-1}, \xi_{k-1})\right\|^2 + \frac{c\alpha_0^2}{2} \sum_{k=3}^{n} I_k^{\le a} \frac{\left\|\nabla_{\theta_k} g(\theta_k, \xi_k)\right\|^2}{S_k^{1+\varepsilon}} \\
&\le \frac{8M^2\alpha_0^3 c^2}{(1+2\varepsilon)S_2^{\frac{1}{2}+\varepsilon}} + \frac{c\alpha_0^2}{2\varepsilon S_3^{\varepsilon}} + \frac{5\alpha_0^3 c^2 (M+1)^2}{\varepsilon S_2^{\varepsilon}} + \frac{\alpha_0^3 c^2 (M+1)}{(1+2\varepsilon)S_2^{\frac{1}{2}+\varepsilon}} + \frac{c\alpha_0^2}{2\varepsilon S_3^{\varepsilon}} := K.
\end{aligned} \tag{118}$$

Substituting equation 118 into equation 114 leads to

$$\begin{aligned}
&\frac{g(\theta_{n+1})}{S_{n+1}^{\varepsilon}} - \frac{g(\theta_3)}{S_3^{\varepsilon}} \\
&\le \frac{\alpha_0}{20} \left( \frac{\left\|\nabla_{\theta_2} g(\theta_2)\right\|^2}{S_1^{\frac{1}{2}+\varepsilon}} - \frac{\left\|\nabla_{\theta_n} g(\theta_n)\right\|^2}{S_{n-1}^{\frac{1}{2}+\varepsilon}} \right) + \frac{\alpha_0}{2}(M+1) \left( \frac{\left\|\nabla_{\theta_2} g(\theta_2)\right\|^2}{S_2^{\frac{1}{2}+\varepsilon}} - \frac{\left\|\nabla_{\theta_n} g(\theta_n)\right\|^2}{S_n^{\frac{1}{2}+\varepsilon}} \right) \\
&- \frac{\alpha_0}{20} \sum_{k=3}^{n} \frac{\left\|\nabla_{\theta_k} g(\theta_k)\right\|^2}{S_{k-1}^{\frac{1}{2}+\varepsilon}} + \frac{\alpha_0 a (M+1)}{2} \left( \frac{1}{S_2^{\frac{1}{2}+\varepsilon}} - \frac{1}{S_n^{\frac{1}{2}+\varepsilon}} \right) + K \\
&+ \sum_{k=3}^{n} \left( I_k^{\le a} A_k^{(\varepsilon)} + I_k^{\le a} B_k^{(\varepsilon)} + I_k^{>a} X_k^{(\varepsilon)} + I_k^{>a} Y_k^{(\varepsilon)} \right).
\end{aligned}$$

It is obvious that

$$\begin{aligned}
&\frac{\alpha_0}{20} \left( \frac{\left\|\nabla_{\theta_0} g(\theta_0)\right\|^2}{S_1^{\frac{1}{2}+\varepsilon}} - \frac{\left\|\nabla_{\theta_n} g(\theta_n)\right\|^2}{S_{n-1}^{\frac{1}{2}+\varepsilon}} \right) + \frac{\alpha_0}{2}(M+1) \left( \frac{\left\|\nabla_{\theta_2} g(\theta_2)\right\|^2}{S_2^{\frac{1}{2}+\varepsilon}} - \frac{\left\|\nabla_{\theta_n} g(\theta_n)\right\|^2}{S_n^{\frac{1}{2}+\varepsilon}} \right) \\
&+ \frac{\alpha_0 a (M+1)}{2} \left( \frac{1}{S_2^{\frac{1}{2}+\varepsilon}} - \frac{1}{S_n^{\frac{1}{2}+\varepsilon}} \right) + K \le \frac{\alpha_0}{20} \frac{\left\|\nabla_{\theta_2} g(\theta_2)\right\|^2}{S_1^{\frac{1}{2}+\varepsilon}} + \frac{\alpha_0(M+1)}{2} \frac{\left\|\nabla_{\theta_2} g(\theta_2)\right\|^2}{S_2^{\frac{1}{2}+\varepsilon}} \\
&+ \frac{\alpha_0 a (M+1)}{2 S_2^{\frac{1}{2}+\varepsilon}} + K := L.
\end{aligned}$$

It follows that

$$\begin{aligned}
&\frac{g(\theta_{n+1})}{S_{n+1}^{\varepsilon}} - \frac{g(\theta_3)}{S_3^{\varepsilon}} \\
&\le -\frac{\alpha_0}{20} \sum_{k=3}^{n} \frac{\left\|\nabla_{\theta_k} g(\theta_k)\right\|^2}{S_{k-1}^{\frac{1}{2}+\varepsilon}} + L + \sum_{k=3}^{n} \left( I_k^{\le a} A_k^{(\varepsilon)} + I_k^{\le a} B_k^{(\varepsilon)} + I_k^{>a} X_k^{(\varepsilon)} + I_k^{>a} Y_k^{(\varepsilon)} \right).
\end{aligned} \tag{119}$$

Note that $\{I_k^{\le a} A_k\}$, $\{I_k^{\le a} B_k\}$, $\{I_k^{>a} X_k\}$ and $\{I_k^{>a} Y_k\}$ are all martingale difference sequences, thus it follows that $\mathbb{E} \left( \sum_{k=1}^{n} \left( I_k^{\le a} A_k + I_k^{\le a} B_k + I_k^{>a} X_k + I_k^{>a} Y_k \right) \right) = 0$. Then we calculate mathematical

expectation on equation 119

$$\mathbb{E}\left(\frac{g(\theta_{n+1})}{S_{n+1}^{\varepsilon}} - \frac{g(\theta_3)}{S_3^{\varepsilon}}\right) \leq -\frac{\alpha_0}{20}\mathbb{E}\left(\sum_{k=3}^{n}\frac{\left\|\nabla_{\theta_k}g(\theta_k)\right\|^2}{S_{k-1}^{\frac{1}{2}+\varepsilon}}\right) + L.$$

That is

$$\mathbb{E}\left(\sum_{k=3}^{n}\frac{\left\|\nabla_{\theta_k}g(\theta_k)\right\|^2}{S_{k-1}^{\frac{1}{2}+\varepsilon}}\right) < \frac{20}{\alpha_0}\left(\frac{g(\theta_3)}{S_3^{\varepsilon}} + L\right) < +\infty. \tag{120}$$

From Lemma 6, it holds that

$$\sum_{k=3}^{n}\frac{\left\|\nabla_{\theta_k}g(\theta_k)\right\|^2}{S_{k-1}^{\frac{1}{2}+\varepsilon}} < +\infty \quad a.s.. \tag{121}$$

## C.6 PROOF OF LEMMA 14

It follows from equation 119 that

$$\frac{g(\theta_{n+1})}{S_{n+1}^{\varepsilon}} \leq \frac{g(\theta_3)}{S_3^{\varepsilon}} + L + \sum_{k=3}^{n}\left(I_k^{\leq a}A_k^{(\varepsilon)} + I_k^{\leq a}B_k^{(\varepsilon)} + I_k^{>a}X_k^{(\varepsilon)} + I_k^{>a}Y_k^{(\varepsilon)}\right). \tag{122}$$

From equation 101, we obtain

$$\sum_{k=3}^{n}\mathbb{E}\left(\left\|I_k^{\leq a}A_k^{(\varepsilon)}\right\|^2\right) \leq \sum_{k=3}^{n}\mathbb{E}\left(I_k^{\leq a}\frac{\alpha_0^2}{S_{n-1}^{1+2\varepsilon}}\left(\left\|\nabla_{\theta_n}g(\theta_n)\right\|^2 - \nabla_{\theta_n}g(\theta_n)^T\nabla_{\theta_n}g(\theta_n,\xi_n)\right)^2\right).$$

With inequality $(a+b)^2 \leq 2(a^2+b^2)$, $2a^Tb \leq a^2+b^2$ $(a, b > 0)$ and equation 120, we get

$$\mathbb{E}\left(I_k^{\leq a}\frac{\alpha_0^2}{S_{n-1}^{1+2\varepsilon}}\left(\left\|\nabla_{\theta_n}g(\theta_n)\right\|^2 - \nabla_{\theta_n}g(\theta_n)^T\nabla_{\theta_n}g(\theta_n,\xi_n)\right)^2\right)$$

$$\leq 2\mathbb{E}\left(\frac{\alpha_0^2\left\|\nabla_{\theta_n}g(\theta_n)\right\|^2}{S_{n-1}^{1+2\varepsilon}}\left(I_k^{\leq a}\left\|\nabla_{\theta_n}g(\theta_n)\right\|^2\right)\right) + 2\mathbb{E}\left(\frac{\alpha_0^2 I_k^{\leq a}}{S_{n-1}^{1+2\varepsilon}}\left\|\nabla_{\theta_n}g(\theta_n)\right\|^2\left\|\nabla_{\theta_n}g(\theta_n,\xi_n)\right\|^2\right)$$

$$\leq 2(M+2)\mathbb{E}\left(\frac{\alpha_0^2 a\left\|\nabla_{\theta_n}g(\theta_n)\right\|^2}{S_{n-1}^{1+\varepsilon}}\right)$$

$$\leq 2(M+2)\alpha_0^2 a\,\mathbb{E}\left(\frac{\alpha_0^2 a\left\|\nabla_{\theta_n}g(\theta_n)\right\|^2}{S_{n-1}^{\frac{1}{2}+\varepsilon}}\right)$$

$$< 40(M+2)\alpha_0 a\left(\frac{g(\theta_1)}{S_1^{\varepsilon}} + L\right).$$
$$\tag{123}$$

It follows from Lemma 5 that $\sum_{k=3}^{n}I_k^{\leq a}A_k^{(\varepsilon)}$ is convergent a.s. Similarly, $\sum_{k=3}^{n}I_k^{\leq a}B_k^{(\varepsilon)}$, $\sum_{k=3}^{n}I_k^{>a}X_k^{(\varepsilon)}$ and $\sum_{k=3}^{n}I_k^{>a}Y_k^{(\varepsilon)}$ are both convergent a.s. It follows that

$$\sum_{k=3}^{n}\left(I_k^{\leq a}A_k^{(\varepsilon)} + I_k^{\leq a}B_k^{(\varepsilon)} + I_k^{>a}X_k^{(\varepsilon)} + I_k^{>a}Y_k^{(\varepsilon)}\right) < \xi' < +\infty \quad a.s.,$$

$$\frac{g(\theta_{n+1})}{S_{n+1}^{\varepsilon}} \leq \frac{g(\theta_3)}{S_3^{\varepsilon}} + L + \xi' < +\infty \quad a.s.$$

For convenience, let $\xi = \frac{g(\theta_3)}{S_3^{\varepsilon}} + L + \xi'$. Thus, it holds that

$$\frac{g(\theta_{n+1}) - g^*}{S_{n+1}^{\varepsilon}} < \xi < +\infty \quad a.s.. \tag{124}$$

## C.7 PROOF OF LEMMA 15

First of all, $\forall\, 0 < \varepsilon_0 < \frac{3}{8}$, there is $0 < \frac{4}{3}\varepsilon_0 < \frac{1}{2}$. From Lemma 13, it follows that

$$\sum_{n=4}^{+\infty} \frac{\left\|\nabla_{\theta_{n-1}} g(\theta_{n-1})\right\|^2}{S_{n-2}^{\frac{1}{2}+\frac{4}{3}\varepsilon_0}} < +\infty \quad a.s.. \tag{125}$$

It follows from equation 90 that

$$\left\|\nabla_{\theta_n} g(\theta_n)\right\|^2 - \left\|\nabla_{\theta_{n-1}} g(\theta_{n-1})\right\|^2 \le \frac{\alpha_0 c}{\sqrt{S_{n-1}}}\left(\left\|\nabla_{\theta_{n-1}} g(\theta_{n-1})\right\|^2 + \left\|\nabla_{\theta_{n-1}} g(\theta_{n-1},\xi_{n-1})\right\|^2\right)$$
$$+ \frac{\alpha_0^2 c^2}{S_{n-1}}\left\|\nabla_{\theta_{n-1}} g(\theta_{n-1},\xi_{n-1})\right\|^2. \tag{126}$$

Divide both sides of equation 126 by $S_{n-1}^{\varepsilon_0}$ and notice that $S_n > S_{n-1} > S_{n-2}$, then we get

$$\frac{\left\|\nabla_{\theta_n} g(\theta_n)\right\|^2}{S_n^{\varepsilon_0}} - \frac{\left\|\nabla_{\theta_{n-1}} g(\theta_{n-1})\right\|^2}{S_{n-1}^{\varepsilon_0}} \le \frac{\alpha_0 c\left\|\nabla_{\theta_{n-1}} g(\theta_{n-1})\right\|^2}{S_{n-2}^{\frac{1}{2}+\varepsilon_0}} + \frac{\alpha_0 c\left\|\nabla_{\theta_{n-1}} g(\theta_{n-1},\xi_{n-1})\right\|^2}{S_{n-2}^{\frac{1}{2}+\varepsilon_0}}$$
$$+ \frac{\alpha_0^2 c^2\left\|\nabla_{\theta_{n-1}} g(\theta_{n-1},\xi_{n-1})\right\|^2}{S_{n-2}^{1+\varepsilon_0}}. \tag{127}$$

Note that $0 < \frac{2}{3}\varepsilon_0 < \frac{1}{2}$, then it follows from Lemma 10 and Lemma 14 that

$$\frac{\left\|\nabla_{\theta_{n-1}} g(\theta_{n-1})\right\|^2}{S_{n-2}^{\frac{2}{3}\varepsilon_0}} \le \frac{2c\left(g(\theta_{n-1}) - g^*\right)}{S_{n-2}^{\frac{2}{3}\varepsilon_0}} < 2c\xi < +\infty.$$

It follows that

$$\frac{\left\|\nabla_{\theta_n} g(\theta_n)\right\|^2}{S_n^{\varepsilon_0}} - \frac{\left\|\nabla_{\theta_{n-1}} g(\theta_{n-1})\right\|^2}{S_{n-1}^{\varepsilon_0}} \le \frac{\alpha_0 ct\xi}{S_{n-2}^{\frac{1}{2}+\frac{1}{3}\varepsilon_0}} + \frac{\alpha_0 cMt\xi}{S_{n-2}^{\frac{1}{2}+\frac{1}{3}\varepsilon_0}} + \frac{\alpha_0^2 c^2 Mt\xi}{S_{n-2}^{\frac{1}{2}+\frac{1}{3}\varepsilon_0}} + \frac{\alpha_0 ca}{S_{n-2}^{\frac{1}{2}+\frac{1}{3}\varepsilon_0}}$$
$$+ \frac{\alpha_0^2 c^2 a}{S_{n-2}^{\frac{1}{2}+\frac{1}{3}\varepsilon_0}} - \frac{\alpha_0 c}{S_{n-2}^{\frac{1}{2}+\varepsilon_0}}\left(\mathbb{E}\left(\left\|\nabla_{\theta_{n-1}} g(\theta_{n-1},\xi_{n-1})\right\|^2\bigg|\mathscr{F}_{n-2}\right) - \left\|\nabla_{\theta_{n-1}} g(\theta_{n-1},\xi_{n-1})\right\|^2\right)$$
$$- \frac{\alpha_0^2 c^2}{S_{n-2}^{1+\varepsilon_0}}\left(\mathbb{E}\left(\left\|\nabla_{\theta_{n-1}} g(\theta_{n-1},\xi_{n-1})\right\|^2\bigg|\mathscr{F}_{n-2}\right) - \left\|\nabla_{\theta_{n-1}} g(\theta_{n-1},\xi_{n-1})\right\|^2\right).$$

Thus, we have

$$\frac{\left\|\nabla_{\theta_n} g(\theta_n)\right\|^2}{S_n^{\varepsilon_0}} - \frac{\left\|\nabla_{\theta_{n-1}} g(\theta_{n-1})\right\|^2}{S_{n-1}^{\varepsilon_0}} \le \frac{\zeta}{S_{n-2}^{\frac{1}{2}+\frac{1}{3}\varepsilon_0}} - K_{n-1}, \tag{128}$$

where

$$\zeta = \alpha_0 ct\xi(M+1+\alpha_0 c) + \alpha_0 c(a+\alpha_0 c+1)$$
$$K_{n-1} = \frac{\alpha_0 c}{S_{n-2}^{\frac{1}{2}+\varepsilon_0}}\left(\mathbb{E}\left(\left\|\nabla_{\theta_{n-1}} g(\theta_{n-1},\xi_{n-1})\right\|^2\bigg|\mathscr{F}_{n-2}\right) - \left\|\nabla_{\theta_{n-1}} g(\theta_{n-1},\xi_{n-1})\right\|^2\right)$$
$$+ \frac{\alpha_0 c}{S_{n-2}^{1+\varepsilon_0}}\left(\mathbb{E}\left(\left\|\nabla_{\theta_{n-1}} g(\theta_{n-1},\xi_{n-1})\right\|^2\bigg|\mathscr{F}_{n-2}\right) - \left\|\nabla_{\theta_{n-1}} g(\theta_{n-1},\xi_{n-1})\right\|^2\right).$$

It follows that

$$\mathbb{E}\left(\frac{1}{S_{n-2}^{\frac{1}{2}+\frac{1}{3}\varepsilon_0}}\right) \ge \frac{1}{a}\mathbb{E}\left(\frac{\left\|\nabla_{\theta_{n-1}} g(\theta_{n-1},\xi_{n-1})\right\|^2 - M\left\|\nabla_{\theta_{n-1}} g(\theta_{n-1})\right\|^2}{S_{n-2}^{\frac{1}{2}+\frac{1}{3}\varepsilon_0}}\right)$$

By Lemma 16, we have that

$$
\begin{aligned}
&\sum_{n=4}^{+\infty} \frac{1}{a} \mathbb{E}\left(\frac{\left\|\nabla_{\theta_{n-1}} g(\theta_{n-1}, \xi_{n-1})\right\|^2}{S_{n-2}^{\frac{1}{2}+\frac{1}{3}\varepsilon_0}}\right) \\
&> \lim_{n\to+\infty} \frac{1}{a} \mathbb{E}\left(\int_{S_2}^{S_{n-1}} \frac{1}{x^{\frac{1}{2}+\frac{1}{3}\varepsilon_0}} dx\right) = \lim_{n\to+\infty} \frac{6}{2-3\varepsilon_0} \frac{1}{a} \mathbb{E}\left(S_{n-1}^{\frac{1}{2}-\frac{1}{3}\varepsilon_0} - S_2^{\frac{1}{2}-\frac{1}{3}\varepsilon_0}\right).
\end{aligned}
$$

Due to $S_{n-1} \to +\infty$ *a.s.*, it follows that

$$
\sum_{n=4}^{+\infty} \frac{1}{a} \mathbb{E}\left(\frac{\left\|\nabla_{\theta_{n-1}} g(\theta_{n-1}, \xi_{n-1})\right\|^2}{S_{n-2}^{\frac{1}{2}+\frac{1}{3}\varepsilon_0}}\right) = +\infty.
\tag{129}
$$

From Lemma 13, we get

$$
\sum_{n=4}^{+\infty} \frac{1}{a} \mathbb{E}\left(\frac{\left\|\nabla_{\theta_{n-1}} g(\theta_{n-1})\right\|^2}{S_{n-2}^{\frac{1}{2}+\frac{1}{3}\varepsilon_0}}\right) < +\infty.
\tag{130}
$$

Combine equation 129 and equation 130, then

$$
\begin{aligned}
&\sum_{n=4}^{+\infty} E\left(\frac{1}{S_{n-2}^{\frac{1}{2}+\frac{1}{3}\varepsilon_0}}\right) > \sum_{n=4}^{+\infty} \frac{1}{a} \mathbb{E}\left(\frac{\left\|\nabla_{\theta_{n-1}} g(\theta_{n-1}, \xi_{n-1})\right\|^2}{S_{n-2}^{\frac{1}{2}+\frac{1}{3}\varepsilon_0}}\right) \\
&- \sum_{n=4}^{+\infty} \frac{1}{a} \mathbb{E}\left(\frac{M\left\|\nabla_{\theta_{n-1}} g(\theta_{n-1})\right\|^2}{S_{n-2}^{\frac{1}{2}+\frac{1}{3}\varepsilon_0}}\right) = +\infty
\end{aligned}
\tag{131}
$$

is divergent a.s. From Lemma 13, we get that $\sum_{n=4}^{+\infty} \mathbb{E}\left(\left\|K_{n-1}\right\|^2\right) < +\infty$ and $K_{n-1}$ is a martingale difference sequence. Thus, $\sum_{n=2}^{+\infty} K_{n-1}$ is convergent a.s.. Combine equation 128, equation 131 and Lemma 2, then we have

$$
\frac{\left\|\nabla_{\theta_n} g(\theta_n)\right\|^2}{S_{n-1}^{\varepsilon_0}} \to 0 \quad a.s..
\tag{132}
$$

## C.8    THE PROOF OF THEOREM 3

We consider the proof under two conditions, namely, $S_n < +\infty$ *a.s.* or $S_n = +\infty$ *a.s.*.

First, if $S_n < +\infty$ *a.s.*, from Lemma 13, we get that $\forall \varepsilon \in (0, \frac{1}{2})$, it holds that

$$
\sum_{k=3}^{n} \frac{\left\|\nabla_{\theta_k} g(\theta_k)\right\|^2}{S_{k-1}^{\frac{1}{2}+\varepsilon}} < +\infty \quad a.s.
$$

Thus, we conclude that

$$
\frac{\left\|\nabla_{\theta_n} g(\theta_n)\right\|^2}{S_{n-1}^{\frac{1}{2}+\varepsilon}} \to 0 \quad a.s..
$$

Due to $S_{n-1} < +\infty$ *a.s.*, we get

$$
\left\|\nabla_{\theta_n} g(\theta_n)\right\|^2 \to 0 \quad a.s.,
\tag{133}
$$

Second, if $S_n = +\infty$ $a.s.$, let $\varepsilon \to 0$ on equation 113, then it holds that

$$
\begin{aligned}
&g(\theta_{n+1}) - g(\theta_n) \\
&\leq \frac{\alpha_0}{20} \left( \frac{\left\| \nabla_{\theta_{n-1}} g(\theta_{n-1}) \right\|^2}{\sqrt{S_{n-2}}} - \frac{\left\| \nabla_{\theta_n} g(\theta_n) \right\|^2}{\sqrt{S_{n-1}}} \right) \\
&\quad + \frac{\alpha_0}{2}(M+1) \left( \frac{\left\| \nabla_{\theta_{n-1}} g(\theta_{n-1}) \right\|^2}{\sqrt{S_{n-1}}} - \frac{\left\| \nabla_{\theta_n} g(\theta_n) \right\|^2}{\sqrt{S_n}} \right) \\
&\quad - \frac{\alpha_0}{20} \frac{\left\| \nabla_{\theta_n} g(\theta_n) \right\|^2}{\sqrt{S_{n-1}}} + 4M^2 \alpha_0^3 c^2 \frac{\left\| \nabla_{\theta_{n-1}} g(\theta_{n-1}), \xi_{n-1} \right\|^2}{S_{n-1}^{\frac{3}{2}}} + c\alpha_0^2 \frac{\left\| \nabla_{\theta_n} g(\theta_n, \xi_n) \right\|^2}{S_n} \quad (134) \\
&\quad + \frac{\alpha_0^3 c^2 (M+1)^2}{2} \frac{\left\| \nabla_{\theta_{n-1}} g(\theta_{n-1}, \xi_{n-1}) \right\|^2}{S_{n-1}} + \frac{(M+1)\alpha_0^3 c^2}{2 S_{n-1}^{\frac{3}{2}}} \left\| \nabla_{\theta_{n-1}} g(\theta_{n-1}, \xi_{n-1}) \right\|^2 \\
&\quad + \frac{\alpha_0 a (M+1)}{2} \left( \frac{1}{\sqrt{S_{n-1}}} - \frac{1}{\sqrt{S_n}} \right) + I_n^{\leq a} A_n + I_n^{\leq a} B_n + I_n^{>a} X_n + I_n^{>a} Y_n,
\end{aligned}
$$

where

$$
\begin{aligned}
&I_n^{\leq a} A_n + I_n^{\leq a} B_n + I_n^{>a} X_n + I_n^{>a} Y_n = \lim_{\varepsilon \to 0} \left( I_n^{\leq a} A_n^{(\varepsilon)} + I_n^{\leq a} B_n^{(\varepsilon)} + I_n^{>a} X_n^{(\varepsilon)} + I_n^{>a} Y_n^{(\varepsilon)} \right) \\
&= \frac{\alpha_0 I_n^{\leq a}}{\sqrt{S_{n-1}}} \nabla_{\theta_n} g(\theta_n)^T \left( \nabla_{\theta_n} g(\theta_n) - \nabla_{\theta_n} g(\theta_n, \xi_n) \right) \\
&\quad + \frac{\alpha_0 \left\| \nabla_{\theta_n} g(\theta_n) \right\|^2 I_n^{\leq a}}{2a(M+1)\sqrt{S_{n-1}}} \left( \left\| \nabla_{\theta_n} g(\theta_n, \xi_n) \right\|^2 - \mathbb{E} \left( \left\| \nabla_{\theta_n} g(\theta_n) \right\|^2 \Big| \mathscr{F}_{n-1} \right) \right) \\
&\quad + \frac{\alpha_0}{2(M+1)} \frac{I_n^{>a}}{\sqrt{S_{n-1}}} \left( \mathbb{E} \left( \left\| \nabla_{\theta_n} g(\theta_n, \xi_n) \right\|^2 \Big| \mathscr{F}_{n-1} \right) - \left\| \nabla_{\theta_n} g(\theta_n, \xi_n) \right\|^2 \right) \\
&\quad + \frac{\alpha_0}{2} \frac{I_n^{>a}}{\sqrt{S_{n-1}}} \nabla_{\theta_n} g(\theta_n)^T \left( \nabla_{\theta_n} g(\theta_n) - \nabla_{\theta_n} g(\theta_n, \xi_n) \right).
\end{aligned}
$$

Make some transformations on $\frac{\left\| \nabla_{\theta_n} g(\theta_n), \xi_n \right\|^2}{S_n}$ to obtain that

$$
\begin{aligned}
\frac{\left\| \nabla_{\theta_n} g(\theta_n, \xi_n) \right\|^2}{S_n} &\leq \frac{\left\| \nabla_{\theta_n} g(\theta_n, \xi_n) \right\|^2}{S_{n-1}} \\
&= \frac{\mathbb{E} \left( \left\| \nabla_{\theta_n} g(\theta_n), \xi_n \right\|^2 \Big| \mathscr{F}_{n-1} \right)}{S_{n-1}} + \frac{1}{S_{n-1}} \left( \left\| \nabla_{\theta_n} g(\theta_n, \xi_n) \right\|^2 - \mathbb{E} \left( \left\| \nabla_{\theta_n} g(\theta_n), \xi_n \right\|^2 \Big| \mathscr{F}_{n-1} \right) \right) \\
&\leq \frac{M \left\| \nabla_{\theta_n} g(\theta_n) \right\|^2}{S_{n-1}} + \frac{a}{S_{n-1}} + \frac{1}{S_{n-1}} \left( \left\| \nabla_{\theta_n} g(\theta_n, \xi_n) \right\|^2 - \mathbb{E} \left( \left\| \nabla_{\theta_n} g(\theta_n), \xi_n \right\|^2 \Big| \mathscr{F}_{n-1} \right) \right),
\end{aligned}
$$

where the last inequality is from Assumption 5 5). Let $0 < \varepsilon' < \frac{3}{8}$. It follows from Lemma 15 that $\frac{\left\| \nabla_{\theta_n} g(\theta_n) \right\|^2}{S_{n-1}^{\varepsilon'}} < \delta < +\infty$. Then we have $\frac{M \left\| \nabla_{\theta_n} g(\theta_n) \right\|^2}{S_{n-1}} \leq \frac{M\delta}{S_{n-1}^{1-\varepsilon'}} = \frac{M\delta}{S_{n-1}^{\frac{1}{2}+\varepsilon_1}}$ $\left( \varepsilon_1 = \frac{1}{2} - \varepsilon' \in (0, \frac{1}{4}) \right)$ and

$$
\begin{aligned}
\frac{\left\| \nabla_{\theta_n} g(\theta_n, \xi_n) \right\|^2}{S_n} &\leq \frac{M\delta + a}{S_n^{\frac{1}{2}+\varepsilon_1}} + \frac{1}{S_{n-1}} \left( \left\| \nabla_{\theta_n} g(\theta_n, \xi_n) \right\|^2 - \mathbb{E} \left( \left\| \nabla_{\theta_n} g(\theta_n), \xi_n \right\|^2 \Big| \mathscr{F}_{n-1} \right) \right) \\
&\quad + (M\delta + a) \left( \frac{1}{S_{n-1}^{\frac{1}{2}+\varepsilon_1}} - \frac{1}{S_n^{\frac{1}{2}+\varepsilon_1}} \right).
\end{aligned}
$$
(135)

Similarly, we have

$$
\begin{aligned}
\frac{\left\|\nabla_{\theta_{n-1}}g(\theta_{n-1},\xi_{n-1})\right\|^2}{S_{n-1}} &\leq \frac{M\delta+a}{S_n^{\frac{1}{2}+\varepsilon_1}} + (M\delta+a)\left(\frac{1}{S_{n-2}^{\frac{1}{2}+\varepsilon_1}} - \frac{1}{S_n^{\frac{1}{2}+\varepsilon_1}}\right) \\
&+ \frac{1}{S_{n-2}}\left(\left\|\nabla_{\theta_{n-1}}g(\theta_{n-1},\xi_{n-1})\right\|^2 - \mathbb{E}\left(\left\|\nabla_{\theta_{n-1}}g(\theta_{n-1}),\xi_{n-1}\right\|^2\Big|\mathscr{F}_{n-2}\right)\right).
\end{aligned}
\tag{136}
$$

Substitute equation 135 and equation 136 into equation 134, then we get

$$
g(\theta_{n+1}) - g(\theta_n) \leq \left(c\alpha_0^2 + \frac{c^3\alpha_0^2(M+1)}{2}\right)(M\delta+a)\frac{1}{S_n^{\frac{1}{2}+\varepsilon_1}} + P_n + Q_n,
\tag{137}
$$

where

$$
\begin{aligned}
Q_n =& \frac{c\alpha_0^2}{S_{n-1}}\left(\left\|\nabla_{\theta_n}g(\theta_n,\xi_n)\right\|^2 - \mathbb{E}\left(\left\|\nabla_{\theta_n}g(\theta_n),\xi_n\right\|^2\Big|\mathscr{F}_{n-1}\right)\right) \\
&+ \frac{\alpha_0^3 c^2(M+1)^2}{2S_{n-2}}\left(\left\|\nabla_{\theta_{n-1}}g(\theta_{n-1},\xi_{n-1})\right\|^2 - \mathbb{E}\left(\left\|\nabla_{\theta_{n-1}}g(\theta_{n-1}),\xi_{n-1}\right\|^2\Big|\mathscr{F}_{n-2}\right)\right) \\
&+ I_n^{\leq a}A_n + I_n^{\leq a}B_n + I_n^{>a}X_n + I_n^{>a}Y_n.
\end{aligned}
$$

$$
\begin{aligned}
P_n =& \frac{\alpha_0}{20}\left(\frac{\left\|\nabla_{\theta_{n-1}}g(\theta_{n-1})\right\|^2}{\sqrt{S_{n-2}}} - \frac{\left\|\nabla_{\theta_n}g(\theta_n)\right\|^2}{\sqrt{S_{n-1}}}\right) + \frac{\alpha_0}{2}(M+1)\left(\frac{\left\|\nabla_{\theta_{n-1}}g(\theta_{n-1})\right\|^2}{\sqrt{S_{n-1}}} - \frac{\left\|\nabla_{\theta_n}g(\theta_n)\right\|^2}{\sqrt{S_n}}\right) \\
&+ 4M^2\alpha_0^3 c^2\frac{\left\|\nabla_{\theta_{n-1}}g(\theta_{n-1}),\xi_{n-1}\right\|^2}{S_{n-1}^{\frac{3}{2}}} + \frac{(M+1)\alpha_0^3 c^2}{2S_{n-1}^{\frac{3}{2}}}\left\|\nabla_{\theta_{n-1}}g(\theta_{n-1},\xi_{n-1})\right\|^2 \\
&+ \frac{\alpha_0 a(M+1)}{2}\left(\frac{1}{\sqrt{S_{n-1}}} - \frac{1}{\sqrt{S_n}}\right) + \frac{\alpha_0^3 c^2(M+1)^2(M\delta+a)}{2}\left(\frac{1}{S_{n-1}^{\frac{1}{2}+\varepsilon_1}} - \frac{1}{S_n^{\frac{1}{2}+\varepsilon_1}}\right) \\
&+ c\alpha_0^2(M\delta+a)\left(\frac{1}{S_{n-2}^{\frac{1}{2}+\varepsilon_1}} - \frac{1}{S_n^{\frac{1}{2}+\varepsilon_1}}\right).
\end{aligned}
$$

It follows from $S_n \to +\infty$ *a.s.* that

$$
\begin{aligned}
\sum_{n=3}^{+\infty} P_n =\ & \frac{\alpha_0}{20} \sum_{n=3}^{+\infty} \left( \frac{\left\| \nabla_{\theta_{n-1}} g(\theta_{n-1}) \right\|^2}{\sqrt{S_{n-2}}} - \frac{\left\| \nabla_{\theta_n} g(\theta_n) \right\|^2}{\sqrt{S_{n-1}}} \right) \\
& + \frac{\alpha_0}{2}(M+1) \sum_{n=3}^{+\infty} \left( \frac{\left\| \nabla_{\theta_{n-1}} g(\theta_{n-1}) \right\|^2}{\sqrt{S_{n-1}}} - \frac{\left\| \nabla_{\theta_n} g(\theta_n) \right\|^2}{\sqrt{S_n}} \right) \\
& + 4M^2 \alpha_0^3 c^2 \sum_{n=3}^{+\infty} \frac{\left\| \nabla_{\theta_{n-1}} g(\theta_{n-1}), \xi_{n-1} \right\|^2}{S_{n-1}^{\frac{3}{2}}} + \frac{(M+1)\alpha_0^3 c^2}{2 S_{n-1}^{\frac{3}{2}}} \sum_{n=3}^{+\infty} \left\| \nabla_{\theta_{n-1}} g(\theta_{n-1}, \xi_{n-1}) \right\|^2 \\
& + \sum_{n=3}^{+\infty} \frac{\alpha_0 a(M+1)}{2} \left( \frac{1}{\sqrt{S_{n-1}}} - \frac{1}{\sqrt{S_n}} \right) + \sum_{n=3}^{+\infty} \frac{\alpha_0^3 c^2 (M+1)^2 (M\delta + a)}{2} \left( \frac{1}{S_{n-1}^{\frac{1}{2}+\varepsilon_1}} - \frac{1}{S_n^{\frac{1}{2}+\varepsilon_1}} \right) \\
& + \sum_{n=3}^{+\infty} c\alpha_0^2 (M\delta + a) \left( \frac{1}{S_{n-2}^{\frac{1}{2}+\varepsilon_1}} - \frac{1}{S_n^{\frac{1}{2}+\varepsilon_1}} \right) \\
=\ & \frac{\alpha_0}{20} \left( \frac{\left\| \nabla_{\theta_2} g(\theta_2) \right\|^2}{\sqrt{S_1}} - \lim_{n \to +\infty} \frac{\left\| \nabla_{\theta_n} g(\theta_n) \right\|^2}{\sqrt{S_{n-1}}} \right) \\
& + \frac{\alpha_0}{2}(M+1) \left( \frac{\left\| \nabla_{\theta_2} g(\theta_2) \right\|^2}{\sqrt{S_2}} - \lim_{n \to +\infty} \frac{\left\| \nabla_{\theta_n} g(\theta_n) \right\|^2}{\sqrt{S_n}} \right) + \frac{\alpha_0 a(M+1)}{2} \left( \frac{1}{\sqrt{S_2}} - \lim_{n \to +\infty} \frac{1}{\sqrt{S_n}} \right) \\
& + 4M^2 \alpha_0^3 c^2 \sum_{n=2}^{+\infty} \frac{\left\| \nabla_{\theta_{n-1}} g(\theta_{n-1}), \xi_{n-1} \right\|^2}{S_{n-1}^{\frac{3}{2}}} + \frac{(M+1)\alpha_0^3 c^2}{2 S_{n-1}^{\frac{3}{2}}} \sum_{n=2}^{+\infty} \left\| \nabla_{\theta_{n-1}} g(\theta_{n-1}, \xi_{n-1}) \right\|^2 \\
& + \frac{\alpha_0 a(M+1)}{2} \frac{1}{\sqrt{S_2}} + \frac{M\delta + a}{S_2^{\frac{1}{2}+\varepsilon_1}} + \frac{\alpha_0^3 c^2 (M+1)^2 (M\delta + a)}{2 S_1^{\frac{1}{2}+\varepsilon_1}}.
\end{aligned}
$$

From Lemma 15, we get that

$$
\lim_{n \to +\infty} \frac{\left\| \nabla_{\theta_n} g(\theta_n) \right\|^2}{\sqrt{S_{n-1}}} \le \lim_{n \to +\infty} \frac{\left\| \nabla_{\theta_n} g(\theta_n) \right\|^2}{S_{n-1}^{\varepsilon_0}} = 0,
$$

and

$$
\lim_{n \to +\infty} \frac{\left\| \nabla_{\theta_n} g(\theta_n) \right\|^2}{\sqrt{S_n}} \le \lim_{n \to +\infty} \frac{\left\| \nabla_{\theta_n} g(\theta_n) \right\|^2}{\sqrt{S_{n-1}}} \le \lim_{n \to +\infty} \frac{\left\| \nabla_{\theta_n} g(\theta_n) \right\|^2}{S_{n-1}^{\varepsilon_0}} = 0.
$$

It follows that

$$
\begin{aligned}
\sum_{n=3}^{+\infty} P_n =\ & \frac{\alpha_0}{20} \frac{\left\| \nabla_{\theta_1} g(\theta_1) \right\|^2}{\sqrt{S_0}} + \frac{\alpha_0}{2}(M+1) \frac{\left\| \nabla_{\theta_1} g(\theta_1) \right\|^2}{\sqrt{S_1}} + \frac{\alpha_0 a(M+1)}{2} \frac{1}{\sqrt{S_1}} \\
& + 4M^2 \alpha_0^3 c^2 \sum_{n=2}^{+\infty} \frac{\left\| \nabla_{\theta_{n-1}} g(\theta_{n-1}), \xi_{n-1} \right\|^2}{S_{n-1}^{\frac{3}{2}}} + \frac{(M+1)\alpha_0^3 c^2}{2 S_{n-1}^{\frac{3}{2}}} \sum_{n=2}^{+\infty} \left\| \nabla_{\theta_{n-1}} g(\theta_{n-1}, \xi_{n-1}) \right\|^2 \\
& + \frac{\alpha_0 a(M+1)}{2} \frac{1}{\sqrt{S_2}} + \frac{M\delta + a}{S_2^{\frac{1}{2}+\varepsilon_1}} + \frac{\alpha_0^3 c^2 (M+1)^2 (M\delta + a)}{2 S_1^{\frac{1}{2}+\varepsilon_1}}.
\end{aligned}
\tag{138}
$$

By Lemma 16, we have

$$
4M^2 \alpha_0^3 c^2 \sum_{n=3}^{+\infty} \frac{\left\| \nabla_{\theta_{n-1}} g(\theta_{n-1}), \xi_{n-1} \right\|^2}{S_{n-1}^{\frac{3}{2}}} + \sum_{n=3}^{+\infty} \frac{(M+1)\alpha_0^3 c^2}{2 S_{n-1}^{\frac{3}{2}}} \left\| \nabla_{\theta_{n-1}} g(\theta_{n-1}, \xi_{n-1}) \right\|^2 < +\infty.
\tag{139}
$$

Thus, $\sum_{n=2}^{+\infty} P_n$ is convergent. In addition, it holds that

$$\sum_{n=3}^{+\infty} \mathbb{E}\left(\left\|I_n^{\leq a} A_n\right\|^2 \Big| \mathscr{F}_{n-1}\right)$$

$$\leq \sum_{n=3}^{+\infty} \mathbb{E}\left(\|A_n\|^2 \Big| \mathscr{F}_{n-1}\right) \leq \sum_{n=3}^{+\infty} \mathbb{E}\left(\left\|\frac{\alpha_0}{\sqrt{S_{n-1}}}\nabla_{\theta_n}g(\theta_n)^T\left(\nabla_{\theta_n}g(\theta_n) - \nabla_{\theta_n}g(\theta_n,\xi_n)\right)\right\|^2 \Big| \mathscr{F}_{n-1}\right)$$

$$\leq 2\sum_{n=3}^{+\infty} \frac{\alpha_0^2\left\|\nabla_{\theta_n}g(\theta_n)\right\|^4}{S_{n-1}} + 2\sum_{n=3}^{+\infty} \mathbb{E}\left(\frac{\alpha_0^2}{S_{n-1}}\left\|\nabla_{\theta_n}g(\theta_n)\right\|^2\left\|\nabla_{\theta_n}g(\theta_n,\xi_n)\right\|^2 \Big| \mathscr{F}_{n-1}\right).$$

It follows from Lemma 15 that $\exists 0 < \varepsilon_0 < \frac{3}{8}$, such that

$$\lim_{n\to+\infty} \frac{\left\|\nabla_{\theta_n}g(\theta_n)\right\|^2}{S_{n-1}^{\varepsilon_0}} = 0 \ \ a.s.,$$

meaning that there is an almost surely bounded random variable $\delta_0$, such that

$$\frac{\left\|\nabla_{\theta_n}g(\theta_n)\right\|^2}{S_{n-1}^{\varepsilon_0}} < \delta_0 < \infty \ \ a.s.$$

It follows that

$$\frac{\left\|\nabla_{\theta_n}g(\theta_n)\right\|^4}{S_{n-1}} < \delta_0 \frac{\left\|\nabla_{\theta_n}g(\theta_n)\right\|^2}{S_{n-1}^{\frac{1}{2}+(\frac{1}{2}-\varepsilon_0)}} \ \ a.s.,$$

Then we derive that

$$\sum_{n=3}^{+\infty} \mathbb{E}\left(\left\|I_n^{\leq a} A_n\right\|^2 \Big| \mathscr{F}_{n-1}\right)$$

$$\leq 2\sum_{n=3}^{+\infty} \frac{\alpha_0^2\left\|\nabla_{\theta_n}g(\theta_n)\right\|^4}{S_{n-1}} + 2\sum_{n=3}^{+\infty} \mathbb{E}\left(\frac{\alpha_0^2}{S_{n-1}}\left\|\nabla_{\theta_n}g(\theta_n)\right\|^2\left\|\nabla_{\theta_n}g(\theta_n,\xi_n)\right\|^2 \Big| \mathscr{F}_{n-1}\right)$$

$$\leq 2\sum_{n=3}^{+\infty} \frac{\alpha_0^2\left\|\nabla_{\theta_n}g(\theta_n)\right\|^4}{S_{n-1}} + 2\sum_{n=3}^{+\infty} \frac{\alpha_0^2}{S_{n-1}}\left\|\nabla_{\theta_n}g(\theta_n)\right\|^2\left(M\left\|\nabla_{\theta_n}g(\theta_n)\right\|^2 + a\right) \tag{140}$$

$$= 2(M+1)\sum_{n=3}^{+\infty} \frac{\alpha_0^2\left\|\nabla_{\theta_n}g(\theta_n)\right\|^4}{S_{n-1}} + 2\sum_{n=3}^{+\infty} \frac{\alpha_0^2 a\left\|\nabla_{\theta_n}g(\theta_n)\right\|^2}{S_{n-1}}$$

$$< 2\left((M+1)\delta_0 + a\right)\sum_{n=3}^{+\infty} \frac{\left\|\nabla_{\theta_n}g(\theta_n)\right\|^2}{S_{n-1}^{\frac{1}{2}+(\frac{1}{2}-\varepsilon_0)}} < +\infty \ \ a.s..$$

From Lemma 5, $\sum_{n=3}^{+\infty} I_n^{\leq a} A_n$ is convergent almost surely. Similarly, we can prove other parts of $\sum_{n=3}^{+\infty} Q_n$ are also convergent almost surely. Thus, $\sum_{n=3}^{+\infty} Q_n$ is convergent almost surely. By summary, we get

$$\sum_{n=1}^{+\infty} \left(P_n + Q_n\right) \text{ is convergent almost surely.} \tag{141}$$

It is easy to find that $J_i$ is a bounded closed set. So $\forall \varepsilon > 0$ we can construct an open cover $H_\varepsilon^{(i)} = \{U(\theta,\varepsilon)\}$ $(\theta \in J_i)$ of $J_i$. Through the *HeineBorel theorem*, we can get a finite open subcover $\{U(\theta_k,\varepsilon)$ $(k = 0, 1, ..., n)$ from $H_\varepsilon^{(i)}$. Then we assign $U_\varepsilon^{(i)} = \bigcup_{k=0}^n U(\theta_k,\varepsilon)$. We can get $U_\varepsilon^{(i)}$ is a open set. Under Assumption 5, $J = \{\theta | \nabla_\theta g(\theta)\}$ has only finite connected components $J_1, J_2, ..., J_m$. So $\inf_{i\neq j} d(J_i, J_j) = \min_{i\neq j} d(J_i, J_j)$. Let $\delta_0 = \min_{i\neq j} d(J_i, J_j)$. It follows from Lemma 4 that $\exists \varepsilon_0 > 0$, when $d(\theta, J_i) < \varepsilon_0$, there is

$$\|\nabla_\theta g(\theta)\|^2 \leq 2c|g(\theta) - g_i|, \tag{142}$$

where $g_i$ denotes $g(\theta)$ $(\theta \in J_i)$. Let $c = min\{\varepsilon_0, \delta_0/4\}$ and construct $U_c^{(1)}, U_c^{(2)}, ..., U_c^{(m)}$. It is obvious that $\forall U_c^{(i)}, U^{(j)c}$ $(i \neq j)$, $d(U_c^{(i)}, U^{(j)}) > \delta_0/2$, and $\|\nabla_\theta g(\theta)\|^2 \leq 2c|g(\theta) - g_i|$ $(\theta \in U_c^{(i)})$.

Since $J$ is a bounded set, $\exists N > 0$, such that $J \subset K$ ($K$ is the closure of $U(0,N)$). Then we construct a set $M = K / \bigcup_{i=1}^{m} U_c^{(i)}$. Since $U_c^{(i)}$ is a open set and $K$ is a closed set, we conclude $M$ is a closed set. Since $\|\nabla_\theta g(\theta)\|$ is a continuous function, $\exists \theta_0 \in M$, $\|\nabla_{\theta_0} g(\theta_0)\| = \min_{\theta \in M} \|\nabla_\theta g(\theta)\|$. Let $r = \|\nabla_{\theta_0} g(\theta_0)\| > 0$.

Then we prove that $\forall u > 0, \theta \in K$, $\exists \delta > 0$, if $\|\nabla_\theta g(\theta)\| < \delta$, makes $d(\theta, J) < u$. We prove it by contradiction. Assume $\exists u_0 > 0$, $\forall \delta_1 > 0$, $\exists \ \theta_{\delta_1}$ holds $\|\nabla_{\theta_{\delta_1}} g(\theta_{\delta_1})\| < \delta_1$ and $d(\theta, J) \geq u_0$. We make $\delta_1 = 1, 1/2, 1/3...$, and we form a sequence $\{\theta_{1/n}\}$. It is obvious that $\|\nabla_{\theta_{1/n}} g(\theta_{1/n}) \to 0\|$. Since $\{\theta_{1/n}\}$ is bounded, through the *Accumulation point theorem*, there exists a convergent subsequence $\{\theta_{1/k_n}\} \subset \{\theta_{1/n}\}$. We defined $\theta^{(0)} = \lim_{n \to +\infty} \theta_{1/k_n}$. Through the continuity of $d(\theta, J)$ and $\|\nabla_\theta g(\theta)\|$, we get $d(\theta^{(0)}) \geq u_0$ and $\|\nabla_{\theta^{(0)}} g(\theta^{(0)})\| = 0$. It is contradiction by the definition of $J$. So $\forall u > 0, \theta \in K$, $\exists \delta > 0$, $\|\nabla_\theta g(\theta)\| < \delta$, makes $d(\theta, J) < u$ ($\exists_1$, makes $d(\theta, J) < u$). And furthermore, due to the continuity of $g(\theta)$, we can get $\forall \varepsilon_1 > 0$, $\exists \ \delta' > 0$, if $d(\theta, J_i) < \delta'$, there is $|g(\theta) - g_i| < \varepsilon_1$. So combine these two consequences. We can prove $\forall \varepsilon_1 > 0$, $\exists b > 0$, if $\theta \in U_c^{(i)}$ and $\|\nabla_\theta g(\theta)\| < b$, there is $|g(\theta) - g_i| < \varepsilon_1$.

Through equation 121 and Lemma 6 we get there is a subsequence $\{\|\nabla_{\theta_{k_n}} g(\theta_{k_n})\|^2\}$ of $\{\|\nabla_{\theta_n} g(\theta_n)\|^2\}$ which satisfies that

$$\lim_{n \to +\infty} \left\|\nabla_{\theta_{k_n}} g(\theta_{k_n})\right\|^2 = 0 \ \ a.s.. \tag{143}$$

Next we aim to prove $\lim_{n \to +\infty} \|\nabla_{\theta_n} g(\theta_n)\|^2 = 0$. It is equivalent to prove that $\{\|\nabla_n g(\theta_n)\|^2\}$ has no positive accumulation points, that is to say, $\forall e_0 > 0$, there are only finite values of $\|\nabla_{\theta_n} g(\theta)\|$ larger than $e_0$. And obviously, we just need to prove $\forall 0 < e_0 < r$, there are only finite values of $\|\nabla_{\theta_n} g(\theta)\|$ larger than $e$. We prove this by contradiction. We suppose $\exists 0 < e < a$, making the set $S = \{\|\nabla_{\theta_n} g(\theta_n)\|^2\}$ be an infinite set. Then we assign $\varepsilon_1 = e/8c$ and define $o = min\{b, e/4\}$. Due to equation 143, we get there exists a subsequence $\{\theta_{p_n}\}$ of $\{\theta_n\}$ which satisfies $\|\nabla_{\theta_{p_n}} g(\theta_{p_n})\| < o$. We rank $S$ as a subsequence $\{\|\nabla_{m_n} g(\theta_{m_n})\|^2\}$ of $\{\|\nabla_n g(\theta_n)\|^2\}$. Then there is an infinite subsequence $\{\|\nabla_{m_{i_n}} g(\theta_{m_{i_n}})\|^2\}$ of $\{\|\nabla_{m_n} g(\theta_{m_n})\|^2\}$ such that $\forall n \in \mathbb{N}_+$, $\exists l$, $n_{p_n} \in (m_{i_l}, m_{i_{l+1}})$. For convenient, we abbreviate $\{m_{i_n}\}$ as $\{i_n\}$. And we construct another infinite sequence $\{q_n\}$ as follows

$$q_1 = \max\left\{n : p_1 < n < \min\{m_{i_l : m_{i_l} > p_1}\}, \left\|\nabla_{\theta_n} g(\theta_n)\right\| \leq o\right\},$$

$$q_2 = \min\left\{n : n > q_1, \left\|\nabla_{\theta_n} g(\theta_n)\right\| > e\right\},$$

$$q_{2n-1} = \max\left\{n : \min\{m_{i_l} : m_{i_l} > q_{2n-3}\} < n < \min\{m_l : m_l > \min\{m_{i_l} : m_{i_l} > q_{2n-3}\},\right.$$
$$\left\|\nabla_{\theta_n} g(\theta_n)\right\| \leq o\right\},$$

$$q_{2n} = \min\left\{n : n > q_{2n-1}, \left\|\nabla_{\theta_n} g(\theta_n)\right\| > e\right\}.$$

Now we prove that $\exists N_0$, when $q_{2n} > N_0$, it has $e < \left\|\nabla_{\theta_{q_{2n}}} g(\theta_{q_{2n}})\right\| < r$. The left side is obvious (the definition of $q_{2n}$). And for the right side, we know $\left\|\nabla_{\theta_{q_{2n}-1}} g(\theta_{q_{2n}-1})\right\| \leq e$. It follows from equation 5 that

$$\|\theta_{n+1} - \theta_n\|^2 = \frac{\alpha_0^2}{S_n} \left\|\nabla_{\theta_n} g(\theta_{q_n}, \xi_n)\right\|^2$$

$$\leq \frac{\alpha_0^2}{S_{n-1}} \left(\left\|\nabla_{\theta_{n-1}} g(\theta_n, \xi_n)\right\|^2 - \mathbb{E}\left(\left\|\nabla_{\theta_n} g(\theta_n, \xi_n)\right\|^2 \big| \mathscr{F}_n\right)\right)$$

$$+ \frac{\alpha_0^2}{S_{n-1}} \left(M\left\|\nabla_{\theta_n} g(\theta_n)\right\|^2 + a\right).$$

Through previous consequences we can easily find that

$$\sum_{n=2}^{+\infty} \left(\frac{\alpha_0^2}{S_{n-1}} \left(\left\|\nabla_{\theta_{n-1}} g(\theta_n, \xi_n)\right\|^2 - \mathbb{E}\left(\left\|\nabla_{\theta_n} g(\theta_n, \xi_n)\right\|^2 \big| \mathscr{F}_n\right)\right) + \frac{\alpha_0^2 M \left\|\nabla_{\theta_n} g(\theta_n)\right\|^2}{S_{n-1}}\right)$$
$$< +\infty \ \ a.s..$$

Note that $\alpha_0^2 a / S_{n-1} \to 0$, $\ \ a.s..$ We conclude

$$\|\theta_{n+1} - \theta_n\| \to 0 \ \ a.s.. \tag{144}$$

Through Assumption 5 2) we get $\left|\|\nabla_{\theta_{n+1}}g(\theta_{n+1})\|^2 - \|\nabla_{\theta_n}g(\theta_n)\|^2\right| \leq \left|\|\nabla_{\theta_{n+1}}g(\theta_{n+1})\| - \|\nabla_{\theta_n}g(\theta_n)\|\right|^2 \leq \|\nabla_{\theta_{n+1}}g(\theta_{n+1}) - \nabla_{\theta_n}g(\theta_n)\|^2 \leq c\|\theta_{n+1} - \theta_n\| \to 0 \quad a.s.$, So $\exists N_0$, when $n > N_0$, there is $\left|\|\nabla_{\theta_{n+1}}g(\theta_{n+1})\|^2 - \|\nabla_{\theta_n}g(\theta_n)\|\right| < r - e$. Then we can get that when $q_{2n} > N_0 + 1$, there is $\left\|\nabla_{\theta_{q_{2n}}}g(\theta_{q_n})\right\| \leq \left\|\nabla_{\theta_{q_{2n}-1}}g(\theta_{q_{2n}-1})\right\| + \left\|\nabla_{\theta_{q_{2n}}}g(\theta_{q_{2n}})\| - \|\nabla_{\theta_{q_{2n}-1}}g(\theta_{q_{2n}-1})\|\right\| \leq e + r - e = r.$ That means that $\theta_{q_{2n}} \in \bigcup_{i=1}^m U_\tau^{(i)}$, so we can prove $\exists i_0$, such that $\theta_{q_{2n}} \in U_\tau^{(i_0)}$. And due to $\|\nabla_{\theta_n}g(\theta_n)\| \leq e < r \quad (n \in [q_{2n-1}, q_{2n}))$, we get $\forall k \in [q_{2n-1}, q_{2n})$, $\exists i_k$, such that $\theta_n \in U_\tau^{(i_k)}$. Due to $\|\theta_{n+1} - \theta_n\| \to 0 a.s.$, we know $i_0 = i_k \quad (\forall j \in [q_{2n-1}, q_{2n}))$. For convenient, we sign $i_0 = i_{q_{2n-1}} = ... = i_{q_{2n}-1} = i_{q_{2n}}$. And then we can conclude that

$$\|\nabla_\theta g(\theta_n)\|^2 \leq 2c|g(\theta_n) - g_{i_{q_{2n}}}| \quad (n \in [q_{2n-1}, q_{2n}]).$$

Due to locally sign-preserving property, we get

$$\|\nabla_\theta g(\theta_n)\|^2 \leq 2c(g(\theta_n) - g_{i_{q_{2n}}}) \quad (g(\theta_n) \geq g_{i_{q_{2n}}}) \quad or$$
$$\|\nabla_\theta g(\theta_n)\|^2 \leq -2c(g(\theta_n) - g_{i_{q_{2n}}}) \quad (g(\theta_n) \leq g_{i_{q_{2n}}}) \quad (n \in [q_{2n-1}, q_{2n}]).$$

Ways to dispose these two cases is same, so we just show how to prove the first case. We get

$$e - o < \left\|\nabla_{\theta_{q_{2n}}}g(\theta_{q_{2n}})\right\|^2 - \left\|\nabla_{\theta_{q_{2n-1}}}g(\theta_{q_{2n-1}})\right\|^2 < 2c\left(g(\theta_{q_{2n}}) - g_{i_{q_{2n}}}\right) - \left\|\nabla_{\theta_{q_{2n-1}}}g(\theta_{q_{2n-1}})\right\|^2$$
$$= \left(2c\sum_{i=0}^{q_{2n}-q_{2n-1}-1}g(\theta_{q_{2n-1}+i+1}) - g(\theta_{q_{2n-1}+i})\right) + 2c\left(g(\theta_{q_{2n-1}}) - g_{i_{q_{2n}}}\right) - \left\|\nabla_{\theta_{q_{2n-1}}}g(\theta_{q_{2n-1}})\right\|^2.$$

From equation 137, we obtain

$$g(\theta_{q_{2n-1}+i+1}) - g(\theta_{q_{2n-1}+i})$$
$$\leq \left(c\alpha_0^2 + \frac{c^3\alpha_0^2(M+1)}{2}\right)(M\delta + a)\frac{1}{S_{q_{2n-1}+i}^{\frac{1}{2}+\varepsilon_1}} + P_{q_{2n-1}+i} + Q_{q_{2n-1}+i}.$$

So there is

$$e - o < \sum_{i=0}^{q_{2n}-q_{2n-1}-1}\frac{L}{S_{q_{2n-1}+i}^{\frac{1}{2}+\varepsilon_1}} + \sum_{i=0}^{q_{2n}-q_{2n-1}-1}\left(P_{q_{2n-1}+i} + Q_{q_{2n-1}+i}\right)$$
$$+ 2c\left(g(\theta_{q_{2n-1}}) - g_{i_{2n-1}}\right) - \left\|\nabla_{\theta_{q_{2n-1}}}g(\theta_{q_{2n-1}})\right\|^2, \tag{145}$$

which

$$L = 2c\left(c\alpha_0^2 + \frac{c^3\alpha_0^2(M+1)}{2}\right)(M\delta + a).$$

Due to $\|\nabla_{\theta_{q_{2n-1}}}g(\theta_{q_{2n-1}})\|^2 < o < b$, so through equation 141, we get that $g(\theta_{q_{2n-1}}) - g_{i_{2n-1}} < e/8c$. Substitute it into equation 146. We get

$$\sum_{i=0}^{q_{2n}-q_{2n-1}-1}\frac{1}{S_{q_{2n-1}+i}^{\frac{1}{2}+\varepsilon_1}} > \frac{e}{2L} - \sum_{i=0}^{q_{2n}-q_{2n-1}-1}\left(P_{q_{2n-1}+i} + Q_{q_{2n-1}+i}\right). \tag{146}$$

Through equation 141, we know $\sum_{n=1}^{+\infty}(P_n + Q_n)$ is convergence almost surely. So we get that $\sum_{i=0}^{q_{2n}-q_{2n-1}-1}\left(P_{q_{2n-1}+i} + Q_{q_{2n-1}+i}\right) \to 0 \quad a.s.$ by the *Cauchy's test for convergence*. Combining $1/S_{q_{2n-1}+i}^{\frac{1}{2}+\varepsilon_1} \to 0 \quad a.s.$, we get

$$\sum_{i=1}^{q_{2n}-q_{2n-1}-1}\frac{1}{S_{q_{2n-1}+i}^{\frac{1}{2}+\varepsilon_1}}$$
$$> \frac{e}{2L} - \frac{1}{S_{q_{2n-1}}^{\frac{1}{2}+\varepsilon_1}} - \sum_{i=0}^{q_{2n}-q_{2n-1}-1}\left(P_{q_{2n-1}+i} + Q_{q_{2n-1}+i}\right) \to \frac{e}{2L} \quad a.s., \tag{147}$$

so there is

$$\sum_{n=1}^{+\infty} \left( \sum_{i=1}^{q_{2n}-q_{2n-1}-1} \frac{1}{S_{q_{2n-1}+i}^{\frac{1}{2}+\varepsilon_1}} \right) = +\infty \ \ a.s.. \tag{148}$$

But on the other hand, we know $\|\nabla_{\theta_{q_{2n-1}+i}} g(\theta_{q_{2n-1}+i})\| > o \ \ (i > 0)$. Together with equation 121, we get

$$\sum_{n=1}^{+\infty} \left( \sum_{i=1}^{q_{2n}-q_{2n-1}-1} \frac{1}{S_{q_{2n-1}+i}^{\frac{1}{2}+\varepsilon_1}} \right) < \frac{1}{o} \sum_{n=1}^{+\infty} \left( \sum_{i=1}^{q_{2n}-q_{2n-1}-1} \frac{\left\| \nabla_{\theta_{q_{2n-1}+i}} g(\theta_{q_{2n-1}+i}) \right\|^2}{S_{q_{2n-1}+i}^{\frac{1}{2}+\varepsilon_1}} \right)$$

$$< \frac{1}{o} \sum_{n=3}^{n} \frac{\left\| \nabla_{\theta_n} g(\theta_n) \right\|^2}{S_{n-1}^{\frac{1}{2}+\varepsilon}} < +\infty \ \ a.s.. \tag{149}$$

It contradicts with equation 148, so we get that $\|\nabla_{\theta_n} g(\theta_n)\| \to 0 \ \ a.s..$ Combining equation 133, we get $\|\nabla_{\theta_n} g(\theta_n)\| \to 0$ no matter $S_n < +\infty \ a.s.$ or $S_n = +\infty$. Under Assumption 5 1), it is safe to conclude that there exists a connected component $J^*$ of $J$ such that $\lim_{n \to \infty} d(\theta_n, J^*) = 0$.

