# OpenReview forum: "On the Convergence of mSGD and AdaGrad for Stochastic Optimization"
_ICLR.cc/2022/Conference — ICLR 2022 Poster_

### Official Review · Reviewer_qmsj · 2021-11-02

**Correctness:** 4
**Technical Novelty And Significance:** 3
**Empirical Novelty And Significance:** 3
**Recommendation:** 6
**Confidence:** 3

**Main Review:**

The paper appears to use a different template other than ICLR's. I encourage the authors to double check this. The current template makes the paper harder to read at least for me.

Strengths:
- Theorem 1 presents asymptotic convergence which complements existing results in the literature. Convergence in theorem 2 is for metric depending on last iterate which is stronger compared to time-averaged metric.
- The convergence on expectation of squared norm gradient on last iterate shows that mSGD is better than SGD.

Weaknesses:
- The clarity of the paper is not good, the proofs in the appendix are highly technical and hard to follow. I would recommend to have an outline to better understand how each lemma play a role in the main convergence analysis. I have gone through the proofs and find no problem but there is a chance I miss something.

Comments:
- What is the motivation of using assumption 4 or 5 rather than assuming convexity. As these assumptions are not standard, it is better to include examples of classes of function that satisfy these assumptions.

**Summary Of The Paper:**

The paper analyzes the convergence of momentum-based SGD (mSGD) and AdaGrad in the sense of asymptotic convergence rather than time average or subsequence convergence in existing results. The analysis shows that mSGD is better than SGD in terms of expectation of squared gradient norms on last iterate.

**Summary Of The Review:**

The paper contains solid contribution in terms of theoretical analysis, especially for mSGD. The convergence results are on last iterate different from existing results. However, given the currents state of the submission, the clarity of the paper needs to be significantly improved. Adding proof outline and providing more explanations for key steps definitely helps to improve the quality of the paper.

---

> ### Author Response · Authors · 2021-11-15
> **Response to Official Review of Paper3772 by Reviewer qmsj**
>
> We thank the reviewer for the elaborate reading and the suggestions. We have revised the manuscript based on your comments. Major changes in the new version are highlighted in blue. We shall respond to your comments point by point in what follows.
>
> Main Review: The paper appears to use a different template other than ICLR's. I encourage the authors to double check this. The current template makes the paper harder to read at least for me.
>
> RESPONSE: We have changed the template now in the new version.
>
> Weaknesses: The clarity of the paper is not good, the proofs in the appendix are highly technical and hard to follow. I would recommend to have an outline to better understand how each lemma play a role in the main convergence analysis. I have gone through the proofs and find no problem but there is a chance I miss something.
>
> RESPONSE: Thanks a lot for this valuable suggestion. In the new version, we have added a proof sketch of Theorems 1 and 2 in Section B.1, and a proof sketch of Theorem 3 in Section C.1.
>
> Comments: What is the motivation of using assumption 4 or 5 rather than assuming convexity. As these assumptions are not standard, it is better to include examples of classes of function that satisfy these assumptions.
>
> RESPONSE: In many problems of machine learning, especially  deep learning, because of strong non-linearity mapping from input data to output data and the structure complexity of employed models, loss functions could be non-convex and may have multiple local critical points. Thus, it is important to study the performance of optimization algorithms for non-convex loss functions.
>
> In  the revised version, we have added more discussions after Assumption 4 to clarify the motivation for the given conditions.  In addition, we have added some examples of loss functions, after Assumption 5, that are not convex but satisfies Assumption 5 1), such as $\sin^{2}{(x)}$, $(x-1)(x-2)(x-3)(x-4)$, and $y=\cos^{2}{(x)}$.

---

> > ### Comment · Reviewer_qmsj · 2021-11-20
> > **Keeping my score**
> >
> > Thank you for clarifying my concerns. After reading your response and comments from other reviewers, I believe the paper does contribute to the literature of stochastic gradient methods. I decide to keep my score.

---

### Official Review · Reviewer_MDU9 · 2021-11-02

**Correctness:** 2
**Technical Novelty And Significance:** 3
**Empirical Novelty And Significance:** Not applicable
**Recommendation:** 6
**Confidence:** 4

**Main Review:**

I read some parts of the proof carefully, and it seems that there are some mistakes that should be resolved. I also have several suggestions for the presentation of the paper.

Regarding the presentation: (* are the major concerns)

1. In the optimization literature, the term "estimate sequence" refers to a sequence of auxiliary functions, which were first used in the analysis of Nesterov's accelerated gradient descent. See Definition 2.2.2 of the following textbook.

Nesterov, Yurii. Introductory lectures on convex optimization: A basic course. Vol. 87. Springer Science & Business Media, 2003.

Please consider just using "iterates" to refer to the \theta_n in the paper.

2*. The updating rules of mSGD and SHB in equations 3 and 4 are actually equivalent. This should be mentioned in the paper, and it is inappropriate to say that "the results in Liu et al (2020) do not apply to mSGD."

For example, to get equ 4 from 3, we have

$v_n = \alpha v_{n-1} + \varepsilon \nabla_n,  \theta_{n+1} = \theta_n - v_n$

$\frac{1}{\alpha + \varepsilon} v_n = \frac{\alpha}{\alpha + \varepsilon} v_{n-1} + \frac{\varepsilon}{\alpha + \varepsilon} \nabla_n, \theta_{n+1} = \theta_n - (\alpha+ \varepsilon) * \frac{1}{\alpha + \varepsilon} v_n$

To get equ 4, we just need to set $\frac{1}{\alpha + \varepsilon} v_n$ as $v_n$ and $\beta$ as $\frac{\alpha}{\alpha + \varepsilon}$. All the parameters can also depend on $n$.

3*. It would be much better if simple proof sketches of theorems 1, 2, and 3 can be added. For example, how is the milder assumption in Assumption 2 4) handled in the analysis? How is the difficulty mentioned in equ (11) handled? Otherwise, readers have to dive into the details of proof to figure them out.

4. In Assumption 4 1), $\varepsilon_1$ has already been used as stepsizes.

5. Assumption 4 2) is very similar to the Polyak-Lojasiewicz condition, this should be mentioned.

6*. It is inappropriate to say that "We can observe that the convergence rate of mSGD is quicker than that of SGD in this sense." In equation (78), It is unclear why the bound $O((\frac{1}{2-\alpha})^n)$ is relaxed to the bound presented in Theorem 1. Looking at $O((\frac{1}{2-\alpha})^n)$, a larger momentum $\alpha \rightarrow 1^-$ actually gives a worse bound.

7. Under Assumption 5, why is "condition 1)" is relatively weak?

Regarding the proof:

I haven't checked all the proof in detail, here are some problems that need to be clarified:

1. Lemma 3 is a standard result, see Lemma 1.2.3 of the textbook mentioned above.

2. In Lemma 4, it seems that the S_i needs to be bounded, as used in the proof.

3. In Lemma 4, $\varepsilon_0$ is actually defined in the proof, this should be clarified.

4. In the second last equation on page 14, the last term should be integrated to t', not x.

5* The last equation on page 14 is incorrect, how is x_2 canceled? It seems that t'' > 0 is needed, why is this true?

6* In equ 17, why is $|g'(t')| = ||\nabla f(x_0)||$? Specifically, we have $g'(t') = \frac{x_0 - x_2}{\|x_0 - x_2\|}\cdot \nabla f(x_0)$. Therefore, $|g'(t')| = ||\nabla f(x_0)||$ if and only if $x_0- x_2$ is parallel to $\nabla f(x_0)$, why is this the case?

7* In the second last equality of the proof of lemma 4, how is the term $g'(t'')$ handled? The equality holds if and only if $g'(t'') = g'(t')$, why is this true?


**Summary Of The Paper:**

This paper focuses on developing asymptotic convergence of mSGD and Adagrad, which is stronger than the existing results on subsequence convergence and iterate average convergence. The assumptions are similar or weaker than previous works.

**Summary Of The Review:**

In summary, the issues mentioned above must be resolved. In my opinion, there are obvious mistakes in the proof.

---

> ### Author Response · Authors · 2021-11-15
> **Response to Official Review of Paper3772 by Reviewer MDU9**
>
> We thank the reviewer for the elaborate reading and the suggestions. We have checked the manuscript carefully, and have corrected the mistakes. Major changes in the new version are highlighted in blue. In the following, we shall respond to your comments point by point. The original comments are omitted due to space limitation.
>
> Response to comments regarding the presentation:
> 1. The reviewer seems to refer to Definition 2.2.1 of the text book, since Definition 2.2.2 is to define a convex set. With the suggestion of the reviewer, we have replaced the term "estimate sequence" by "iterates" in the revised version.
>
> 2. As the reviewer pointed out, the update rules of mSGD and SHB are equivalent. We have revised the statement, emphasizing that our condition on step sizes is different from theirs. In the new version, we have discussed the difference in detail on page 5.
>
> 3. Thanks a lot for this valuable suggestion. In the new version, we have added a proof sketch of Theorems 1, 2 in Section B.1 and a proof sketch of Theorem 3 in Section C.1.  In addition, the two questions have been addressed in the proof sketches.
>
> 4. We have replaced $\varepsilon_1$ with $\varepsilon^{\prime}$ in   Assumption 4 1).
>
> 5. We have added a remark in the new version to state that Assumption 4 2) is a local P-L condition.
>
> 6. We have rephrased the statement after Theorem 2 in the new version. Note that for $W(\alpha)=e^{-\sum_{i=1}^{n} s\epsilon_{i}/(p(1-\alpha)^{2})}$ and $0\leq \alpha_1\leq \alpha_2<1$, $W(\alpha_2)\leq W(\alpha_1)$ holds.
>
> The proof of Theorem 2 shows how $O((\frac{1}{2-\alpha})^n)$ is relaxed to the bound $O(W(\alpha))$. Let us discuss it in more details here. It suffices to prove that, for a fixed $\alpha\in[0,1)$, $\exists N(\alpha)$ depending on $\alpha$, s.t. $\forall n>N(\alpha) $ $$\bigg(\frac{1}{2-\alpha}\bigg)^{n}<e^{-\sum_{i=1}^{n} s\epsilon_{i}/(p(1-\alpha)^{2})},~(*)$$
> or equivalently, $$-n\ln(2-\alpha)<-\frac{s}{p(1-\alpha)^{2}}\sum_{i=1}^{n}\epsilon_{i}. $$   Since $\alpha\in [0,1)$, $\ln{(2-\alpha)}>0$, it suffices to prove, $\forall n>N(\alpha)$, $$\sum_{i=1}^{n}\epsilon_{i}<\frac{p(1-\alpha)^{2}\ln{(2-\alpha)}}{s}n.$$ From $\sum_{i=1}^{+\infty}\epsilon_{i}^{2}<+\infty$, we know $\epsilon_{i}\rightarrow \ 0$. So there is a constant $N_{0}(\alpha)$, s.t. $\epsilon_{i}<\frac{p(1-\alpha)^{2}\ln{(2-\alpha)}}{2s}, i>N_{0}(\alpha)$. Hence, $\forall n>N_{0}(\alpha)$, $$\sum_{i=1}^{n}\epsilon_{i}<r(n)=\frac{p(1-\alpha)^{2}\ln{(2-\alpha)}}{2s}(n-N_{0}(\alpha))+\sum_{i=1}^{N_{0}(\alpha)}\epsilon_{i}.$$ It holds that $$r(n)<\frac{p(1-\alpha)^{2}\ln{(2-\alpha)}}{s}n,$$ for all $$n >\bigg[\frac{2s}{p(1-\alpha)^{2}\ln{(2-\alpha)}}\sum_{i=1}^{N_{0}(\alpha)}\epsilon_{i}-N_{0}(\alpha)\bigg]+1.$$ Let $N(\alpha)$ be the maximum of $N_{0}(\alpha)$ and $[\frac{2s}{p(1-\alpha)^{2}\ln (2-\alpha)}\sum_{i=1}^{N_{0}(\alpha)}\epsilon_{i}-N_{0}(\alpha)]+1,$ and we have that, $\forall n>N(\alpha)$, equation (\*) holds. So $\big(\frac{1}{2-\alpha})^{n} = O(e^{-\sum_{i=1}^{n} s\epsilon_{i}/(p(1-\alpha)^{2})})$.
>
> The reviewer may wonder why when $\alpha\rightarrow 1^{-}$, $(\frac{1}{2-\alpha})^{n}\to 1$, but $e^{-\sum_{i=1}^{n} s\epsilon_{i}/(p(1-\alpha)^{2})}\to 0$, which does not coincide with $$\lim_{\alpha\rightarrow 1^{-}}\limsup_{n\rightarrow+\infty}\frac{\big(\frac{1}{2-\alpha})^{n}}{e^{-\sum_{i=1}^{n} s\epsilon_{i}/(p(1-\alpha)^{2})}} \le 1.$$
>
> This fact exists because $N(\alpha)$ is not uniform bounded on $\alpha \in [0,1)$. Hence, $$\lim_{\alpha\rightarrow 1^{-}}\limsup_{n\rightarrow+\infty}\frac{\big(\frac{1}{2-\alpha})^{n}}{e^{-\sum_{i=1}^{n} s\epsilon_{i}/(p(1-\alpha)^{2})}}\neq \limsup_{n\rightarrow+\infty}\lim_{\alpha\rightarrow 1^{-}}\frac{\big(\frac{1}{2-\alpha})^{n}}{e^{-\sum_{i=1}^{n} s\epsilon_{i}/(p(1-\alpha)^{2})}} = +\infty.$$
>
> 7. This is because condition 1) does not require any convexity of the loss function or global conditions as P-L condition.  We have added more discussions in the new version to support this statement.
>
> Response to comments regarding the proof:
>
> 1. We have cited the textbook and removed the proof for this lemma in the new version.
>
> 2. We have added the assumption that $S$ is bounded in Lemma 4 of the new version. Note that Lemma 4 is used in the proof of Theorem 3, and Theorem 3 assumes the boundedness of $S$ (i.e., Assumption 5 1)).
>
> 3. We have changed the notation in the new version.
>
> 4. We have corrected this typo in the new version.
>
> 5. We have discussed the range of $t^{\prime\prime}$ in the proof of Lemma 4 in the new version.
>
> 6. We have revised the proof in the new version. In the revised proof, we construct a line parallel to $\nabla f(x_{0})$. In this way, there is no need to show that $x_0 - x_2$ is parallel to $\nabla f(x_0)$.
>
> 7. Thank you for the elaborate reading. We have fixed the mistakes and revised the proof of Lemma 4 in the new version.
>
> We hope that we have addressed your comments, and we believed that the modifications have improved our paper considerably.

---

> > ### Comment · Reviewer_MDU9 · 2021-11-19
> > **Regarding my question 6**
> >
> >
> > Hello,
> >
> > Thanks for your reply to my questions. Let me clarify more on my question 6.
> >
> > What I would like to ask is, if we have already got a bound of $O((\frac{1}{2-\alpha})^n)$, why do we want to relax it to $O(W(\alpha))$? We cannot just relax it to $O(W(\alpha))$ so that "mSGD is faster than SGD".
> >
> > Instead, we should look at the original bound $O((\frac{1}{2-\alpha})^n)$, which is actually larger when the momentum parameter $\alpha$ is larger.
> >
> > In order to say that "mSGD is faster than SGD", we should get a tighter bound(but not a weaker bound) than $O((\frac{1}{2-\alpha})^n)$, and it should be smaller for a larger $\alpha$.

---

> > > ### Author Response · Authors · 2021-11-19
> > > **Response to the further comment on question 6 of Reviewer MDU9**
> > >
> > > Thanks for your further comment. Sorry that we misunderstood your question.
> > >
> > > In the proof, we studied two different cases, namely,
> > > $$\sum\_{t=1}^{n}(\frac{1}{2-\alpha})^{-t} \mathbb{E}(\|\nabla_{\theta\_{t+1}}g(\theta\_{t+1})\|^{2})=+\infty, (*)$$ and $$\sum\_{t=1}^{n}(\frac{1}{2-\alpha})^{-t} \mathbb{E}(\|\nabla_{\theta\_{t+1}}g(\theta\_{t+1})\|^{2})<+\infty. (**)$$
> > >
> > > In case (*), we can get the uniform bound $O(W(\alpha))$ directly. In case (**), we get a bound $O(\frac{1}{(2-\alpha)^n})$ first and then prove this bound is smaller than $O(W(\alpha))$. Thus we obtain the uniform bound $O(W(\alpha))$ which covers the two cases.
> > >
> > > It is reasonable to use the uniform upper bound covering all cases to measure the performance of one optimization algorithm, although the algorithm can perform well in special cases, such as its initial point close to true parameter vector. As we can see that most comparisons from the literature  are conducted based on upper bounds.
> > >
> > > In our proof, case ($\*\*$) is the special case and it can only be achieved in particular scenarios.  For example, if we consider SGD  for convex loss functions, we can prove that case ($\*\*$)  occurs only when the algorithm converges in  finite time, which however is difficult to achieve in general scenarios of SGD algorithms. The proof is given as follows.
> > >
> > > Consider SGD for a convex loss function $g(\theta)$. Suppose SGD does not coverge in finite time, i.e., $g(\theta\_{n}) \neq g\^{\*}$ for any finite step $n$. Then it follows from the setting in the paper that $g(\theta_n)> g^\*$. Thus, we have $g(\theta\_{n})> g\^{\*}$ for any finite step $n$.
> > > Note that
> > > $$g(\theta\_{n+1})-g(\theta\_{n})\geq\nabla_{\theta\_{n}}g(\theta\_{n})^{T}(\theta\_{n+1}-\theta\_{n})=-\epsilon\_{n}\nabla\_{\theta\_{n}}g(\theta\_{n},\xi\_{n})^{T}\nabla\_{\theta}g(\theta\_{n}).$$
> > > It follows that
> > > $$\mathbb{E}{\big(g(\theta\_{n+1})}-g\^{\*}\big)-\mathbb{E}{\big(g(\theta\_{n})-g\^{\*}\big)}\geq -\epsilon\_{n}\mathbb{E}(\|\nabla\_{\theta\_{n}}g(\theta\_{n})\|^{2}).$$
> > > Note that $\mathbb{E}{g(\theta\_{n})}\neq g\^{\*}$ (due to $g(\theta\_{n})> g\^{\*}$), then it holds that
> > > $$\mathbb{E}\big({g(\theta\_{n+1})}-g^{\*}\big)\geq \bigg(1-\frac{\mathbb{E}(\|\nabla\_{\theta\_{n}}g(\theta\_{n})\|^{2})\epsilon\_{n}}{\mathbb{E}{\big(g(\theta\_{n})-g\^{\*}\big)}}\bigg)\mathbb{E}{\big(g(\theta\_{n})-g\^{\*}\big)}.$$
> > > It follows from Lemma 10 that
> > > $$\mathbb{E}\big({g(\theta\_{n+1})}-g\^{\*}\big)\geq (1-2c\epsilon\_{n}) \mathbb{E}{\big(g(\theta\_{n})-g\^{\*}\big)}\geq \bigg(\prod\_{k=1}^{n}(1-2c\epsilon\_{k})\bigg)(\mathbb{E}(g(\theta\_{1}))-g\^{\*}).$$
> > > Since  $\epsilon\_{n}\rightarrow 0$,
> > > it holds that
> > > $$\prod\_{k=1}^{n}(1-2c\epsilon\_{k})>\bigg(\frac{1}{2}\bigg)^{n},$$ for sufficiently large $n$. Then we have $\sum\_{n=1}^{+\infty}{\mathbb{E}\big({g(\theta\_{n})}-g^{\*}\big)}2^n\geq \sum\_{n=1}^{+\infty}1=+\infty$. It follows from Assumption 4 2) that $\sum\_{n}^{+\infty}{\mathbb{E}\big(\|\nabla\_{\theta\_{n}}g(\theta\_n)\|^{2}\big)}2^n=+\infty.$

---

> > > > ### Comment · Reviewer_MDU9 · 2021-11-19
> > > > **Thanks a lot for the clarification!**
> > > >
> > > >
> > > > Hello,
> > > >
> > > > The mSGD proof looks correct to me now. The proof of Adagrad also looks fine to me, but I haven't checked all the details.
> > > >
> > > > I have raised my score to 6.

---

> ### Comment · Reviewer_MDU9 · 2021-11-19
> **The definition of $S'_i$ and $S''_i$**
>
>
> It seems that the definition of $S'_i$ and $S''_i$ should exclude $S_i$, as we don't want $\nabla f(x_0) = 0$.

---

> > ### Author Response · Authors · 2021-11-19
> > **Response to Reviewer MDU9's question about the definition of $S_{i}^{\prime}$ and $S_{i}^{\prime\prime}$**
> >
> > With due respect to the reviewer, the case that $x_0$ belongs to $S_i$ is a special case in our proof without violating the derivation. In other words, we do not need the assumption that $\nabla f(x_0)\neq 0$ in the proof.

---

### Official Review · Reviewer_ecdk · 2021-11-03

**Correctness:** 4
**Technical Novelty And Significance:** 3
**Empirical Novelty And Significance:** Not applicable
**Recommendation:** 6
**Confidence:** 2

**Main Review:**

PAPER SUMMARY: This paper presents new proofs to the convergence of momentum-SGD and AdaGrad. The new theoretical results show they both converge (in the sense of the estimate sequence). Also, the momentum-SGD converges at a faster rate than SGD (with additional assumption on the properties of the loss function used).

NOVELTY & SIGNIFICANCE: momentum-SGD and AdaGrad are extensively used algorithms in practice. Providing further understanding of their properties is always a good direction to go. This paper proves that under some mild assumptions, momentum-SGD can converges faster than SGD (with additional assumption on the properties of the loss function used). This seems to be a new result for nonconvex objectives (for convex/strongly convex ones we already have similar results). For AdaGrad, the paper proves that the estimate sequence itself is convergent, but not convergence rate is provided. Such result for AdaGrad seems a bit weak. In the theoretical aspect, this paper paves new ways to prove the convergence for such algorithms.

TECHNICAL SOUNDNESS: The theoretical results and assumptions seem legitimate and reasonable. But I have not checked the proof details.


**Summary Of The Paper:**

New convergence results for momentum-SGD and AdaGrad

**Summary Of The Review:**

The paper provies new convergence results for momentum-SGD and AdaGrad, showing that momentum-SGD converges faster than SGD and AdaGrad's estimate sequence converges. The momentum-SGD proof is beneficial for understanding why momentum works in nonconvex objectives, which is still a not well understood topic.

---

> ### Author Response · Authors · 2021-11-15
> **Response to Official Review of Paper3772 by Reviewer ecdk**
>
> Many thanks for the careful reading and recommendation. We appreciate your comments, which have helped improve our work significantly. Major changes in the new version are highlighted in blue.
>
> Regarding the AdaGrad, the study of its asymptotic properties is important but rather challenging, as shown in the appendix where we have made a lot of efforts to prove the convergence of the iterates. We believe both the developed results and the used techniques in the proofs are helpful for a better understanding of AdaGrad.
>
> We agree with the reviewer that convergence rate of AdaGrad is also important, but this is a very challenging problem. Thus, we would like to treat is as our future work as stated at the end of the revised version. As the reviewer pointed out, this paper has made several contributions in the study of mSGD and AdaGrad, which make the paper sufficient to be presented at this conference.
>
> We hope that we have addressed your comments, and we believed that the modifications have improved our paper considerably.

---

### Comment · Area_Chair_GkmD · 2021-11-15
**Some missing references**

Dear Authors,

There are some important missing references.

Important ones:

- Sébastien Gadat, Ioana Gavra. Asymptotic study of stochastic adaptive algorithm in non-convex landscape. https://arxiv.org/abs/2012.05640
It seems very relevant to your work and it should be clearly discussed if and how it differs from your results.

- Othmane Sebbouh, Robert M Gower, Aaron Defazio. On the (asymptotic) convergence of Stochastic Gradient Descent and Stochastic Heavy Ball. COLT 2021. http://proceedings.mlr.press/v134/sebbouh21a/sebbouh21a.pdf

This deals with SGD with momentum and a different class of adaptive stepsizes

Less important ones:

- Before equation (5), you cite (Ward et al., 2019; Duchi et al., 2011) for the particular form of stepsizes you consider. However, both citations are inexact. First, Duchi et al. only considered coordinate-wise stepsizes, not the norm version you consider. The norm version was actually proposed in

M. Streeter and H. B. McMahan. Less regret via online conditioning, 2010. https://arxiv.org/abs/1002.4862

and later proposed as a general scheme in

F. Orabona and D. Pál. Scale-free algorithms for online linear optimization. In International Conference on Algorithmic Learning Theory, 2015. https://arxiv.org/abs/1502.05744

Moreover, even if many other authors called them "AdaGrad" stepsizes, another name should be used to distinguish the norm version from the coordinate-wise version.


- X. Li and F. Orabona. A High Probability Analysis of Adaptive SGD with Momentum. Workshop on Beyond First Order Methods in ML Systems at ICML 2020 https://arxiv.org/abs/2007.14294

High probability analysis of SGD with momentum, even with AdaGrad stepsizes (coordinate-wise), but without current gradient

- H. B. McMahan and M. J. Streeter. Adaptive bound optimization for online convex optimization. In COLT, 2010

This paper also introduces exactly the same algorithm of AdaGrad, at the very same conference of the Duchi et al. paper. It is often forgotten for no good reasons.

---

> ### Author Response · Authors · 2021-11-17
> **Response to Area Chair GkmD**
>
> Dear Area Chair,
>
> First of all, we thank you for introducing  these related references, which have been cited and discussed in the updated version.  For the ease of tracking, the modifications in the new version are marked in red. We shall respond to your comments below.
>
> COMMENT 1:
>
> There are some important missing references. Important ones:
>
> [1] Sébastien Gadat, Ioana Gavra. Asymptotic study of stochastic adaptive algorithm in non-convex landscape. https://arxiv.org/abs/2012.05640
>
> [2] Othmane Sebbouh, Robert M Gower, Aaron Defazio. On the (asymptotic) convergence of Stochastic Gradient Descent and Stochastic Heavy Ball. COLT 2021. http://proceedings.mlr.press/v134/sebbouh21a/sebbouh21a.pdf
>
> Paper [1] seems very relevant to your work and it should be clearly discussed if and how it differs from your results. Paper [2] deals with SGD with momentum and a different class of adaptive stepsizes.
>
> RESPONSE:
>
> Although [1] also studied the asymptotic properties of a class of adaptive stochastic optimization algorithms, our work has major differences from it, especially in the learning rate. The algorithm in [1] is as follows $$\theta_{n+1}=\theta_{n}-\gamma_{n+1}\frac{\nabla_{\theta_{n}}g(\theta_{n},\xi_{n})}{\sqrt{\omega_{n}+\epsilon}},$$ $$ \omega_{n+1}=\omega_{n}+\gamma_{n+1}(p_{n}\|\nabla_{\theta_{n}}g(\theta_{n},\xi_{n})\|^{2}-q_{n}\omega_{n}).$$
> First, as we can see that $\omega_{n}$ does not include the current gradient $\nabla_{\theta_{n}}g(\theta_{n},\xi_{n})$,  which makes it different from the term $S_n$ in our paper. Second, [1] requires $$\gamma_{n}\rightarrow+\infty, \lim_{n\rightarrow+\infty}q_{n}>0,$$which however cannot make $\gamma_{n+1}/\sqrt{\omega_{n}+\epsilon}=1/\sqrt{\sum_{t=1}^{n}\|\nabla_{\theta_{t}}g(\theta_{t},\xi_{n})\|^{2}+\epsilon}$. In other words,  the learning rate in [1] cannot cover the case of the AdaGrad in our paper. Note that the existence of the current gradient in the learning rate increases the difficulty in the convergence analysis.
>
> Regarding [2], we have also cited and discussed it in the updated version. One major difference between our work and [2] is that the authors in [2] studied the asymptotic property of $g(\bar{\theta}_n)$, where $\bar{\theta}_n$ is constructed in a time-average way as follows $$\bar{\theta}_0=0, \bar{\theta}\_{n+1} = \omega_n \theta_n + (1-\omega_n)\bar{\theta}_n,$$ where $\omega_n \in [0,1]$ is a parameter. If we let $\bar{\theta}\_{n+1}=\theta_n$, which means $\omega_n=1$. According to the requirement in [2],
> $$\omega_n = \frac{2 \eta_n}{\sum\_{j=0}^{n} \eta_j} = 1.$$
> As a result, $2\eta_n=\sum\_{j=0}^{n}\eta_j$, which however contradicts with the required condition in [2] that $\eta_n$ is decreasing and $\sum\_{n=1}^{+\infty} \eta_n=+\infty$. Therefore, the result in [2] cannot cover the case studied in our work. Actually, it can be proved that our results are more general than [2] in the sense that the asymptotic convergence of $\theta_n$ can lead to asymptotic convergence of $\bar{\theta}_n$.
>
> COMMENT 2:
>
> Less important ones:
>
> Before equation (5), you cite (Ward et al., 2019; Duchi et al., 2011) for the particular form of stepsizes you consider. However, both citations are inexact. First, Duchi et al. only considered coordinate-wise stepsizes, not the norm version you consider. The norm version was actually proposed in [3] and later proposed as a general scheme in [4].
>
> [3] M. Streeter and H. B. McMahan. Less regret via online conditioning, 2010. https://arxiv.org/abs/1002.4862
>
> [4] F. Orabona and D. Pál. Scale-free algorithms for online linear optimization. In International Conference on Algorithmic Learning Theory, 2015. https://arxiv.org/abs/1502.05744
>
> Moreover, even if many other authors called them "AdaGrad" stepsizes, another name should be used to distinguish the norm version from the coordinate-wise version.
>
> RESPONSE:
>
> Thank you for pointing out this mistake, which has been fixed in the new version. We totally agree with you that the coordinate-wise version and the norm version are different, which has been  clarified  in the new version.
>
> COMMENT 3:
>
> [5] X. Li and F. Orabona. A High Probability Analysis of Adaptive SGD with Momentum. Workshop on Beyond First Order Methods in ML Systems at ICML 2020 https://arxiv.org/abs/2007.14294
>
> High probability analysis of SGD with momentum, even with AdaGrad stepsizes (coordinate-wise), but without current gradient
>
> RESPONSE:
>
> In the former version, we cited another paper of theirs. In the revised version, we have cited and discussed this paper as well.
>
> COMMENT 4:
>
> [6] H. B. McMahan and M. J. Streeter. Adaptive bound optimization for online convex optimization. In COLT, 2010
>
> Paper [6] also introduces exactly the same algorithm of AdaGrad, at the very same conference of the Duchi et al. paper. It is often forgotten for no good reasons.
>
> RESPONSE:
>
> Thank for your reminder. But we think this work is about a new form of regularization, and it has less connection to our work.

---

> > ### Comment · Area_Chair_GkmD · 2021-11-20
> > **Still some things to discuss**
> >
> > Dear authors,
> >
> > thanks for your reply.
> > I agree that [1] does not include the current gradient, thanks for pointing it out.
> >
> > Regarding [2], you seem to have missed their results on the convergence of the last iterate. Could you please comment on that ones?
> >
> > Regarding [6], you don't seem to have realized that their Algorithm 1 is essentially AdaGrad. This is well-known and it is also plainly acknowledged in the journal version of the AdaGrad paper (https://jmlr.org/papers/volume12/duchi11a/duchi11a.pdf), page 8.

---

> > > ### Author Response · Authors · 2021-11-20
> > > **Response to further comments of Area Chair GkmD**
> > >
> > > Dear Area Chair,
> > >
> > > Thank you for the reminder.
> > >
> > > 1. Regarding the results in [2] on the last iterate: the setting for parameters of Theorem 13 in [2] cannot cover our setting for parameters, including step size $\epsilon_n$ and momentum coefficient $\alpha$. The latter setting is widely used in practical applications. We have revised page 5 of our paper to discuss the difference. Let us elaborate on why the setting for parameters in [2] cannot cover ours.
> > >
> > > The algorithm in Theorem 13 of [2] is as follows
> > > $$\theta_{n+1}=\theta_{n}-\tau_{n}\nabla_{\theta_{n}}g(\theta_{n},\xi_{n})+\beta_{n}(\theta_{n}-\theta_{n-1}),$$
> > > where $\tau_{n}=\frac{\eta_{n}}{1+\lambda_{n+1}}$, $\beta_{n}=\frac{\lambda_{n}}{1+\lambda_{n+1}},$
> > > and
> > > $$\lambda_{n}=\frac{\sum_{k=1}^{n-1}\eta_{k}}{4\eta_{n}}, (*)$$
> > > whereas, for mSGD given in (3) of our paper, $\tau_n = \epsilon_n$ and $\beta_n \equiv \alpha$.
> > > Moreover, Theorem 13 of [2] requires the following conditions
> > >
> > > $\eta_{k}$  is decreasing $(**),$
> > > $$\sum\_{k=1}^{+\infty}\eta\_{k}=+\infty,$$
> > > $$\sum\_{k=1}^{+\infty}\eta\_{k}^{2}\sigma^{2}<+\infty,$$
> > > and
> > > $$\sum\_{k=1}^{+\infty}\frac{\eta\_{k}}{\sum\_{i=1}^{k}\eta_{i}}=+\infty \quad  (\*\*\*).$$
> > >
> > > It can be proved that $\limsup_{n\rightarrow+\infty}\lambda_{n}/\lambda_{n+1}=1$. Otherwise, assume that $\limsup\_{n\rightarrow+\infty}\lambda\_{n}/\lambda\_{n+1}=t<1$. Note it follows from $(\*)$ and $(\*\*)$ that
> > > $$\lambda\_{n}=\frac{\sum\_{k=1}^{n-1}\eta\_{k}}{4\eta\_{n}}>\frac{\sum\_{k=1}^{n-1}\eta\_{k}}{4\eta\_{1}} \to +\infty.$$
> > > If $\limsup_{n\rightarrow+\infty}\lambda_{n}/\lambda_{n+1}=t<1$, then there exists $ N_{0}>0$, such that $\forall n\geq N_{0}$, it holds that $\lambda_{n+1}/\lambda_{n}>(t+1)/(2t) >1$. So $\forall n>N_{0}$, we have that
> > > \begin{equation}\nonumber\begin{aligned}
> > > \lambda_{n}=\lambda_{N_{0}}\prod_{k=N_{0}}^{n-1}\frac{\lambda_{k+1}}{\lambda_{k}}>\lambda_{N_{0}}\Big(\frac{t+1}{2t}\Big)^{n-N_{0}}.
> > > \end{aligned}\end{equation}
> > > Hence,
> > > \begin{equation}\nonumber\begin{aligned}
> > > \sum_{k=1}^{+\infty}\frac{\eta_{k}}{\sum_{i=1}^{k}\eta_{i}}<\sum_{k=1}^{+\infty}\frac{\eta_{k}}{\sum_{i=1}^{k-1}\eta_{i}}=\sum_{k=1}^{+\infty}\frac{1}{4\lambda_{k}}=\sum_{k=1}^{N_{0}-1}\frac{1}{4\lambda_{k}}+\sum_{k=N_{0}}^{+\infty}\frac{1}{4\lambda_{k}}<\sum_{k=1}^{N_{0}-1}\frac{1}{4\lambda_{k}}+\frac{1}{4\lambda_{N_{0}}}\sum_{k=N_{0}}^{+\infty}\Big(\frac{2t}{t+1}\Big)^{n-N_{0}}<+\infty,
> > > \end{aligned}\end{equation}
> > > which contradicts $(***)$. Therefore, $\limsup_{n\rightarrow+\infty}\lambda_{n}/\lambda_{n+1}\ge1$. Furthermore, from (*) it follows that $\lambda_n$ is increasing, so $\limsup_{n\rightarrow+\infty}\lambda_{n}/\lambda_{n+1}\le 1$, implying $\limsup_{n\rightarrow+\infty}\lambda_{n}/\lambda_{n+1}=1$.
> > >
> > > As a consequence,
> > > \begin{equation}\nonumber\begin{aligned}
> > > &\limsup_{n\rightarrow+\infty}\beta_{n}=\limsup_{n\rightarrow+\infty}\frac{\lambda_{n}}{1+\lambda_{n+1}}=\frac{1}{\lim_{n+\infty}\frac{1}{\lambda_{n}}+\liminf_{n\rightarrow+\infty}\frac{\lambda_{n+1}}{\lambda_{n}}}=1.
> > > \end{aligned}\end{equation}
> > >
> > > In addition, it holds that
> > > \begin{equation}\nonumber\begin{aligned}
> > > &\sum_{n=1}^{+\infty}\tau_{n}=\sum_{n=1}^{+\infty}\frac{\eta_{n}}{1+\lambda_{n+1}}<\sum_{n=1}^{+\infty}\frac{\eta_{n}}{\lambda_{n+1}}=\sum_{n=1}^{+\infty}\frac{4\eta_{n}\eta_{n+1}}{\sum_{k=1}^{n}\eta_{k}}<\sum_{n=1}^{+\infty}\frac{4\eta_{n}^{2}}{\eta_{1}}<\frac{4}{\eta_{1}\sigma^{2}}\sum_{n=1}^{+\infty}\eta_{n}^{2}\sigma^{2}<+\infty.
> > > \end{aligned}\end{equation}
> > >
> > > To sum up, the conditions for parameters of Theorem 13 in [2] imply that $\limsup_{n\rightarrow+\infty}\beta_n=1$ and
> > > $\sum_{n=1}^{+\infty}\tau_{n} < +\infty$. However, the conditions for $\epsilon_n$ and $\alpha$ in our paper are $\beta_n \equiv \alpha \in [0,1)$ and $\sum_{n=1}^{+\infty}\tau_{n} = \sum_{n=1}^{+\infty}\epsilon_{n}=+\infty$.
> > > Therefore, the setting in [2] cannot cover ours. Besides, there is no dependence between our two parameters, $\alpha$ and $\epsilon_{n}$, but parameters $\tau_{n}$ and $\beta_{n}$ in [2] are determined by $\{\eta_{n}\}$. It is more flexible to tune the parameters of mSGD in our paper, comparing with the parameters in [2].
> > >
> > > 2. Regarding [6]: Thank you for pointing out the mistake. We have now cited the paper in the revised version.

---

### Author Response · Authors · 2021-11-19
**Feedback request**

Dear  Reviewer ecdk and Reviewer qmsj,

      Many thanks for your comments and suggestions, which have helped us improve the paper considerably.

      Since we haven't received your feedback, we would appreciate it if you can inform us whether you are satisfied with the revised paper. If you think we have addressed your comments properly, please re-evaluate the recommendation level of this work.

     Thanks a lot!

   Best regards,

All authors of Paper 3772

---

### Decision · Program_Chairs · 2022-01-20

**Decision:**

Accept (Poster)

**Comment:**

The paper presents an asymptotic analysis of the convergence of the last iterate of mSGD and Adagrad. This result extends previous work providing stronger results under weaker assumptions. Even if these topics received less attention from the community, they are key problems in stochastic optimization.

The reviewers and I had several doubts about the proofs and relation with previous work. However, the rebuttal phase essential acted as a minor revision process. In fact, the authors fixed all the issues, convincing the reviewers (and me) that the results are novel, correct, and interesting.

For the above reasons, I recommend the acceptance of this paper.